# A Single-Loop Smoothed Gradient Descent-Ascent Algorithm for Nonconvex-Concave Min-Max Problems

**Jiawei Zhang**
216019001@link.cuhk.edu.cn [*]

**Peijun Xiao**
peijunx2@illinois.edu [†]

**Ruoyu Sun**
ruoyus@illinois.edu [†]

**Zhi-Quan Luo**
luozq@cuhk.edu.cn [*]

## Abstract

Nonconvex-concave min-max problem arises in many machine learning applications including minimizing a pointwise maximum of a set of nonconvex functions and robust adversarial training of neural networks. A popular approach to solve this problem is the gradient descent-ascent (GDA) algorithm which unfortunately can exhibit oscillation in case of nonconvexity. In this paper, we introduce a "smoothing" scheme which can be combined with GDA to stabilize the oscillation and ensure convergence to a stationary solution. We prove that the stabilized GDA algorithm can achieve an $O(1/\epsilon^2)$ iteration complexity for minimizing the pointwise maximum of a finite collection of nonconvex functions. Moreover, the smoothed GDA algorithm achieves an $O(1/\epsilon^4)$ iteration complexity for general nonconvex-concave problems. Extensions of this stabilized GDA algorithm to multi-block cases are presented. To the best of our knowledge, this is the first algorithm to achieve $O(1/\epsilon^2)$ for a class of nonconvex-concave problem. We illustrate the practical efficiency of the stabilized GDA algorithm on robust training.

## 1 Introduction

Min-max problems have drawn considerable interest from the machine learning and other engineering communities. They appear in applications such as adversarial learning [1–3], robust optimization [4–7], empirical risk minimization [8, 9], and reinforcement learning [10, 11]. Concretely speaking, a min-max problem is in the form:

$$\min_{x\in X} \max_{y\in Y} f(x,y), \tag{1.1}$$

where $X \subseteq \mathbb{R}^n$ and $Y \in \mathbb{R}^m$ are convex and closed sets and $f$ is a smooth function. In the literature, the convex-concave min-max problem, where $f$ is convex in $x$ and concave in $y$, is well-studied [12–19]. However, many practical applications involve nonconvexity, and this is the focus of the current paper. Unlike the convex-concave setting where we can compute the global stationary solution efficiently, to obtain a global optimal solution for the setting where $f$ is nonconvex with respect to $x$ is difficult.

---

[*]Shenzhen Research Institute of Big Data, School of science and engineering, The Chinese University of Hong Kong, Shenzhen, China

[†]Coordinated Science Laboratory, Department of ISE, University of Illinois at Urbana-Champaign, Urbana, IL

| ALGORITHM | COMPLEXITY | SIMPLICITY | MULTI-BLOCK |
|:---:|:---:|:---:|:---:|
| [22] | $\mathcal{O}(1/\epsilon^{2.5})$ | TRIPLE-LOOP | ✗ |
| [23] | $\mathcal{O}(1/\epsilon^{2.5})$ | TRIPLE-LOOP | ✗ |
| [20] | $\mathcal{O}(1/\epsilon^{3.5})$ | DOUBLE-LOOP | ✗ |
| [24] | $\mathcal{O}(1/\epsilon^{4})$ | SINGLE-LOOP | ✔ |
| THIS PAPER | $\mathcal{O}(1/\epsilon^{2})$ | SINGLE-LOOP | ✔ |

**Table 1:** Comparison of the algorithm in this paper with other works in solving problem (1.2). Our algorithm has a better convergence rate among others, and the single-loop and multi-block design make the algorithm suitable for solving large-scale problems efficiently.

In this paper, we consider the nonconvex-concave min-max problem (1.1) where $f$ is nonconvex in $x$ but concave of $y$, as well as a special case in the following form:

$$\min_x \max_{y \in Y} F(x)^T y, \tag{1.2}$$

where $Y = \{(y_1, \cdots, y_m)^T \mid \sum_{i=1}^m y_i = 1, y_i \geq 0\}$ is a probability simplex and $F(x) = (f_1(x), f_2(x), \cdots, f_m(x))^T$ is a smooth map from $\mathbb{R}^n$ to $\mathbb{R}^m$. Note that (1.2) is equivalent to the problem of minimizing the point-wise maximum of a finite collection of functions:

$$\min_x \max_{1 \leq i \leq m} f_i(x). \tag{1.3}$$

If $f_i(x) = g(x, \xi_i)$ is a loss function or a negative utility function at a data point $\xi_i$, then problem (1.3) is to find the best parameter of the worst data points. This formulation is frequently used in machine learning and other fields. For example, adversarial training [3,20], fairness training [20] and distribution-agnostic meta-learning [21] can be formulated as (1.3). We will discuss the formulations for these applications in details in Section 2.

Recently, various algorithms have been proposed for nonconvex-concave min-max problems [20, 22–28]. These algorithms can be classified into three types based on the structure: single-loop, double-loop and triple loop. Here a single-loop algorithm is an iterative algorithm where each iteration step has a closed form update, while a double-loop algorithm uses an iterative algorithm to approximately solve the sub-problem at each iteration. A triple-loop algorithm uses a double-loop algorithm to approximately solve a sub-problem at every iteration. To find an $\epsilon$-stationary solution, double-loop and tripe-loop algorithms have two main drawbacks. First, these existing multi-loop algorithms require at least $\mathcal{O}(1/\epsilon^2)$ outer iterations, while the iteration numbers of the other inner loop(s) also depend on $\epsilon$. Thus, the iteration complexity of the existing multi-loop algorithms is more than $\mathcal{O}(1/\epsilon^2)$ for (1.2). Among all the existing algorithms, the best known iteration complexity is $\mathcal{O}(1/\epsilon^{2.5})$ from two triple-loop algorithms [22, 23]. Since the best-known lower bound for solving (1.2) using first-order algorithms is $\mathcal{O}(1/\epsilon^2)$, so there is a gap between the existing upper bounds and the lower bound. Another drawback of multi-loop algorithms is their difficulty in solving problems with multi-block structure, since the acceleration steps used in their inner loops cannot be easily extended to multi-block cases, and a standard double-loop algorithm without acceleration can be very slow. This is unfortunate because the min-max problems with block structure is important for distributed training [24] in machine learning and signal processing.

Due to the aforementioned two drawbacks of double-loop and triple-loops algorithms, we focus in this paper on single-loop algorithms in hope to achieve the optimal iteration complexity $\mathcal{O}(1/\epsilon^2)$ for the nonconvex-concave problem (1.2). Notice that the nonconvex-concave applications in the aforementioned studies [20, 22–28] can all be formulated as (1.2), although the iteration complexity results derived in these papers are only for general nonconvex-concave problems. In other words, the structure of (1.2) is not used in the theoretical analysis. One natural question to ask is: **can we design a single loop algorithm with an iteration complexity lower than $\mathcal{O}(1/\epsilon^{2.5})$ for the min-max problem** (1.2)**?**

**Existing Single-loop algorithms.** A simple single-loop algorithm is the so-called Gradient Descent Ascent (GDA) which alternatively performs gradient descent to the minimization problem and gradient ascent to the maximization problem. GDA can generate an $\epsilon$-stationary solution for a nonconvex-strongly-concave problem with iteration complexity $\mathcal{O}(1/\epsilon^2)$ [28]. However, GDA will oscillate with constant stepsizes around the solution if the maximization problem is not strongly

concave [19]. So the stepsize should be proportional to $\epsilon$ if we want an $\epsilon$-solution. These limitations slow down GDA which has an $\mathcal{O}(1/\epsilon^5)$ iteration complexity for nonconvex-concave problems. Another single-loop algorithm [24] requires diminishing step-sizes to guarantee convergence and its complexity is $\mathcal{O}(1/\epsilon^4)$. [29] also proposes a single-loop algorithm for min-max problems by performing GDA to a regularized version of the original min-max problem and the regularization term is diminishing. The iteration complexity bounds given in the references [24, 28, 29] are worse than the ones from multi-loop algorithms using acceleration in the subproblems.

In this paper, we propose a single-loop "smoothed gradient descent-ascent" algorithm with optimal iteration complexity for the nonconvex-concave problem (1.2). Inspired by [30], to fix the oscillation issue of GDA discussed above, we introduce an exponentially weighted sequence $z^t$ of the primal iteration sequence $x^t$ and include a quadratic proximal term centered at $z^t$ to objective function. Then we perform a GDA step to the proximal function instead of the original objective. With this smoothing technique, an $\mathcal{O}(1/\epsilon^2)$ iteration complexity can be achieved for problem (1.2) under mild assumptions. Our contributions are three fold.

- **Optimal order in convergence rate.** We propose a single-loop algorithm **Smoothed-GDA** for nonconvex-concave problems which finds an $\epsilon$-stationary solution within $\mathcal{O}(1/\epsilon^2)$ iterations for problem (1.2) under mild assumptions.

- **General convergence results.** The **Smoothed-GDA** algorithm can also be applied to solve general nonconvex-concave problems with an $\mathcal{O}(1/\epsilon^4)$ iteration complexity. This complexity is the same as in [24]. However, the current algorithm does not require the compactness of the domain $X$, which significantly extends the applicability of the algorithm.

- **Multi-block settings.** We extend the **Smoothed-GDA** algorithm to the multi-block setting and give the same convergence guarantee as the one-block case.

The paper is organized as follows. In Section 2, we describe some applications of nonconvex-concave problem (1.2) or (1.3). The details of the **Smoothed-GDA** algorithm as well as the main theoretical results are given in Section 3. The proof sketch is given in Section 4. The proofs and the details of the numerical experiments are in the appendix.

## 2   Representative Applications

We give three application examples which are in the min-max form (1.2).

**1. Robust learning from multiple distributions.** Suppose the data set is from $n$ distributions: $D_1, \cdots, D_n$. Each $D_i$ is a different perturbed version of the underlying true distribution $D_0$. Robust training is formulated as minimizing the maximum of expected loss over the $n$ distributions as

$$\min_{x \in X} \max_i \mathbb{E}_{a \sim D_i}[F(x; a)] = \min_{x \in X} \max_{y \in Y} \sum_{i=1}^{m} y_i f_i(x), \qquad (2.1)$$

where $Y$ is a probability simplex, $F(x; a)$ represents the loss with model parameter $x$ on a data sample $a$. Notice that $f_i(x) = \mathbb{E}_{a \sim D_i}[F(x; a)]$ is the expected loss under distribution $D_i$. In adversarial learning [3, 31, 32], $D_i$ corresponds to the distribution that is used to generate adversarial examples. In Section 5, we will provide a detailed formulation of adversarial learning on the data set MNIST and apply the Smoothed GDA algorithm to this application.

**2. Fair models**. In machine learning, it is common that the models may be unfair, i.e. the models might discriminate against individuals based on their membership in some group [33, 34]. For example, an algorithm for predicting a person's salary might use that person's protected attributes, such as gender, race, and color. Another example is training a logistic regression model for classification which can be biased against certain categories. To promote fairness, [35] proposes a framework to minimize the maximum loss incurred by the different categories:

$$\min_{x \in X} \max_i f_i(x), \qquad (2.2)$$

where $x$ represents the model parameters and $f_i$ is the corresponding loss for category $i$.

**3. Distribution-agnostic meta-learning.** Meta-learning is a field about learning to learn, i.e. to learn the optimal model properties so that the model performance can be improved. One popular

choice of meta-learning problem is called gradient-based Model-Agnostic Meta-Learning (MAML) [36]. The goal of MAML is to learn a good global initialization such that for any new tasks, the model still performs well after one gradient update from the initialization.

One limitation of MAML is that it implicitly assumes the tasks come from a particular distribution, and optimizes the expected or sample average loss over tasks drawn from this distribution. This limitation might lead to arbitrarily bad worst-case performance and unfairness. To mitigate these difficulties, [21] proposed a distribution-agnostic formulation of MAML:

$$\min_{x \in X} \max_i f_i(x - \alpha \nabla f_i(x)). \tag{2.3}$$

Here, $f_i$ is the loss function associated with the $i$-th task, $x$ is the parameter taken from the feasible set $X$, and $\alpha$ is the stepsize used in the MAML for the gradient update. Notice that each $f_i$ is still a function over $x$, even though we take one gradient step before evaluating the function. This formulation (2.3) finds the initial point that minimizes the objective function after one step of gradient over all possible loss functions. It is shown that solving the distribution-agnostic meta-learning problem improves the worst-case performance over that of the original MAML [21] across the tasks.

## 3  Smoothed GDA Algorithm and Its Convergence

Before we introduce the Smoothed-GDA algorithm, we first define the stationary solution and the $\epsilon$-stationary solution of problem (1.1).

**Definition 3.1** *Let $\mathbf{1}_X(x), \mathbf{1}_Y(y)$ be the indicator functions of the sets $X$ and $Y$ respectively. A pair $(x, y)$ is an $\epsilon$-solution set of problem* (1.1) *if there exists a pair $(u, v)$ such that*

$$u \in \nabla_x f(x, y) + \partial \mathbf{1}_X(x), \quad v \in -\nabla_y f(x, y) + \partial \mathbf{1}_Y(y), \quad \text{and} \quad \|u\|, \|v\| \le \epsilon, \tag{3.1}$$

*where $\partial g(\cdot)$ denotes the sub-gradient of a function $g$. A pair $(x, y)$ is a stationary solution if $u = 0, v = 0$.*

**Definition 3.2** *The projection of a point $y$ onto a set $X$ is defined as $P_X(y) = \operatorname{argmin}_{x \in X} \frac{1}{2} \|x - y\|^2$.*

### 3.1  Smoothed Gradient Descent Ascent (Smoothed-GDA)

A simple algorithm for solving min-max problems is the Gradient Descent Ascent (GDA) algorithm (Algorithm 1), which performs a gradient descent to the $\min$ problem and a gradient ascent to the $\max$ problem alternatively. It is well-known that with constant step size, GDA can oscillate between iterates and fail to converge even for a simple bilinear min-max problem: $\min_{x \in \mathbb{R}^n} \max_{y \in \mathbb{R}^n} x^T y$.

To fix the oscillation issue, we introduce a "smoothing" technique to the primal updates. Note that smoothing is a common technique in traditional optimization such as Moreau-Yosida smoothing [37] and Nesterov's smoothing [38]. More concretely, we introduce an auxiliary sequence $\{z^t\}$ and define a function $K(x, z; y)$ as

$$K(x, z; y) = f(x, y) + \frac{p}{2}\|x - z\|^2, \tag{3.2}$$

where $p > 0$ is a constant, and we perform gradient descent and gradient ascent alternatively on this function instead of the original function $f(x, y)$. After performing one-step of GDA to the function $K(x^t, z^t; y^t)$, $z^t$ is updated by an averaging step. The "Smoothed GDA" algorithm is formally presented in Algorithm 2. Note that our algorithm is different from the one in [29], as [29] uses an regularization term $\alpha_t(\|x\|^2 - \|y\|^2)$ and requires this term to diminishing.

---

**Algorithm 1** GDA

1: Initialize $x^0, y^0$;
2: Choose $c, \alpha > 0$;
3: **for** $t = 0, 1, 2, \ldots,$ **do**
4:   $x^{t+1} = P_X(x^t - c\nabla_x f(x^t, y^t));$
5:   $y^{t+1} = P_Y(y^t + \alpha \nabla_y f(x^{t+1}, y^t));$
6: **end for**

---

**Algorithm 2** Smoothed-GDA

1: Initialize $x^0, z^0, y^0$ and $0 < \beta \le 1$.
2: **for** $t = 0, 1, 2, \ldots,$ **do**
3:   $x^{t+1} = P_X(x^t - c\nabla_x K(x^t, z^t; y^t));$
4:   $y^{t+1} = P_Y(y^t + \alpha \nabla_y K(x^{t+1}, z^t; y^t));$

5:   $z^{t+1} = z^t + \beta(x^{t+1} - z^t),$
6: **end for**

---

Notice that when $\beta = 1$, Smoothed-GDA is just the standard GDA. Furthermore, if the variable $x$ has a block structure, i.e., $x$ can be decomposed into $N$ blocks as

$$x = (x_1^T, \cdots, x_N^T)^T,$$

then Algorithm 2 can be extended to a multi-block version which we call the Smoothed Block Gradient Descent Ascent (Smoothed-BGDA) Algorithm (see Algorithm 3). In the multi-block version, we update the primal variable blocks alternatingly and use the same strategy to update the dual variable and the auxiliary variable as in the single-block version.

---

**Algorithm 3** Smoothed Block Gradient Descent Ascent (Smoothed-BGDA)

---

1: Initialize $x^0, z^0, y^0$;
2: **for** $t = 0, 1, 2, \ldots,$ **do**
3:     **for** $i = 1, 2, \ldots, N$ **do**
4:         $x_i^{t+1} = P_X(x_i^t - c\nabla_{x_i} K(x_1^{t+1}, x_2^{t+1}, \cdots, x_{i-1}^{t+1}, x_i^t, \cdots, x_N^t, z^t; y^t));$
5:     **end for**
6:     $y^{t+1} = P_Y(y^t + \alpha\nabla_y K(x^{t+1}, z^t; y^t));$
7:     $z^{t+1} = z^t + \beta(x^{t+1} - z^t)$, where $0 < \beta \leq 1$;
8: **end for**

---

## 3.2 Iteration Complexity for Nonconvex-concave Problems

In this subsection, we present the iteration complexities of Algorithm 2 and Algorithm 3 for general nonconvex-concave problems (1.1). We first state some basic assumptions.

**Assumption 3.3** *We assume the following.*

1. *$f(x, y)$ is smooth and the gradients $\nabla_x f(x, y), \nabla_y f(x, y)$ are L-Lipschitz continuous.*

2. *$Y$ is a closed, convex and compact set of $\mathbb{R}^m$. $X$ is a closed and convex set.*

3. *The function $\psi(x) = \max_{y \in Y} f(x, y)$ is bounded from below by some finite constant $\underline{f} > -\infty$.*

**Theorem 3.4** *Consider solving problem (1.1) by Algorithm 2 (or Algorithm 3). Suppose Assumption 3.3 holds, and we choose the algorithm parameters to satisfy $p > 3L$, $c < 1/(p + L)$ and*

$$\alpha < \min\left\{\frac{1}{11L}, \frac{c^2(p-L)^2}{4L(1+c(p-L))^2}\right\}, \beta \leq \min\left\{\frac{1}{36}, \frac{(p-L)^2}{384p(p+L)^2}\right\}. \tag{3.3}$$

*Then, the following holds:*

- *(One-block case) For any integer $T > 0$, if we further let $\beta < 1/\sqrt{T}$, then there exists a $t \in \{1, 2, \cdots, T\}$ such that $(x^{t+1}, y^{t+1})$ is a $\mathcal{O}(T^{-1/4})$-stationary solution. This means we can obtain an $\epsilon$-stationary solution within $\mathcal{O}(\epsilon^{-4})$ iterations.*

- *(Multi-block case) If we replace the condition of $\alpha$ in Algorithm 3 by*

$$\alpha \leq \min\left\{\frac{1}{11L}, \frac{c^2(p-L)^2}{4L(1+c(p+L)N^{3/2}+c(p-L))^2}\right\} \tag{3.4}$$

*and further require $\beta \leq \epsilon^2$, then we can obtain an $\epsilon$-stationary solution within $\mathcal{O}(\epsilon^{-4})$ iterations of Algorithm 3.*

**Remark.** The reference [24] derived the same iteration complexity of $\mathcal{O}(\epsilon^{-4})$ under the additional compactness assumption on $X$. This assumption may not be satisfied for some applications where $X$ can the entire space.

## 3.3 Convergence Results for Minimizing the Point-wise Maximum of Finite Functions

Now we state the improved iteration complexity results for the special min-max problem (1.2). We claim that our algorithms (Algorithm 2 and Algorithm 3) can achieve the optimal order of iteration complexity of $\mathcal{O}(\epsilon^{-2})$ in this case.

For any stationary solution of (1.2) denoted as $(x^*, y^*)$, the following KKT conditions hold:

$$\nabla F(x^*)y^* = 0, \tag{3.5}$$

$$\sum_{i=1}^{m} y_i^* = 1, \tag{3.6}$$

$$y_i^* \geq 0, \forall i \in [m] \tag{3.7}$$

$$\mu - \nu_i = f_i(x^*), \forall i \in [m], \tag{3.8}$$

$$\nu_i \geq 0, \nu_i y_i^* = 0, \forall i \in [m], \tag{3.9}$$

where $\nabla F(x)$ denotes the Jacobian matrix of $F$ at $x$, while $\mu$, $\nu$ are the multipliers for the equality constraint $\sum_{i=1}^{m} y_i = 1$ and the inequality constraint $y_i \geq 0$ respectively.

At any stationary solution $(x^*, y^*)$, only the functions $f_i(x^*)$ for any index $i$ with $y_i^* > 0$ contribute to the objective function $\sum_{i=1}^{N} y_i^* f_i(x^*)$ and they correspond to the worst cases in the robust learning task. In other words, any function $f_i(\cdot)$ with $y_i^* > 0$ at $(x^*, y^*)$ contains important information of the solution. We denote a set $\mathcal{I}_+(y^*)$ to represent the set of indices for which $y_i^* > 0$. We will make a mild assumption on this set.

**Assumption 3.5** *For any* $(x^*, y^*)$ *satisfying* (3.5)*, we have* $\nu_i > 0, \forall i \notin \mathcal{I}_+(y^*)$.

**Remark.** The assumption is called "strict complementarity", a common assumption in the field of variation inequality [39, 40] which is closely related to the study of min-max problems. This assumption is used in many other optimization papers [6, 41–44]. Strict complementarity is generically true (i.e. holds with probability 1) if there is a linear term in the objective function and the data is from a continuous distribution (similar to [30, 44]). Moreover, we will show that we can prove Theorem 3.8 using a weaker regularity assumption rather than the strict complementarity assumption:

**Assumption 3.6** *For any* $(x^*, y^*) \in W^*$*, the matrix* $M(x^*)$ *is of full column rank, where*

$$M(x^*) = \left\{ J_{\mathcal{T}(x^*)} \quad \mathbf{1} \right\}.$$

We say that Assumption 3.6 is weaker since the strict complementarity assumption (Assumption 3.5) can imply Assumption 3.6 according to Lemma D.7 in the appendix. In the appendix, we will see that Assumption 3.6 holds with probability 1 for a robust regression problem with a square loss (see Proposition E.7).

We also make the following common "bounded level set" assumption.

**Assumption 3.7** *The set* $\{x \mid \psi(x) \leq R\}$ *is bounded for any* $R > 0$. *Here* $\psi(x) = \max_{y \in Y} f(x, y)$.

**Remark.** This bounded-level-set assumption is to ensure the iterates would stay bounded. Actually, assuming the iterates $\{z^t\}$ are bounded will be enough for our proof. The bounded level set assumption, a.k.a. coerciveness assumption, is widely used in many papers [45–47]. Bounded-iterates-assumption itself is also common in optimization [42, 48, 49]. In practice, people usually add a regularizer to the objective function to make the level set and the iterates bounded (see [50] for a neural network example).

**Theorem 3.8** *Consider solving problem 1.2 by Algorithm 2 or Algorithm 3. Suppose that Assumption 3.3, 3.5 holds and either Assumption 3.7 holds or assume* $\{z^t\}$ *is bounded. Then there exist constants* $\beta'$ *and* $\beta''$ *(independent of* $\epsilon$ *and* $T$*) such that the following holds:*

1. *(One-block case) If we choose the parameters in Algorithm 2 as in* (3.3) *and further let* $\beta < \beta'$ *, then*

(a) *Every limit point of $(x^t, y^t)$ is a solution of* (1.2).

(b) *The iteration complexity of Algorithm 2 to obtain an $\epsilon$-stationary solution is $\mathcal{O}(1/\epsilon^2)$.*

2. *(Multi-block case) Consider using Algorithm 3 to solve Problem 1.2. If we replace the condition for $\alpha$ in* (3.3) *by* (3.4) *and require $\beta$ satisfying $\beta < \epsilon^2$ and $\beta < \beta''$, then we have the same results as in the one-block case.*

# 4 Proof Sketch

In this section, we give a proof sketch of the main theorem on the one-block cases; the proof details will be given in the appendix.

## 4.1 The Potential Function and Basic Estimates

To analyze the convergence of the algorithms, we construct a potential function and study its behavior along the iterations. We first give the intuition why our algorithm works. We define the dual function $d(\cdot)$ and the proximal function $P(\cdot)$ as

$$d(y, z) = \min_{x \in X} K(x, z; y), \quad P(z) = \min_{x \in X}\{\max_{y \in Y} K(x, z; y)\}.$$

We also let

$$
\begin{aligned}
x(y, z) &= \arg\min_{x \in X} K(x, z; y), \\
x^*(z) &= \arg\min_{x \in X} \max_{y \in Y} K(x, z; y), \\
y_+^t(z^t) &= P_Y(y^t + \alpha \nabla_y K(x(y^t, z^t), z^t; y^t)).
\end{aligned}
$$

Notice that by Danskin's Theorem, we have $\nabla_y d(y, z) = \nabla_y K(x(y, z), z; y)$ and $\nabla_z P(z) = p(z - x^*(z))$. Recall in Algorithm 2, the update for $x^t, y^t$ and $z^t$ can be respectively viewed as a primal descent for the function $K(x^t, z^t; y^t)$, approximating dual ascent to the dual function $d(y^t, z^t)$ and approximating proximal descent to the proximal function $P(z^t)$. We define a potential function as follows:

$$\phi^t = \phi(x^t, y^t, z^t) = K(x^t, z^t; y^t) - 2d(y^t, z^t) + 2P(z^t), \tag{4.1}$$

which is a linear combination of the primal function $K(\cdot)$, the dual function $d(\cdot)$ and the proximal function $P(\cdot)$. We hope the potential function decreases after each iteration and is bounded from below. In fact, it is easy to prove that $\phi^t \geq \underline{f}$ for any $t$ (see appendix), but it is harder to prove the decrease of $\phi^t$. Since the ascent for dual and the descent for proximal is approximate, an error term occurs when estimating the decrease of the potential function. Hence, certain error bounds are needed.

Using some primal error bounds, we have the following basic descent estimate.

**Proposition 4.1** *Suppose the parameters of Algorithm 2 satisfy* (3.3)*, then*

$$
\begin{aligned}
\phi^t - \phi^{t+1} &\geq \frac{1}{8c}\|x^t - x^{t+1}\|^2 + \frac{1}{8\alpha}\|y^t - y_+^t(z^t)\|^2 + \frac{p\beta}{8}\|z^t - x^{t+1}\|^2 \tag{4.2} \\
&\quad -24p\beta\|x^*(z^t) - x(y_+^t(z^t), z^t)\|^2. \tag{4.3}
\end{aligned}
$$

We would like the potential function $\phi^t$ to decrease sufficiently after each iteration. Concretely speaking, we want to eliminate the negative term (4.3) and show that the following "sufficient-decrease" holds for each iteration $t$:

$$\phi^t - \phi^{t+1} \geq \frac{1}{16c}\|x^t - x^{t+1}\|^2 + \frac{1}{16\alpha}\|y^t - y_+^t(z^t)\|^2 + \frac{p\beta}{16}\|z^t - x^{t+1}\|^2. \tag{4.4}$$

It is not hard to prove that if (4.4) holds for $t \in \{0, 1, \cdots, T - 1\}$, then there exists a $t \in \{1, 2, \cdots, T\}$ such that $(x^t, y^t)$ is a $C/\sqrt{T\beta}$-solution for some constant $C > 0$. Moreover, if (4.4) holds for any $t$, then the iteration complexity is $\mathcal{O}(1/\epsilon^2)$ and we can also prove that every limit point of the iterates is a min-max solution. Therefore by the above analysis, the most important thing is to bound the term $\|x^*(z^t) - x(y_+^t(z^t), z^t)\|^2$, which is related to the so-call "dual error bound".

If $\|y^t - y^t_+(z^t)\| = 0$, then $y^t_+(z^t)$ is the maximizer of $d(y, z^t)$ over $y$, and thus $x^*(z^t)$ is the same as $x(y^t_+(z^t), z^t)$. A natural question is whether we can use the term $\|y^t - y^t_+(z^t)\|$ to bound $\|x^*(z^t) - x(y^t_+(z^t), z^t)\|^2$? The answer is yes, and we have the following "dual error bound".

**Lemma 4.2** *If Assumptions 3.3 and 3.5 hold for* (1.2) *and there is an $R > 0$ with $\|z^t\| \leq R$, then there exists $\delta > 0$ such that if*

$$\max\{\|x^t - x^{t+1}\|, \|y^t - y^t_+(z^t)\|, \|x^{t+1} - z^t\|\} \leq \delta,$$

*then*

$$\|x(y^t_+(z^t), z^t) - x^*(z^t)\| \leq \sigma_5 \|y^t - y^t_+(z^t)\|.$$

*holds for some constant $\sigma_5 > 0$.*

Using this lemma, we can prove Theorem 3.8. We choose $\beta$ sufficiently small, then when the residuals appear in (4.2) are large, we can prove that $\phi^t$ decreases sufficiently using the compactness of $Y$. When the residuals are small, the error bound Lemma 4.2 can be used to guarantee the sufficient decrease of $\phi^t$. Therefore, (4.4) always holds, which yields Theorem 3.8. However, for the general nonconvex-concave problem 1.1, we can only have a "weaker" bound.

**Lemma 4.3** *Suppose Assumption 3.3 holds for problem 1.1. Define $D(Y)$ to be the diameter of $Y$. If Assumption 3.3 holds, we have*

$$(p - L)\|x^*(z^t) - x(y^t_+(z^t), z^t)\|^2 \leq (1 + \alpha L)\|y^t - y^t_+(z^t)\| \cdot D(Y)$$

Note that this is a nonhomogeneous error bound, which can help us bound the term $\|x^*(z^t) - x(y^t_+(z^t), z^t)\|$ only when $\|y^t - y^t_+(z^t)\|$ is not too small. Therefore, we say it is "weaker" dual error bound. To obtain an $\epsilon$-stationary solution, we need to choose $\beta$ sufficiently small and proportional to $\epsilon^2$. In this case, we can prove that if $\phi^t$ stops to decrease, we have already obtained an $\epsilon$-stationary solution by Lemma 4.3. By the remark after (4.4), we need $\mathcal{O}((1/(\epsilon\sqrt{\beta}))^2) = \mathcal{O}(1/\epsilon^4)$ iterations to obtain an $\epsilon$-stationary solution.

**Remark.** For the general nonconvex-concave problem (1.1), we need to choose $\beta$ proportional to $\epsilon^2$ and hence the iteration complexity is higher than the previous case. However, it is expected that for a concrete problem with some special structure, the "weaker" error bound Lemma 4.3 can be improved, as is the iteration complexity bound. This is left as a future work.

The proof sketch can be summarized in the following steps:

- In Step 1, we introduce the potential function $\phi^t$ which is shown to be bounded below. To obtain the convergence rate of the algorithms, we want to prove the potential function can make sufficient decrease at every iterate $t$, i.e., we want to show $\phi^t - \phi^{t+1} > 0$.

- In Step 2, we study this difference $\phi^t - \phi^{t+1}$ and provide a lower bound of it in Proposition 4.2. Notice that a negative term (4.3) will show up in the lower bound, and we have to carefully analyze the magnitude of this term to obtain $\phi^t - \phi^{t+1} > 0$.

- Analyzing the negative term is the main difficulty of the proof. In Step 3, we discuss how to deal with this difficulty for solving Problem 1.1 and Problem 1.2 separately.

- Finally, we show the potential function makes a sufficient decrease at every iterate as stated in (4.4), and conclude our proof by computing the number of iterations to achieve an $\epsilon$-solution (as shown in Lemma B.12 the appendix).

## 5 Numerical Results on Robust Neural Network Training

In this section, we apply the Smoothed-GDA algorithm to train a robust neural network on MNIST data set against adversarial attacks [3, 31, 32]. The optimization formulation is

$$\min_{\mathbf{w}} \sum_{i=1}^{N} \max_{\delta_i, \text{ s.t. } |\delta_i|_\infty \leq \varepsilon} \ell(f(x_i + \delta_i; \mathbf{w}), y_i), \tag{5.1}$$

where $\mathbf{w}$ is the parameter of the neural network, the pair $(x_i, y_i)$ denotes the $i$-th data point, and $\delta_i$ is the perturbation added to data point $i$. As (5.1) is difficult to solve directly, researchers [20] have

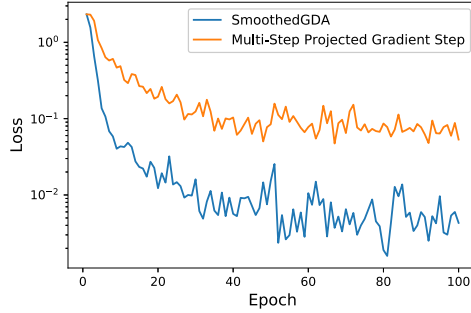

**Figure 1:** Convergence speed of Smoothed-GDA and the algorithm in [20].

proposed an approximation of (5.1) as the following nonconvex-concave problem, which is in the form of (1.2) we discussed before.

$$\min_{\mathbf{w}} \sum_{i=1}^{N} \max_{\mathbf{t} \in \mathcal{T}} \sum_{j=0}^{9} t_j \ell \left( f \left( x_{ij}^K; \mathbf{w} \right), y_i \right), \; \mathcal{T} = \left\{ (t_1, \cdots, t_m) \mid \sum_{i=1}^{m} t_i = 1, t_i \geq 0 \right\}, \quad (5.2)$$

where $K$ is a parameter in the approximation, and $x_{ij}^K$ is an approximated attack on sample $x_i$ by changing the output of the network to label $j$. The details of this formulation and the structure of the network in experiments are provided in the appendix.

| | Natural | FGSM $L_\infty$ [32] | | | PGD$^{40}$ $L_\infty$ [31] | | |
|---|---|---|---|---|---|---|---|
| | | $\varepsilon = 0.2$ | $\varepsilon = 0.3$ | $\varepsilon = 0.4$ | $\varepsilon = 0.2$ | $\varepsilon = 0.3$ | $\varepsilon = 0.4$ |
| [3] with $\varepsilon = 0.35$ | 98.58% | 96.09% | 94.82% | 89.84% | 94.64% | 91.41% | 78.67% |
| [51] with $\varepsilon = 0.35$ | 97.37% | 95.47% | 94.86% | 79.04% | 94.41% | 92.69% | 85.74% |
| [51] with $\varepsilon = 0.40$ | 97.21% | 96.19% | 96.17% | 96.14% | 95.01% | 94.36% | 94.11% |
| [20] with $\varepsilon = 0.40$ | 98.20% | 97.04% | 96.66% | **96.23%** | 96.00% | 95.17 % | 94.22% |
| Smoothed-GDA with $\varepsilon = 0.40$ | **98.89%** | **97.87%** | **97.23%** | 95.81% | **96.71%** | **95.62%** | **94.51%** |

**Table 2:** Test accuracies under FGSM and PGD attacks.

**Results:** We compare our results with three algorithms from [3, 20, 51]. The references [3, 51] are two classical algorithms in adversarial training, while the recent reference [20] considers the same problem formulation as (1.2) and has an algorithm with $\mathcal{O}(1/\epsilon^{3.5})$ iteration complexity. The accuracy of our formulation are summarized in Table 2 which shows that the formulation (1.2) leads to a comparable or slightly better performance to the other algorithms. We also compare the convergence on the loss function when using the Smoothed-GDA algorithm and the one in [20]. In Figure 1, Smoothed-GDA algorithm takes only 5 epochs to get the loss values below 0.2 while the algorithm proposed in [20] takes more than 14 epochs. In addition, the loss obtained from the Smoothed-GDA algorithm has a smaller variance.

## 6 Conclusion

In this paper, we propose a simple single-loop algorithm for nonconvex min-max problems (1.1). For an important family of problems (1.2), the algorithm is even more efficient due to the **dual error bound**, and it is well-suited for problems in large-size dimensions and distributed setting. The algorithmic framework is flexible, and hence in the future work, we can extend the algorithm to more practical problems and derive **stronger** error bounds to attain lower iteration complexity.

## Broader Impact

In this paper, we propose a single-loop algorithm for min-max problem. This algorithm is easy to implemented and proved to be efficient in a family of nonconvex minimax problems and have good numerical behavior in robust training. This paper focuses on theoretical study of the algorithms. In industrial applications, several aspects of impact can be expected:

1. **Save energy by improving efficiency.** The trick developed in this paper has the potential to accelerate the training for machine learning problems involving a minimax problem such robust training for uncertain data, generative adversarial net(GAN) and AI for games. This means that the actual training time will decrease dramatically by using our algorithm. Training neural network is very energy-consuming, and reducing the training time can help the industries or companies to save energy.

2. **Promote fairness.** We consider min-max problems in this paper. A model that is trained under this framework will not allow poor performance on some objectives in order to boost performance on the others. Therefore, even if the training data itself is biased, the model will not allow some objectives to contribute heavily to minimizing the average loss due to the min-max framework. In other words, this framework promotes fairness, and model that is trained under this framework will provide fair solutions to the problems.

3. **Provide flexible framework.** Our algorithmic framework is flexible. Though in the paper, we only discuss some general formulation, our algorithm can be easily extended to many practical settings. For example, based on our general framework for multi-block problems, we can design algorithms efficiently solving problems with distributedly stored data, decentralized control or privacy concern. Therefore, our algorithm may have an impact on some popular big data applications such as distributed training, federated learning and so on.

## Funding Disclosure

This research is supported by the leading talents of Guangdong Province program [Grant 00201501]; the National Science Foundation of China [Grant 61731018]; the Air Force Office of Scientific Research [Grant FA9550-12-1-0396]; the National Science Foundation [Grant CCF 1755847]; Shenzhen Peacock Plan [Grant KQTD2015033114415450]; the Development and Reform Commission of Shenzhen Municipality; and Shenzhen Research Institute of Big Data.

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
