[Supplementary Material 1]

In the appendix, we will give the proof of the main theorems. The appendix is organized as follows:

1. In section A, we list some notations used in the appendix.

2. In section B, we prove the two main theorems in one-block case.

3. In section C, we briefly state the proof of the two main theorems in multi-block setting.

4. In section D, we prove the two main error bound lemmas (Lemma 4.3 and Lemma 4.2) under the strict complementarity assumption.

5. In section E, we see that the strict complementarity assumption can be relaxed to a weaker regularity assumption. We also prove that this weaker regularity assumption is generic for robust regression problems with square loss, i.e., we prove that our regularity assumption holds with probability 1 if the data points are joint from a continuous distribution.

6. In the last section F, we give some more details about the experiment.

## A  Notations

We first list some notations which will be used in the appendix.

1. $[m] = \{1, 2, \cdots, m\}$.

2. $W^*$ is the solution set of (1.1) or (1.2). $X^*$ is the set of all solutions $x^*$, i.e., $x^* \in X^*$ if there exists a $y^*$ such that $(x^*, y^*) \in W^*$.

3. $\mathcal{B}(r)$ is a Euclidian ball of radius $r$ for proper dimension.

4. $\mathrm{dist}(v, S)$ means the Euclidian distance from a point $v$ to a set $S$.

5. For a vector $v$, $v_i$ means the $i$-th component of $v$. For a set $\mathcal{S}$, $v_{\mathcal{S}} \in \mathbb{R}^{|\mathcal{S}|}$ is the vector containing all components $v_i$'s with $i \in \mathcal{S}$.

6. Let $\mathbf{A} \in \mathbb{R}^{n \times m}$ be a matrix and $\mathcal{S} \subseteq [m]$ be an index set. Then $\mathbf{A}_{\mathcal{S}}$ represents the row sub-matrix of $A$ corresponding to the rows with index in $\mathcal{S}$.

7. For a matrix $\mathbf{M}$, $\gamma(\mathbf{M})$ is the smallest singular value of $\mathbf{M}$.

8. The projection of a point $y$, onto a set $X$ is defined as $P_X(y) = \mathrm{argmin}_{x \in X} \frac{1}{2} \|x - y\|^2$.

## B  Proof of the two main theorems: one-block case

In this section, we prove the two main theorems in one-block case. The proof of the multi-block case is similar and will be given in the next section.

**Proof Sketch.**

- In Step 1, we will introduce the potential function $\phi^t$ which is shown to be bounded below. To obtain the convergence rate of the algorithms, we want to prove the potential function can make sufficient decrease at every iterate $t$, i.e., we want to show $\phi^t - \phi^{t+1} > 0$.

- In Step 2, we will study this difference $\phi^t - \phi^{t+1}$ and provide a lower bound of it in Proposition 4.2. Notice that a negative term (4.3) will show up in the lower bound, and we have to carefully analyze the magnitude of this term to obtain $\phi^t - \phi^{t+1} > 0$.

- Analyzing the negative term will be the main difficulty of the proof. In Step 3, we will discuss how to deal with this difficulty for solving Problem 1.1 and Problem 1.2 separately.

- Finally, we will show the potential function makes a sufficient decrease at every iterate as stated in (4.4), and will conclude our proof by computing the number of iterations to achieve an $\epsilon$-solution in Lemma B.12.

### B.1  The potential function and basic estimate

Recall that the potential function is:

$$\phi^t = \Phi(x^t, z^t; y^t) = K(x^t, z^t; y^t) - 2d(y^t, z^t) + 2P(z^t),$$

where

$$
\begin{aligned}
K(x, z; y) &= f(x, y) + \frac{p}{2}\|x - z\|^2, \\
d(y, z) &= \min_{x \in X} K(x, z; y), \\
P(z) &= \min_{x \in X} \max_{y \in Y} K(x, z; y).
\end{aligned}
$$

Also note that if $p > L$, $K(x, z; y)$ is strongly convex of $x$ with modular $p - L$ and $\nabla_x K(x, z; y)$ is Lipschitz-continuous of $x$ with a constant $L + p$. We also use the following notations:

1. $h(x, z) = \max_{y \in Y} K(x, z; y)$.
2. $x(y, z) = \arg\min_{x \in X} K(x, z; y)$, $x^*(z) = \arg\min_{x \in X} h(x, z)$.
3. The set $Y(z) = \arg\max_{y \in Y} d(y, z)$.
4. $y_+(z) = P_Y(y + \alpha \nabla_y K(x(y, z), z; y))$.
5. $x_+(y, z) = P_X(x - c\nabla_x K(x, z; y))$.

First of all, we can prove that $\phi^t$ is bounded from below:

**Lemma B.1** *We have*

$$
\phi(x, y, z) \geq \underline{f}.
$$

**Proof** By the definition of $d(\cdot)$ and $P(\cdot)$, we have

$$
K(x, z; y) \geq d(y, z), \quad P(z) \geq d(y, z), \quad , P(z) \geq \underline{f}. \tag{B.1}
$$

Hence, we have

$$
\begin{aligned}
\phi(x, y, z) &= P(z) + (K(x, z; y) - d(y, z)) + (P(z) - d(y, z)) \\
&\geq P(z) \\
&\geq \underline{f}.
\end{aligned}
$$

■

Next, we state some "error bounds".

**Lemma B.2** *There exist constants $\sigma_1, \sigma_2, \sigma_3$ independent of $y$ such that*

$$
\|x(y, z) - x(y, z')\| \leq \sigma_1 \|z - z'\|, \tag{B.2}
$$
$$
\|x^*(z) - x^*(z')\| \leq \sigma_1 \|z - z'\|, \tag{B.3}
$$
$$
\|x(y, z) - x(y', z)\| \leq \sigma_2 \|y - y'\|, \tag{B.4}
$$
$$
\|x^{t+1} - x(y^t, z^t)\| \leq \sigma_3 \|x^t - x^{t+1}\|, \tag{B.5}
$$

*for any $y, y' \in Y$ and $z, z' \in X$, where $\sigma_1 = \frac{p}{-L+p}$, $\sigma_2 = \frac{2(p+L)}{p-L}$, $\sigma_3 = \frac{1+(c(-L+p))}{c(-L+p)}$..*

**Proof** The proofs of (B.2), (B.3) and (B.5) are the same as those in Lemma 3.6 in [29] and hence omitted. We only need to prove (B.4). Using the strong convexity of $K(\cdot, z; y)$ of $x$, we have

$$
K(x(y, z), z; y) - K(x(y', z), z; y) \leq -\frac{-L + p}{2}\|x(y, z) - x(y', z)\|^2, \tag{B.6}
$$

$$
K(x(y, z), z; y') - K(x(y', z), z; y') \geq \frac{-L + p}{2}\|x(y, z) - x(y', z)\|^2. \tag{B.7}
$$

Moreover, using the concavity of $K(x, z; \cdot)$ of $y$, we have

$$
\begin{aligned}
&K(x(y, z), z; y') - K(x(y, z), z; y) \\
&\leq \langle \nabla_y K(x(y, z), z; y), y' - y \rangle.
\end{aligned} \tag{B.8}
$$

Using the Lipschitz-continuity of $\nabla_y K(x, z; \cdot)$, we have

$$
\begin{aligned}
&K(x(y', z), z; y) - K(x(y', z), z; y') \\
&\leq \langle \nabla_y K(x(y', z), z; y), y' - y \rangle + \frac{L}{2}\|y - y'\|^2.
\end{aligned} \tag{B.9}
$$

352 Combining (B.6) and (B.9), we have

$$(-L + p)\|x(y, z) - x(y', z)\|^2 \tag{B.10}$$

$$\leq \langle \nabla_y K(x(y, z), z; y) - \nabla_y K(x(y', z), z; y), y' - y \rangle + \frac{L}{2}\|y - y'\|^2 \tag{B.11}$$

$$\leq (p + L)\|x(y', z) - x(y, z)\|\|y - y'\| + \frac{L}{2}\|y - y'\|^2, \tag{B.12}$$

353 where the last inequality uses the Cauchy-schwarz inequality and the Lipschitz-continuity of
354 $\nabla_x K(\cdot, z; y)$ of $x$.

Let $\zeta = \|x(y, z) - x(y', z)\|/\|y - y'\|$. Then (B.10) becomes

$$\zeta^2 \leq \frac{p + L}{p - L}\zeta + \frac{L}{2(p - L)}.$$

355 Hence, we only need to solve the above quadratic inequality. We have

$$\begin{aligned}
\zeta^2 &\leq \frac{1}{2}\zeta^2 + \frac{1}{2}\left(\frac{p + L}{p - L}\right)^2 + \frac{L}{2(p - L)} \\
&\leq \frac{1}{2}\zeta^2 + \frac{1}{2}\left(\frac{p + L}{p - L}\right)^2 + \frac{p + L}{2(p - L)} \\
&\leq \frac{1}{2}\zeta^2 + \frac{1}{2}\left(\frac{p + L}{p - L}\right)^2 + \frac{1}{2}\left(\frac{p + L}{p - L}\right)^2 \\
&= \frac{1}{2}\zeta^2 + \left(\frac{p + L}{p - L}\right)^2,
\end{aligned}$$

where the first inequality is due to the AM-GM inequality and the third inequality is because $(p + L)/(p - L) > 1$. Therefore

$$\zeta \leq \sqrt{2}\frac{p + L}{p - L} < 2\frac{p + L}{p - L}.$$

356 Hence, we can take $\sigma_2 = 2(p + L)/(p - L)$ and finish the proof. ∎

357 The following lemma is a direct corollary of the above lemma:

**Lemma B.3** *The dual function* $d(\cdot, z)$ *is a differentiable function of* $y$ *with Lipschitz continuous gradient*

$$\nabla_y d(y, z) = \nabla_y K(x(y, z), z; y) = \nabla_y f(x(y, z), y)$$

358 *and*

$$\|\nabla_y d(y', z) - \nabla_y d(y'', z)\| \leq L_d \|y' - y''\|, \quad \forall y', y'' \in Y.$$

359 *with* $L_d = L + L\sigma_2$.

360 **Remark.** Note that if $p \geq 3L$, then we have $\sigma_2 = 2(p + L)/(p - L) \geq 4L$ and hence

$$L_d \geq 5L. \tag{B.13}$$

**Proof** Using Danskin's theorem in convex analysis [39], we know that $d$ is a differentiable function
with

$$\nabla_y d(y, z) = \nabla_y K(x(y, z), z; y) = \nabla_y f(x(y, z), y).$$

361 To prove the Lipschitz-continuity, we have

$$\begin{aligned}
\|\nabla_y d(y', z) - \nabla_y d(y'', z)\| &= \|\nabla_y K(x(y', z), z; y') - \nabla_y K(x(y'', z), z; y'')\| \\
&\leq \|\nabla_y K(x(y', z), z; y') - \nabla_y K(x(y', z), z; y'')\| \\
&\quad + \|\nabla_y K(x(y', z), z; y'') - \nabla_y K(x(y'', z), z; y'')\| \\
&\leq L\|y' - y''\| + L\|x(y', z) - x(y'', z)\| \\
&\leq L\|y' - y''\| + L\sigma_2\|y' - y''\| = L_d\|y' - y''\|,
\end{aligned}$$

362 where the last inequality is due to Lemma B.2. ∎

363 We then prove the following basic estimate.

364 **Proposition B.4** *We let*

$$p > 3L, c < \frac{1}{p+L}, \alpha < \min\{\frac{1}{11L}, \frac{1}{4L\sigma_3^2}\} = \min\{\frac{1}{11L}, \frac{c^2(p-L)^2}{4L(1+c(p-L))^2}\}, \beta < \min\{\frac{1}{20}, \frac{(p-L)^2}{384p(p+L)^2}\}.$$

$$(B.14)$$

365 *Then we have*

$$\phi^t - \phi^{t+1} \tag{B.15}$$

$$\geq \quad \frac{1}{8c}\|x^t - x^{t+1}\|^2$$

$$+\frac{1}{8\alpha}\|y^t - y_+^t(z^t)\|^2 + \frac{p}{8\beta}\|z^t - z^{t+1}\|^2 \tag{B.16}$$

$$-24p\beta\|x^*(z^t) - x(y_+^t(z^t), z^t)\|^2 \tag{B.17}$$

366 To prove this basic estimate, we need a series of lemmas.

367 **Lemma B.5 (Primal Descent)** *For any t, we have*

$$K(x^t, z^t; y^t) - K(x^{t+1}, z^{t+1}; y^{t+1}) \quad \geq \quad \frac{1}{2c}\|x^t - x^{t+1}\|^2 + \langle \nabla_y K(x^{t+1}, z^t; y^t), y^t - y^{t+1}\rangle$$

$$-\frac{L}{2}\|y^t - y^{t+1}\|^2 + \frac{p}{2\beta}\|z^t - z^{t+1}\|^2. \tag{B.18}$$

368 **Proof** Notice that the step of updating $x$ is a standard gradient projection, hence we have

$$K(x^t, z^t; y^t) - K(x^{t+1}, z^t; y^t) \geq \frac{1}{2c}\|x^t - x^{t+1}\|^2. \tag{B.19}$$

369 Next, because $\nabla_y K(x, z; y)$ is $L$-Lipschitz-continuous of $y$, we have

$$K(x^{t+1}, z^t; y^t) - K(x^{t+1}, z^t; y^{t+1}) \geq \langle \nabla_y K(x^{t+1}, z^t; y^t), y^t - y^{t+1}\rangle - \frac{L}{2}\|y^t - y^{t+1}\|^2. \tag{B.20}$$

370 Based on the update of variable $z^{t+1}$, i.e. $z^{t+1} = z^t + \beta(x^{t+1} - z^t)$, it is easy to show that

$$K(x^{t+1}, z^t; y^{t+1}) - K(x^{t+1}, z^{t+1}; y^{t+1}) \geq \frac{p}{2\beta}\|z^t - z^{t+1}\|^2.$$

371 Combining (B.19)-(B.21), we finish the proof. ∎

372 **Lemma B.6 (Dual Ascent)** *For any t, we have*

$$d(y^{t+1}, z^{t+1}) - d(y^t, z^t) \quad \geq \quad \langle \nabla_y d(y^t, z^t), y^{t+1} - y^t\rangle - \frac{L_d}{2}\|y^t - y^{t+1}\|^2$$

$$+\frac{p}{2}(z^{t+1} - z^t)^T(z^{t+1} + z^t - 2x(y^{t+1}, z^{t+1})) \tag{B.21}$$

$$= \quad \langle \nabla_y K(x(y^t, z^t), z^t; y^t), y^{t+1} - y^t\rangle - \frac{L_d}{2}\|y^t - y^{t+1}\|^2 \tag{B.22}$$

$$+\frac{p}{2}(z^{t+1} - z^t)^T(z^{t+1} + z^t - 2x(y^{t+1}, z^{t+1})) \tag{B.23}$$

373 **Proof** Using Lemma B.3, we have

$$-d(y^{t+1}, z^t) - (-d(y^t, z^t)) \quad \leq \quad -\langle \nabla_y d(y^t, z^t), y^{t+1} - y^t\rangle + \frac{L_d}{2}\|y^t - y^{t+1}\|^2$$

$$= \quad -\langle \nabla_y K(x^{t+1}, z^t; y^t), y^{t+1} - y^t\rangle + \frac{L_d}{2}\|y^t - y^{t+1}\|^2 \tag{B.24}$$

374 Next,

$$d(y^{t+1}, z^{t+1}) - d(y^{t+1}, z^t)$$

$$= \quad K(x(y^{t+1}, z^{t+1}), z^{t+1}; y^{t+1}) - K(x(y^{t+1}, z^t), z^t; y^{t+1})$$

$$\geq \quad K(x(y^{t+1}, z^{t+1}), z^{t+1}; y^{t+1}) - K(x(y^{t+1}, z^{t+1}), z^t; y^{t+1})$$

$$= \quad \frac{p}{2}\|x(y^{t+1}, z^{t+1}) - z^{t+1}\|^2 - \frac{p}{2}\|x(y^{t+1}, z^{t+1}) - z^t\|^2$$

$$= \quad \frac{p}{2}(z^{t+1} - z^t)^T(z^{t+1} + z^t - 2x(y^{t+1}, z^{t+1})). \tag{B.25}$$

375    Finally, using (B.24)-(B.25) we finish the proof.                                                          ∎

Recall that
$$Y(z) = \{y \in Y \mid \arg\max_{y \in Y} d(y, z)\}$$

Note that
$$P(z) = d(y(z), z),$$

376    for any $y(z) \in Y(z)$.

377    **Lemma B.7 (Proximal Descent)** *For any $t \geq 0$, there holds*
$$P(z^{t+1}) - P(z^t) \leq \frac{p}{2}(z^{t+1} - z^t)^T(z^t + z^{t+1} - 2x(y(z^{t+1}, z^t)), \qquad \text{(B.26)}$$

378    *where $y(z^{t+1}$ is arbitrary $y$ belongs to the set $Y(z^{t+1})$.*

**Proof** In the rest of the Appendix, $y(z^{t+1})$ denote any $y$ belongs to the set $Y(z^{t+1})$. Using Kaku-toni's Theorem, we have
$$\min_{x \in X} \max_{y \in Y} K(x, z; y) = \max_{y \in Y} \min_{x \in X} K(x, z; y),$$

which implies
$$\max_{y \in Y} d(y, z) = \min_{x \in X} h(x, z) = P(z).$$

379    Hence we have
$$P(z^{t+1}) - P(z^t)$$
$$\overset{(i)}{\leq} \quad P(z^{t+1}) - d(y(z^{t+1}), z^t)$$
$$\overset{(ii)}{=} \quad d(y(z^{t+1}), z^{t+1}) - d(y(z^{t+1}), z^t)$$
$$\overset{(iii)}{=} \quad K(x(y(z^{t+1}), z^t), z^{t+1}; y(z^{t+1})) - K(x(y(z^{t+1}), z^t), z^t; y(z^{t+1}))$$
$$\overset{(iv)}{=} \quad \frac{p}{2}(z^{t+1} - z^t)^T(z^{t+1} + z^t - 2x(y(z^{t+1}), z^t)),$$

380    where (i) and (iii) are because of (B.1), (ii) is because of the definition of $y(z^{t+1})$ and (iv) is from
381    direct calculation.                                                                                        ∎

**Lemma B.8** $x^*(z) = x(y, z)$ *for any $y \in Y(z)$ and $x^*(z)$ is continuous of $z$. Moreover, we have*
$$z = x^*(z)$$

382    *if and only if $z \in X^*$.*

We define
$$y_+(z) = P_Y(y + \alpha\nabla_y K(x(y, z), z; y)).$$

383    Then we have

**Lemma B.9** *We have*
$$\|y^{t+1} - y_+^t(z^t)\| \leq \kappa\|x^t - x^{t+1}\|,$$

384    *where $\kappa = \alpha L \sigma_3$.*

385    **Proof** By the nonexpansiveness of the projection operator, we have
$$\|y^{t+1} - y_+^t(z^t)\| = \|P_Y(y^t + \alpha\nabla_y K(x(y^t, z^t), z^t; y^t)) - P_Y(y^t + \alpha\nabla_y K(x^{t+1}, z^t; y^t))\|$$
$$\leq \|(y^t + \alpha\nabla_y K(x(y^t, z^t), z^t; y^t)) - (y^t + \alpha\nabla_y K(x^{t+1}, z^t; y^t))\|$$
$$\leq \alpha L\|x^{t+1} - x(y^t, z^t)\|$$
$$\leq \alpha L\sigma_3\|x^t - x^{t+1}\|,$$

386    where the first inequality is due to the nonexpansiveness of the projection operator, the second is
387    because of the Lipschitz-continuity of $\nabla_y K$ and the last inequality is because of (B.5).          ∎

388   Now we can prove Proposition B.4.

389   **Proof** Using the three descent lemma, we have

$$\Phi^t - \Phi^{t+1}$$

$$\geq \frac{1}{2c}\|x^t - x^{t+1}\|^2 + \langle \nabla_y K(x^{t+1}, z^t; y^t), y^t - y^{t+1}\rangle - \frac{L}{2}\|y^t - y^{t+1}\|^2 + \frac{p}{2\beta}\|z^t - z^{t+1}\|^2$$

$$+ 2\langle \nabla_y K(x(y^t, z^t), z^t; y^t), y^{t+1} - y^t\rangle - \frac{2L_d}{2}\|y^t - y^{t+1}\|^2 + p(z^{t+1} - z^t)^T(z^{t+1} + z^t - 2x(y^{t+1}, z^{t+1}))$$

$$- p(z^{t+1} - z^t)^T(z^t + z^{t+1} - 2x(y(z^{t+1}), z^t))$$

$$= \frac{1}{2c}\|x^t - x^{t+1}\|^2 - \frac{L + 2L_d}{2}\|y^t - y^{t+1}\|^2 + \frac{p}{2\beta}\|z^t - z^{t+1}\|^2 + \langle \nabla_y K(x^{t+1}, z^t; y^t), y^{t+1} - y^t\rangle$$

$$+ 2\langle \nabla_y K(x(y^t, z^t), z^t; y^t) - \nabla_y K(x^{t+1}, z^t; y^t), y^{t+1} - y^t\rangle \tag{B.27}$$

$$+ 2p(z^{t+1} - z^t)^T(x(y(z^{t+1}), z^t) - x(y^{t+1}, z^{t+1})). \tag{B.28}$$

390   Using the property of the projection operator and the update of the dual variable $y^t$, we have

$$\langle \nabla_y K(x^{t+1}, z^t; y^t), y^{t+1} - y^t\rangle \geq \frac{1}{\alpha}\|y^t - y^{t+1}\|^2.$$

391   Substituting the above inequality into (B.27), we get

$$\Phi^t - \Phi^{t+1}$$

$$\geq \frac{1}{2c}\|x^t - x^{t+1}\|^2 + (\frac{1}{\alpha} - \frac{L + 2L_d}{2})\|y^t - y^{t+1}\|^2 + \frac{p}{2\beta}\|z^t - z^{t+1}\|^2$$

$$+ 2\langle \nabla_y K(x(y^t, z^t), z^t; y^t) - \nabla_y K(x^{t+1}, z^t; y^t), y^{t+1} - y^t\rangle$$

$$+ 2p(z^{t+1} - z^t)^T(x(y(z^{t+1}), z^t) - x(y^{t+1}, z^{t+1}))$$

$$\geq \frac{1}{2c}\|x^t - x^{t+1}\|^2 + (\frac{1}{\alpha} - \frac{L + 10L}{2})\|y^t - y^{t+1}\|^2 + \frac{p}{2\beta}\|z^t - z^{t+1}\|^2$$

$$+ 2\langle \nabla_y K(x(y^t, z^t), z^t; y^t) - \nabla_y K(x^{t+1}, z^t; y^t), y^{t+1} - y^t\rangle$$

$$+ 2p(z^{t+1} - z^t)^T(x(y(z^{t+1}), z^t) - x(y^{t+1}, z^{t+1}))$$

$$\geq \frac{1}{2c}\|x^t - x^{t+1}\|^2 + \frac{1}{2\alpha}\|y^t - y^{t+1}\|^2 + \frac{p}{2\beta}\|z^t - z^{t+1}\|^2$$

$$+ 2\langle \nabla_y K(x(y^t, z^t), z^t; y^t) - \nabla_y K(x^{t+1}, z^t; y^t), y^{t+1} - y^t\rangle$$

$$+ 2p(z^{t+1} - z^t)^T(x(y(z^{t+1}), z^t) - x(y^{t+1}, z^{t+1}))$$

392   where the second inequality is because of (B.13) and the last inequality is because $\alpha \leq \frac{1}{11L}$.

393   Notice that

$$2p(z^{t+1} - z^t)^T(x(y(z^{t+1}), z^t) - x(y^{t+1}, z^{t+1}))$$

$$= 2p(z^{t+1} - z^t)^T((x(y(z^{t+1}), z^t) - x(y(z^{t+1}), z^{t+1})) + (x(y(z^{t+1})), z^{t+1}) - x(y^{t+1}, z^{t+1})))$$

$$= 2p(z^{t+1} - z^t)^T(x(y(z^{t+1}), z^t) - x(y(z^{t+1}), z^{t+1}))$$

$$+ 2p(z^{t+1} - z^t)^T(x(y(z^{t+1})), z^{t+1}) - x(y^{t+1}, z^{t+1}))$$

$$\overset{(i)}{\geq} -2p\sigma_1\|z^{t+1} - z^t\|^2 + 2p(z^{t+1} - z^t)^T(x(y(z^{t+1})), z^{t+1}) - x(y^{t+1}, z^{t+1}))$$

$$\overset{(ii)}{\geq} -2p\sigma_1\|z^{t+1} - z^t\|^2 - \frac{p}{6\beta}\|z^{t+1} - z^t\|^2 - 6p\beta\|x(y(z^{t+1}), z^{t+1}) - x(y^{t+1}, z^{t+1})\|^2,$$

394   where (i) is because of the Cauchy-Schwarz inequality and Lemma B.2 and (ii) is due to the AM-GM
395   inequality. Also we have

$$2\langle \nabla_y K(x(y^t, z^t), z^t; y^t) - \nabla_y K(x^{t+1}, z^t; y^t), y^{t+1} - y^t\rangle$$

$$\geq -2\|\nabla_y K(x(y^t, z^t), z^t; y^t) - \nabla_y K(x^{t+1}, z^t; y^t)\| \cdot \|y^{t+1} - y^t\|$$

$$\geq -2L\|x^{t+1} - x(y^t, z^t)\| \cdot \|y^t - y^{t+1}\|$$

$$\geq -L\sigma_3^2\|y^t - y^{t+1}\|^2 - L\sigma_3^{-2}\|x^{t+1} - x(y^t, z^t)\|^2$$

$$\geq -L\sigma_3^2\|y^t - y^{t+1}\|^2 - L\|x^{t+1} - x^t\|^2,$$

where the first inequality is because of the Cauchy-Schwarz in equality, the second inequality is because $\nabla_y K = \nabla_y f$ is $L$-Lipschitz-continuous, the third inequality is due to the AM-GM inequality and the last is because of (B.5).

Hence we have

$$\Phi^t - \Phi^{t+1}$$
$$\geq \quad (\frac{1}{2c} - L)\|x^t - x^{t+1}\|^2 + (\frac{1}{2\alpha} - L\sigma_3^2)\|y^t - y^{t+1}\|^2 + (\frac{p}{2\beta} - 2p\sigma_1 - \frac{p}{6\beta})\|z^t - z^{t+1}\|^2$$
$$-6p\beta\|x(y(z^{t+1}), z^{t+1}) - x(y^{t+1}, z^{t+1})\|^2$$

By the conditions of $p, c$, we have

$$1/2c - L \geq 1/4c.$$

By the condition for $\alpha$, we have

$$\alpha < 1/(4L\sigma_3^2),$$

which yields

$$1/(2\alpha) - L\sigma_3^2 \geq 1/(4\alpha)$$

And by the conditions that $\beta < \frac{1}{20}$ and $p \geq 3L$ together with the definition of $\sigma_1$,

$$\frac{p}{2\beta} - 2p\sigma_1 - \frac{p}{6\beta} \geq \frac{p}{4\beta}.$$

Then we have

$$\Phi^t - \Phi^{t+1}$$
$$\geq \quad \frac{1}{4c}\|x^t - x^{t+1}\|^2 + \frac{1}{4\alpha}\|y^t - y^{t+1}\|^2 + \frac{p}{4\beta}\|z^t - z^{t+1}\|^2$$
$$-6p\beta\|x(y(z^{t+1}), z^{t+1}) - x(y^{t+1}, z^{t+1})\|^2 \tag{B.29}$$
$$= \quad \frac{1}{4c}\|x^t - x^{t+1}\|^2 + \frac{1}{4\alpha}\|y^t - y^{t+1}\|^2 + \frac{p}{4\beta}\|z^t - z^{t+1}\|^2$$
$$-6p\beta\|x^*(z^{t+1}) - x(y^{t+1}, z^{t+1})\|^2, \tag{B.30}$$

where the last equality is because of Lemma B.8. By Lemma B.9 and the convexity of the norm square function, we have

$$\|y^{t+1} - y^t\|^2 \quad = \quad \|(y^{t+1} - y_+^t(z^t)) + (y_+^t(z^t) - y^t)\|^2 \tag{B.31}$$
$$\geq \quad \|y^t - y_+^t(z^t)\|^2/2 - \|y^{t+1} - y_+^t(z^t)\|^2 \tag{B.32}$$
$$\geq \quad \|y^t - y_+^t(z^t)\|^2/2 - \kappa^2\|x^t - x^{t+1}\|^2. \tag{B.33}$$

Similarly, by Lemma B.9, (B.4) and the convexity of norm square function, we have

$$\|x^*(z^{t+1}) - x(y^{t+1}, z^{t+1})\|^2 \tag{B.34}$$
$$= \quad \|(x^*(z^{t+1}) - x^*(z^t)) + (x^*(z^t) - x(y_+^t(z^t), z^t)) \tag{B.35}$$
$$+(x(y_+^t(z^t), z^t) - x(y^{t+1}, z^t)) + (x(y^{t+1}, z^t) - x(y^{t+1}, z^{t+1}))\|^2 \tag{B.36}$$
$$\leq \quad 4\|x^*(z^{t+1}) - x^*(z^t)\|^2 + 4\|x^*(z^t) - x(y_+^t(z^t), z^t)\|^2 \tag{B.37}$$
$$+4\|x(y_+^t(z^t), z^t) - x(y^{t+1}, z^t)\|^2 + 4\|x(y^{t+1}, z^t) - x(y^{t+1}, z^{t+1})\|^2 \tag{B.38}$$
$$\leq \quad 4\sigma_1^2\|z^t - z^{t+1}\|^2 + 4\|x^*(z^t) - x(y_+^t(z^t), z^t)\|^2 \tag{B.39}$$
$$+4\sigma_2^2\kappa^2\|x^t - x^{t+1}\|^2 + 4\sigma_1^2\|z^t - z^{t+1}\|^2 \tag{B.40}$$
$$= \quad 8\sigma_1^2\|z^t - z^{t+1}\|^2 + 4\|x^*(z^t) - x(y_+^t(z^t), z^t)\|^2 \tag{B.41}$$
$$+4\sigma_2^2\kappa^2\|x^t - x^{t+1}\|^2. \tag{B.42}$$

Substituting (B.31) and (B.34) to (B.29) yields

$$\phi^t - \phi^{t+1}$$
$$\geq \quad (\frac{1}{4c} - 24p\beta\sigma_2^2\kappa^2 - \kappa^2/(4\alpha))\|x^t - x^{t+1}\|^2$$
$$+\frac{1}{8\alpha}\|y^t - y_+^t(z^t)\|^2 + (\frac{p}{4\beta} - 48p\beta\sigma_1^2)\|z^t - z^{t+1}\|^2$$
$$-24p\beta\|x^*(z^t) - x(y_+^t(z^t), z^t)\|^2.$$

Notice that
$$\alpha < 1/(4L\sigma_3^2) < 1/(4cL^2\sigma_3^2),$$
and hence $\kappa^2/(4\alpha) = \alpha^2 L^2 \sigma_3^2/(4\alpha) < 1/(16c)$. Also we have
$$\beta < 1/(96p\alpha\sigma_2^2),$$
thus
$$24p\beta\sigma_2^2\kappa^2 < \kappa^2/(4\alpha) \le 1/(16c).$$
Consequently, we have
$$(\frac{1}{4c} - 24p\beta\sigma_2^2\kappa^2 - \kappa^2/(4\alpha)) < 1/(8c).$$
By the definition of $\sigma_1$ and the conditions $p \ge 3L, \beta < \frac{1}{20}$, we have
$$(\frac{p}{4\beta} - 48p\beta\sigma_1^2) \ge \frac{p}{8\beta}.$$

405 Combining the above, we have
$$\begin{aligned}
\phi^t &- \phi^{t+1} \\
\ge \quad &\frac{1}{8c}\|x^t - x^{t+1}\|^2 \\
&+ \frac{1}{8\alpha}\|y^t - y_+^t(z^t)\|^2 + \frac{p}{8\beta}\|z^t - z^{t+1}\|^2 \\
&- 24p\beta\|x^*(z^t) - x(y_+^t(z^t), z^t)\|^2,
\end{aligned}$$

406 which finishes the proof. ∎

## B.2 General nonconvex-concave case

408 We have the following error bound:

409 **Lemma B.10** *We have*
$$\begin{aligned}
(p - L)\|x^*(z^t) - x(y_+^t(z^t), z^t)\|^2 \quad &< \quad (1 + \alpha L)\|y^t - y_+^t(z^t)\| \cdot \text{dist}(y_+^t(z^t), Y(z^t)). \\
&\le \quad (1 + \alpha L)\|y^t - y_+^t(z^t)\| \cdot DY),
\end{aligned}$$

410 *where $D(Y)$ is the diameter of $Y$.*

411 The proof will be given in the next section.

412 **Lemma B.11** *If*
$$\max\{\|x^t - x^{t+1}\|, \|y^t - y_+^t(z^t)\|, \|z^t - x^{t+1}\|\} \le \bar{\lambda}\epsilon, \tag{B.43}$$

413 *then $(x^{t+1}, y^{t+1})$ is a $\lambda\epsilon-$ solution for some $\lambda > 0$.*

414 **Proof**

By the update of $x^{t+1}$, we have
$$x^{t+1} = \arg\min_{x \in X}\{\langle \nabla_x f(x^t, y^t) + p(x^t - z^t), x - x^t\rangle + \frac{1}{c}\|x - x^t\|^2 + \iota(x)\}.$$

415 Therefore, we have
$$0 \in \nabla_x f(x^{t+1}, y^t) + p(x^{t+1} - z^t) + \frac{1}{c}(x^{t+1} - x^t) + \iota(x^{t+1}). \tag{B.44}$$

416 Similarly, we have
$$0 \in \arg\min_{y \in Y}\{-\nabla_y f(x^{t+1}, y^t) + \frac{1}{\alpha}(y^{t+1} - y^t) + \iota(y^{t+1})\}. \tag{B.45}$$

We let
$$u = (\nabla_x f(x^{t+1}, y^t) - \nabla_x f(x^t, y^t)) + (\nabla_x f(x^{t+1}, y^{t+1}) - \nabla_x f(x^{t+1}, y^t)) - p(x^{t+1} - z^t) - \frac{1}{c}(x^{t+1} - x^t)$$

and
$$v = \nabla_y f(x^{t+1}, y^t) - \nabla_y f(x^{t+1}, y^{t+1}) - \frac{1}{\alpha}(y^{t+1} - y^t).$$

By the Lipschitz-continuity of $\nabla_x f(x, y)$, Lemma B.9 and (B.43), we have

$$
\begin{aligned}
\|u\| &\leq L\|x^t - x^{t+1}\| + L\|y^t - y^{t+1}\| + p\epsilon + \frac{1}{\mathbb{C}}\epsilon \\
&\leq (1 + p + 1/c)\epsilon + L\|y^t - y_+^t(z^t)\| + \|y_+^t(z^t) - y^{t+1}\| \\
&\leq (1 + p + 1/c)\epsilon + \epsilon + \kappa\epsilon,
\end{aligned}
$$

where the first and the second inequalities are both due to (B.43), the triangular inequality and the Lipschitz-continuity of $\nabla_x f(\cdot)$ and the last inequality is because of Lemma B.9. Similarly, we can prove that

$$\|v\| \leq (L + 1 + \kappa + \frac{1}{\alpha})\epsilon.$$

Hence, we finish the proof with $\eta = 2 + L + p + \kappa + \max\{1/c, 1/\alpha\}$.

$\blacksquare$

We say that $\phi^t$ decreases sufficiently if

$$\phi^t - \phi^{t+1} \geq \frac{1}{16c}\|x^t - x^{t+1}\|^2 + \frac{1}{16\alpha}\|y^t - y_+^t(z^t)\|^2 + \frac{p\beta}{16}\|z^t - x^{t+1}\|^2. \tag{B.46}$$

**Lemma B.12** *Let $T > 0$. Then if for any $t \in \{0, 1, \cdots, T-1\}$, (B.46) holds, there must exist a $t \in \{1, 2, \cdots, T\}$ such that $(x^t, y^t)$ is an $C/\sqrt{T\beta}$-solution. Moreover, if for any $t \geq 0$, (B.46) holds, Then any limit point of $(x^t, y^t)$ is a solution of (1.2), and the iteration complexity of attaining an $\epsilon-$solution is $\mathcal{O}(1/\epsilon^2)$.*

**Proof**

We have

$$
\begin{aligned}
\phi^0 - \underline{f} &\geq \sum_{t=0}^{T-1}(\phi^t - \phi^{t+1}) && \text{(B.47)} \\
&\geq (1/(16c) + 1/(16\alpha) + p/16)\sum_{t=0}^{T-1}\max\{\|x^t - x^{t+1}\|^2, \|y^t - y_+^t(z^t)\|^2, \beta\|x^{t+1} - z^t\|^2\} && \text{(B.48)}
\end{aligned}
$$

where the last inequality is due to (B.46). Therefore, there exists a $t \in \{0, 1, \cdots, T-1\}$ such that

$$(1/(16c) + 1/(16\alpha) + p/16)\max\{\|x^t - x^{t+1}\|^2, \|y^t - y_+^t(z^t)\|^2, \beta\|x^{t+1} - z^t\|^2\} \leq (\phi^0 - \underline{f})/T.$$

Since $\beta < 1$, we further get

$$(1/(16c) + 1/(16\alpha) + p/16)\max\{\|x^t - x^{t+1}\|^2, \|y^t - y_+^t(z^t)\|^2, \|x^{t+1} - z^t\|^2\} \leq (\phi^0 - \underline{f})/(T\beta).$$

Hence, by Lemma B.11, $(x^{t+1}, y^{t+1})$ is a $\sqrt{(\phi^0 - \underline{f})/((1/8c + 1/8\alpha + 16)T\beta)}$-solution. According to above analysis, If (B.46) holds for any $t$, we can attain an $\epsilon$-solution within $(\phi^0 - \underline{f})/(\beta(1/(16c) + 1/(16\alpha) + p/16)\epsilon^2)$ iterations. Moreover, if (B.46) holds for any $t$, by (B.47), we have

$$\max\{\|x^t - x^{t+1}\|, \|y^t - y_+^t(z^t)\|, \|z^t - x^{t+1}\|\} \rightarrow 0. \tag{B.49}$$

Consequently, for any limit point $(\bar{x}, \bar{y})$ of $(x^t, y^t)$, there exists a $\bar{z}$ such that

$$\max\{\|\bar{x} - \bar{x}_+(\bar{y}, \bar{z})\|, \|\bar{y} - \bar{y}_+(\bar{z})\|, \|\bar{x}_+(\bar{y}, \bar{z}) - \bar{z}\|\} = 0,$$

which yields $(\bar{x}, \bar{y})$ is a stationary solution. Here

$$x_+(y, z) = P_X(x - \nabla_x K(x, z; y)).$$

$\blacksquare$

Now we are ready to prove Theorem 3.3.

**Proof** [Proof of Theorem 3.3] There are two cases (B.50) and (B.51) as discussed in the proof for the general nonconvex-concave problems in last subsection.

1. For some $t \in \{0, 1, \cdots, T-1\}$, we have

$$\frac{1}{2}\max\{\frac{1}{8c}\|x^t - x^{t+1}\|^2, \frac{1}{8\alpha}\|y^t - y_+^t(z^t)\|^2, \frac{p}{8\beta}\|z^t - z^{t+1}\|^2\} \leq 24p\beta\|x^*(z^t) - x(y_+^t(z^t), z^t)\|^2.$$
(B.50)

2. For any $t \in \{0, 1, \cdots, T-1\}$,

$$\frac{1}{2}\max\{\frac{1}{8c}\|x^t - x^{t+1}\|^2, \frac{1}{8\alpha}\|y^t - y_+^t(z^t)\|^2, \frac{p}{8\beta}\|z^t - z^{t+1}\|^2\} \geq 24p\beta\|x^*(z^t) - x(y_+^t(z^t), z^t)\|^2$$
(B.51)

In the first case (B.50), we have

$$\begin{aligned}
\|y^t - y_+^t(z^t)\|^2 &\leq 384p\beta\alpha\|x(y_+^t(z^t), z^t) - x^*(z^t)\|^2 \\
&\leq 384p\beta\alpha\frac{(1+\alpha L)}{p-L}\|y^t - y_+^t(z^t)\|D(Y).
\end{aligned}$$

Hence, letting $\lambda_1 = 384p\alpha\frac{(1+\alpha L)}{p-L} \cdot D(Y)$, we have

$$\|y^t - y_+^t(z^t)\| \leq \lambda_1\beta.$$
(B.52)

Moreover,

$$\begin{aligned}
\|x^{t+1} - z^t\|^2 &= \|(z^{t+1} - z^t)/\beta\|^2 & \text{(B.53)} \\
&\overset{\text{(i)}}{\leq} 384p\|x(y_+^t(z^t), z^t) - x^*(z^t)\|^2 & \text{(B.54)} \\
&\overset{\text{(ii)}}{\leq} 384p\frac{1+\alpha L}{p-L}D(Y)\|y^t - y_+^t(z^t)\| & \text{(B.55)} \\
&\overset{\text{(iii)}}{\leq} 384p\frac{1+\alpha L}{p-L}D(Y)\lambda_1\beta, & \text{(B.56)}
\end{aligned}$$

where Inequality (i) is due to Inequality (B.50) and (ii) is because of Lemma B.10 and (iii) is due to (B.52). We also have

$$\begin{aligned}
\|x^t - x^{t+1}\|^2 &\overset{\text{(i)}}{\leq} 384cp\beta\|x^*(z^t) - x(y_+^t(z^t), z^t)\|^2 & \text{(B.57)} \\
&\overset{\text{(ii)}}{\leq} 384pc\beta\frac{1+\alpha L}{p-L}D(Y)\|y^t - y_+^t(z^t)\| & \text{(B.58)} \\
&\overset{\text{(iii)}}{\leq} 384pc\frac{1+\alpha L}{p-L}\lambda_1 D(Y)\beta^2, & \text{(B.59)}
\end{aligned}$$

where (i) is due to (B.50), (ii) is due to Lemma B.10 and (iii) is because of (B.52). Combining the above, in the first case, we have

$$\max\{\|x^t - x^{t+1}\|^2, \|y^t - y_+^t(z^t)\|^2, \|z^t - x^{t+1}\|^2\}$$
(B.60)
$$\leq \max\{\lambda_2\beta^2, \lambda_1^2\beta^2, \lambda_3\beta\},$$
(B.61)

where $\lambda_2 = 384p\frac{1+\alpha L}{p-L}D(Y)\lambda_1$ and $\lambda_3 = 192pc\frac{1+\alpha L}{p-L}\lambda_1 D(Y)$ According to Lemma B.11, there exists a $\lambda > 0$ such that $(x^{t+1}, y^{t+1})$ is a $\lambda$ $max\{\beta, \sqrt{\beta}\}$-solution.

In the second case, we have

$$\phi^t - \phi^{t+1} \geq \frac{1}{16c}\|x^t - x^{t+1}\|^2 + \frac{1}{16\alpha}\|y^t - y_+^t(z^t)\|^2 + \frac{1}{16\beta}\|z^t - z^{t+1}\|^2$$

for any $t \in \{0, 1, \cdots, T-1\}$. By Lemma B.12, there exists a $t \in \{0, 1, \cdots, T-1\}$, such that $(x^{t+1}, y^{t+1})$ is a $\sqrt{(\phi^0 - \underline{f})/((1/8c + 1/8\alpha + 16)T\beta)}$-solution Finally taking $\beta = 1/\sqrt{T}$ and combining the two cases with Lemma B.12 yield the desired results.

∎

## B.3 The max problem is over a discrete set

In this subsection, we prove Theorem 3.6. We will prove that under the strict complementarity assumption, the potential function $\phi^t$ decreases sufficiently after any iteration. Then by the following simple lemma, we can prove Theorem 3.6.

By the bounded level set assumption (Assumption 3.5) and the fact that $\psi(z) \leq P(z)$, for any $(x^0, y^0, z^0) \in \mathbb{R}^{n+m+n}$, there exists a constant $R(x^0, y^0, z^0) > 0$ such that

$$\{z \mid P(z) \leq \phi(x^0, y^0, z^0)\} \subseteq \mathcal{B}(R(x^0, y^0, z^0)).$$

Then we have the following "dual error bound". Note that this error bound is homogeneous compared to Lemma B.10.

**Lemma B.13** *Let*
$$x_+(y, z) = P_X(x - \nabla_x K(x, z; y)).$$
*If the strict complementarity assumption and the bounded level set assumption hold for* (1.2) *, there exists $\delta > 0$, such that if*
$$\|z\| \leq R(x^0, y^0, z^0),$$
*and*
$$\max\{\|x - x_+(y, z)\|, \|y - y_+(z)\|, \|x_+(y, z) - z\|\} < \delta$$
*we have*
$$\|x(y_+(z), z) - x^*(z)\| < \sigma_5 \|y - y_+(z)\|$$
*for some constant $\sigma_5 > 0$.*

Equipped with the dual error bound, we can prove the that the potential function decreases after any iteration in the following proposition:

**Proposition B.14** *Suppose the conditions in Theorem 3.6 holds, we have*

$$\phi^t - \phi^{t+1} \geq \frac{1}{16c}\|x^t - x^{t+1}\|^2 + \frac{1}{16\alpha}\|y^t - y_+^t(z^t)\|^2 + \frac{p}{16\beta}\|z^t - z^{t+1}\|^2. \tag{B.62}$$

**Proof** We set $\beta < \min\{\delta/\sqrt{\lambda_2}, \delta/\lambda_1, \delta^2/\lambda_3, 1/(384p\alpha\sigma_5^2)\}$. First, we prove that

$$\phi^t - \phi^{t+1} \geq \frac{1}{16c}\|x^t - x^{t+1}\|^2 + \frac{1}{16\alpha}\|y^t - y_+^t(z^t)\|^2 + \frac{p}{16\beta}\|z^t - z^{t+1}\|^2. \tag{B.63}$$

and

$$\|z^t\| < R(x^0, y^0, z^0)$$

for any $t \geq 0$. We prove it by induction. We will prove that

    1. If $\|z^t\| \leq R(x^0, y^0, z^0)$, then

$$\phi^t - \phi^{t+1} \geq \frac{1}{16c}\|x^t - x^{t+1}\|^2 + \frac{1}{16\alpha}\|y^t - y_+^t(z^t)\|^2 + \frac{p}{16\beta}\|z^t - z^{t+1}\|^2. \tag{B.64}$$

    2. If $\phi^{t+1} \leq \phi^t$, we have $\|z^{t+1}\| \leq R(x^0, y^0, z^0)$.

For $t = 0$, it is trivial that $\|z^t\| \leq R(x^0, y^0, z^0)$. For the first step, assume that we have $\|z^t\| \leq R(x^0, y^0, z^0)$. There are two cases:

    1. For some $t$, we have

$$\frac{1}{2}\max\{\frac{1}{8c}\|x^t - x^{t+1}\|^2, \frac{1}{8\alpha}\|y - y_+^t(z^t)\|^2, \frac{p}{8\beta}\|z^t - z^{t+1}\|^2\} \leq 24p\beta\|x^*(z^t) - x(y_+^t(z^t), z^t)\|^2.$$
$$\tag{B.65}$$

    2. For any $t$,

$$\frac{1}{2}\max\{\frac{1}{8c}\|x^t - x^{t+1}\|^2, \frac{1}{8\alpha}\|y - y_+^t(z^t)\|^2, \frac{p}{8\beta}\|z^t - z^{t+1}\|^2\} \geq 24p\beta\|x^*(z^t) - x(y_+^t(z^t), z^t)\|^2$$
$$\tag{B.66}$$

For the first case, as in the last subsection, we have

$$\max\{\|x^t - x^{t+1}\|^2, \|y^t - y^t_+(z^t)\|^2, \|x^{t+1} - z^t\|^2\}$$
$$\leq \max\{\lambda_2 \beta^2, \lambda_1^2 \beta^2, \lambda_3 \beta\}$$
$$\leq \delta^2.$$

Hence, we can make use of Lemma B.13. In fact, we have

$$24p\beta\|x(y^t_+(z^t), z^t) - x^*(z^t)\|^2 \leq 24p\beta\sigma_5^2\|y^t - y^t_+(z^t)\|$$
$$\leq \frac{1}{16\alpha}\|y^t - y^t_+(z^t)\|^2,$$

which yields (B.46) together with (B.15). For the second case, (B.46) holds as in the last subsection. Hence, if $\|z^t\| \leq R(x^0, y^0, z^0)$, we have (B.46). For the second step, if (B.46) holds for $0, 1, \cdots, (t-1)$, we have

$$P(z^{t+1}) \leq \phi^{t+1}$$
$$\leq \phi^0.$$

Hence, $z^{t+1} \in \mathcal{B}(R(x^0, y^0, z^0))$. Combining these, for any $t \geq 0$, $\|z^t\| \leq R(x^0, y^0, z^0)$ and (B.46) holds. Then the theorem comes from Lemma B.12. ∎

# C   The multi-block cases

The proofs for the multi-block case is similar to the one-block case. In this section, we briefly introduce the proof of them. Note that the only differences for proving the theorem s are Lemma B.5, (B.5) and Proposition 4.1. Instead, we have the following:

**Lemma C.1 (Primal Descent)** *For any t, we have*

$$K(x^t, z^t; y^t) - K(x^{t+1}, z^{t+1}; y^{t+1}) \geq \frac{1}{2c}\|x^t - x^{t+1}\|^2 + \langle \nabla_y K(x^{t+1}, z^t; y^t), y^t - y^{t+1}\rangle$$
$$-\frac{L}{2}\|y^t - y^{t+1}\|^2 + \frac{p}{2\beta}\|z^t - z^{t+1}\|^2. \tag{C.1}$$

The proof of it is the same as Lemma 5.3 in [29]. The error bound (B.5) becomes:

**Lemma C.2** *We have*
$$\|x^{t+1} - x(y^t, z^t)\| \leq \sigma_3'\|x^t - x^{t+1}\|,$$
*where $\sigma_3' = (c(p - L) + 1 + c(L + p)N^{3/2})/c(p - L)$.*

The proof of Lemma C.2 is similar to Lemma 5.2 in [29] hence omitted here. Because of the above two differences, we have a replacement of Proposition B.4:

**Proposition C.3** *We let*

$$p > 3L, c < \frac{1}{p+L}, \alpha < \min\{\frac{1}{11L}, \frac{c^2(p-L)^2}{4L(1 + c(p+L)N^{3/2} + c(p-L))^2}\}, \min\{\frac{1}{20}, \beta < \frac{(p-L)^2}{384p(p+L)^2}\}.$$
$$\tag{C.2}$$

*Then we have*

$$\phi^t - \phi^{t+1} \tag{C.3}$$
$$\geq \frac{1}{4c}\|x^t - x^{t+1}\|^2$$
$$+\frac{1}{4\alpha}\|y^t - y^t_+(z^t)\|^2 + \frac{p}{8\beta}\|z^t - z^{t+1}\|^2 \tag{C.4}$$
$$-24p\beta\|x^*(z^t) - x(y^t_+(z^t), z^t)\|^2 \tag{C.5}$$

The proof of Proposition C.3 is similar to Proposition B.4 hence omitted.

 # D    Proof of the error bound lemmas

 ## D.1    Proof of lemma B.10

Let

$$x_+(y, z) = P_X(x - c\nabla_x K(x, z; y))$$

and

$$y_+(z) = P_Y(y + \alpha\nabla_y K(x(y, z), z; y)).$$

 Then Lemma B.10 can be written as

 **Lemma D.1** *We have*

$$
\begin{aligned}
(p - L)\|x^*(z) - x(y_+(z), z)\|^2 \quad &< \quad (1 + \alpha L)\|y - y_+(z)\| \cdot \mathrm{dist}(y_+(z), Y(z)). \\
&\leq \quad (1 + \alpha L)\|y - y_+(z)\| \cdot D(Y),
\end{aligned}
$$

 *where $D(Y)$ is the diameter of $Y$.*

 **Proof**  By the strong convexity of $K(\cdot, z; y)$, we have

$$K(x^*(z), z, ; y_+(z)) - K(x(y_+(z), z)z; y_+(z)) \geq \frac{p - L}{2}\|x(y_+(z), z) - x^*(z)\|^2 \quad \text{(D.1)}$$

$$K(x(y_+(z), z), z; y(z)) - K(x^*(z), z; y(z)) \geq \frac{p - L}{2}\|x(y_+(z), z) - x^*(z)\|^2, \quad \text{(D.2)}$$

where $y(z)$ is an arbitrary vector in $Y(z)$. Notice that $y_+(z)$ is the maximizer of the following problem:

$$\max_{\bar{y} \in Y}\{K(x(y_+(z), z), z; \bar{y}) - \delta^T(y, y_+(z); z)\bar{y}\},$$

where

$$\delta(y, y_+(z); z) = (y_+(z) + \alpha\nabla_{\bar{y}} K(x(y_+(z), z), z; y_+(z))) - (y + \alpha_{\bar{y}} K(x(y_+(z), z), z; y))$$

satisfies

$$\|\delta(y, y_+(z); z)\| < (1 + \alpha L)\|y - y_+(z)\|,$$

 by the Lipschitz-continuity of $\nabla_y K = \nabla_y f$. Hence, we have

$$
\begin{aligned}
&K(x(y_+(z), z), z; y(z)) - \delta^T(y, y_+(z); z)y(z) \\
\leq \quad &K(x(y_+(z), z), z; y_+(z)) - \delta^T(y, y_+(z); z)y_+(z).
\end{aligned}
$$

 Then, we have the following estimates:

$$K(x(y_+(z), z), z; y(z)) - K(x(y_+(z), z), z; y_+(z)) \quad \text{(D.3)}$$

$$\leq \quad (y(z) - y_+(z))^T \delta(y, y_+(z); z) \quad \text{(D.4)}$$

$$\leq \quad \|y_+(z) - y(z)\| \cdot (1 + \alpha L)\|y - y_+(z)\|. \quad \text{(D.5)}$$

Also because $y(z)$ maximizes

$$\max_{\bar{y} \in Y} K(x^*(z), \bar{y}; z),$$

 we have

$$K(x^*(z), z; y(z)) \geq K(x^*(z), z; y_+(z)). \quad \text{(D.6)}$$

Since $y(z)$ is an arbitrary vector in $Y(z)$, combining (D.1), (D.3), (D.6), we have

$$(p - L)\|x^*(z) - x(y_+(z), z)\|^2 < (1 + \alpha L)\|y - y_+(z)\| \cdot \mathrm{dist}(y_+(z), Y(z)),$$

 which is the desired result.  ∎

## D.2 Proof of Lemma B.13

For a pair of min-max solution of (1.2), the KKT conditions in the following hold:

$$J^T F(x^*)y = 0, \tag{D.7}$$

$$\sum_{i=1}^{m} y_i = 1, \tag{D.8}$$

$$y_i \geq 0, \forall i \in [m] \tag{D.9}$$

$$\mu - \nu_i = f_i(x), \forall i \in [m], \tag{D.10}$$

$$\nu_i \geq 0, \nu_i y_i = 0, \forall i \in [m], \tag{D.11}$$

where $\mu$ is the multiplier of the equality constraint $\sum_{i=1}^{m} y_i = 1$ and $\nu_i$ is the multiplier for the inequality constraint $y_i \geq 0$.

**Definition D.2** *For $y \in Y$, we define the active set*

$$\mathcal{A}[y] = \{i \in [m] \mid y_i = 0\}.$$

*We also define the inactive set of $y$ as follows:*

$$\mathcal{I}[y] = \{i \in [m] \mid y_i > 0\}.$$

**Definition D.3** *For an $x \in \mathbb{R}^n$, we define the top coordinate set $\mathcal{T}(x)$ as the collection of all indexes of the top coordinates of $F(x)$, i.e., $f_i(x) > f_j(x)$ if $i \in \mathcal{T}(x), j \notin \mathcal{T}(x)$ and $f_i(x) = f_j(x)$ if $i, j \in \mathcal{T}(x)$.*

According to the KKT conditions, it is easy to see that for $(x, y) \in W^*$,

$$\mathcal{I}[y] \subseteq \mathcal{T}(x).$$

Recall that we have the following strict complementarity condition:

**Assumption D.4** *For any $(x, y)$ satisfying (D.7), we have*

$$\nu_i > 0, \forall i \in \mathcal{A}[y].$$

It is easy to see that if the strict complementarity assumption holds,

$$\mathcal{I}[y] = \mathcal{T}(x)$$

for $(x, y) \in W^*$. Then we can prove the following "dual error bound".

**Lemma D.5** *If the strict complementarity assumption holds for (1.2), there exists $\delta > 0$, such that if*

$$\|z\| \leq R(x^0, y^0, z^0),$$

*and*

$$\max\{\|x - x_+(y, z)\|, \|y - y_+(z)\|, \|x_+(y, z) - z\|\} < \delta$$

*we have*

$$\|x(y_+(z), z) - x^*(z)\| < \sigma_5 \|y - y_+(z)\|$$

*for some constant $\sigma_5 > 0$.*

To prove this, we need the following lemmas. First, we prove that if the residuals go to zero, the iteration points converge to a solution.

**Lemma D.6** *If $\{z^k\}$ is a sequence with $\|z^k\| \leq R(x^0, y^0, z^0)$ and*

$$\max\{\|x^k - x_+^k(y^k, z^k)\|, \|y^k - y_+^k(z^k)\|, \|x_+^k(y^k, z^k) - z^k\|\} \to 0,$$

*there exists a sub-sequence of $\{z^k\}$ converging to some $\bar{z} \in X^*$.*

**Proof** It is just a direct corollary of Lemma B.11. ∎

**Lemma D.7** *Let*
$$M(x) = \left\{ J_{\mathcal{T}(x)} F(x) \quad \mathbf{1} \right\}.$$

512    *Then if $(x, y) \in W^*$, the matrix $M(x)(x)$ is of full row rank.*

**Proof** We prove it by contradiction. If for some $(x^*, y^*, \mu^*, \nu^*)$ satisfying (D.7), $M(x^*)$ is not of full row rank. Without loss of generality, we assume that $\mathcal{T}(x^*) = \{1, 2, \cdots, |\mathcal{T}(x^*)|\}$ Then there exists a nonzero vector $v \in \mathbb{R}^{|\mathcal{T}(x^*)|}$ such that
$$M^T(x^*)v = 0.$$

513    Let $d = \min_{i \in \mathcal{I}[y^*]} \{y_i / |v_i|\}$. Then we define a vector $y' \in \mathbb{R}^m$ as:
$$\begin{aligned} y_i' &= y^* - dv_i, \quad when \quad i \in \mathcal{I}; \\ y_i' &= 0, \quad when \quad otherwise. \end{aligned}$$

514    Notice that $y_i^* = 0$ for any $i \notin \mathcal{T}(x^*)$. Then $y'$ satisfies
$$\begin{aligned} J^T F(x^*) y' &= 0, \\ \sum_{i=1}^m y_i' &= 1, \\ y_i' &\geq 0, i \in \mathcal{T}(x^*), \\ y_i' &= 0, i \notin \mathcal{T}(x^*). \end{aligned}$$

515    Therefore, $(x^*, y', \mu, \nu)$ still satisfies (D.7) . Moreover, let $i_0 \in \mathcal{I}$ satisfying $d = y_{i_0}^* / v_{i_0}$. Then
516    $y_{i_0}' = \nu_{i_0} = 0$. This is a contradiction to the strict complementarity assumption. ∎

517    We then have the following corollary from the above lemma and (D.7):

518    **Corollary D.8** *For any $x^* \in X^*$, there exists only one $y \in Y$ such that $(x^*, y) \in W^*$ and there*
519    *exists only one $(\mu, \nu)$ such that $(x^*, y, \mu, \nu)$ satisfies (D.7).*

**Proof** First, this $(y, \mu, \nu)$ must exist due to the existence of a solution. Next, the solution $y$ must satisfy
$$M^T(x^*)y = (0, 0, \cdots, 0, 1)^T.$$

520    By Lemma D.7, $M^T(x^*)$ is of full column rank hence the solution of $y$ is unique. Furthermore, since
521    $\sum_{i=1}^m y_i = 1$, there is at least one $i$ such that $y_i > 0, \nu_i = 0$. Without loss of generality, we assume
522    that $y_1 > 0, \nu_1 = 0$. Then $\mu = f_1(x^*)$ by (3.5). Further by (3.5), $\nu_i = f_i(x^*) - \mu, i = 2, 3, \cdots, m$.
523    Hence, $\mu, \nu_i$ are uniquely defined. ∎

**Lemma D.9** *If the strict complementarity assumption holds for (1.2) , there exists $\delta > 0, \gamma > 0$, such that if*
$$\|z\| \leq R(x^0, y^0, z^0),$$
*and*
$$\max\{\|x - x_+(y, z)\|, \|y - y_+(z)\|, \|x_+(y, z) - z\|\} < \delta$$
524    $\gamma(M(x^*(z))) \geq \gamma$ *and* $\gamma(M(x(y, z))) \geq \gamma$.

**Proof** We prove it by contradiction. Suppose it is not true, there exists $\{z^k\} \subseteq \mathcal{B}(R(x^0, y^0, z^0))$ such that $\gamma(M(x^*(z^k))) \to 0$ and
$$\max\{\|x^k - x_+^k(y^k, z^k)\|, \|y^k - y_+^k(z^k)\|, \|x_+^k(y^k, z^k) - z^k\|\} \to 0.$$

Since $\mathcal{T}(x)$ has only finite choice, without loss of generality, we assume that $\mathcal{T}(x^*(z^k)) = \mathcal{T}$ for any $k$(passing to a sub-sequence if necessary). By Lemma D.6, there exists a $\bar{z} \in X^*$ such that $z^k \to \bar{z}$. We let
$$\tilde{M}(\bar{z}) = \lim_{k \to \infty} M(x^*(z^k)) = \left\{ J_{\mathcal{T}}(x^*(\bar{z})) \quad \mathbf{1} \right\}.$$

By the continuity of $x^*(\cdot)$ ((B.3) of Lemma B.2) and the continuity of the function of taking the least singular value, we know that
$$\gamma(\tilde{M}(\bar{z})) = 0,$$
where we also use the fact that $x^*(\bar{z}) = \bar{z}$ by Lemma B.8. Moreover, according to the definition of $\mathcal{T}[\bar{z}]$, we have $f_i(x^*(z^k)) > f_j(x^*(z^k))$ for any $k$ with $i \in \mathcal{T}, j \notin \mathcal{T}$. Therefore, we have $f_i(\bar{z}) \geq f_j(\bar{z})$ for $i \in \mathcal{T}, j \notin \mathcal{T}$. Consequently, we have
$$\mathcal{T} \subseteq \mathcal{T}[\bar{z}].$$

Therefore $\tilde{M}(x^*(\bar{z}))$ is a row sub-matrix of $M(\bar{z})$. Consequently, $M(\bar{z})$ is not of full row rank. This is a contradiction! For $x(y, z)$, it is similar to prove the desired result. Hence the details are omitted. ∎

The following lemma shows that if the residuals are small, the active set of $y_+(z)$ and $y(z) \in Y(z)$ are the same.

**Lemma D.10** *If the strict complementarity assumption holds for* (1.2) *, there exists $\delta > 0$, such that if*
$$\|z\| \leq R(x^0, y^0, z^0),$$
*and*
$$\max\{\|x - x_+(y, z)\|, \|y - y_+(z)\|, \|x_+(y, z) - z\|\} < \delta,$$
*we have*
$$\mathcal{A}[y_+(z)] = \mathcal{A}[y(z)], \, for \, some \, y(z) \in Y(z).$$

**Proof** We prove it by contradiction. Suppose that there exists a sequence $\{(x^k, y^k, z^k)\}$ such that
$$\max\{\|x^k - x_+^k(y^k, z^k)\|, \|y^k - y_+^k(z^k)\|, \|x_+^k(y^k, z^k) - z^k\|\} \to 0$$
and
$$\mathcal{A}[y_+^k(z^k)] \neq \mathcal{A}[y(z^k)].$$
Since $\{y_+^k(z^k)\}, \{z^k\}$ are bounded, we assume that $y_+^k(z^k) \to \bar{y}, z^k \to \bar{z}$. We write down the KKT condition for $(x(y_+^k(z^k), z^k), y_+^k(z^k))$ as follows:

$$J^T F(x(y_+^k(z^k), z^k)) y_+^k(z^k) + p(x(y_+^k(z^k), z^k) - z^k) = 0, \tag{D.12}$$

$$\sum_{i=1}^m (y_i^k)_+(z^k) = 1, \tag{D.13}$$

$$(y_i^k)_+(z^k) \geq 0, \forall i \in [m] \tag{D.14}$$

$$\frac{1}{\alpha}(y_i^k)_+(z^k) - \frac{1}{\alpha}y_i^k + f_i(x(y^k, z^k)) + \mu^k - \nu_i^k = f_i(x(y_+^k(z^k), z^k)), \forall i \in [m], \tag{D.15}$$

$$\nu_i^k \geq 0, \nu_i^k(y_i^k)_+(z^k) = 0, \forall i \in [m], \tag{D.16}$$

It is not hard to check that $\mu, \nu$ are bounded. Hence, we assume that $\mu^k \to \bar{\mu}$ and $\nu^k \to \bar{\nu}$. We take limit to (D.12) and make use of the fact that
$$\|y^k - y_+^k(z^k)\| \to 0$$
together with Lemma B.2. We then attain that $(x(\bar{y}, \bar{z}), \bar{y})$ is a min-max solution of (1.2), i.e., $(x(\bar{y}, barz), \bar{y}, \bar{\mu}, \bar{\nu})$ satisfies (D.7). By the strict complementarity assumption, $\bar{\nu}_i > 0$ for $i \in \mathcal{A}[\bar{y}]$ and $\bar{y}_i > 0$ for $i \notin \mathcal{A}[\bar{y}]$. Hence, for $k$ sufficiently large, we have $\mathcal{A}[y_+^k(z^k)] = \mathcal{A}[\bar{y}]$. Similarly, when $k$ is sufficiently large, we have
$$\mathcal{A}[y(z^k)] = \mathcal{A}[\bar{y}].$$

∎

We also write down the KKT conditions for $x^*(z)$ for some $z$.

$$J^T F(x^*(z))y + p(x^*(z) - z) = 0, \tag{D.17}$$

$$\sum_{i=1}^m y_i = 1, \tag{D.18}$$

$$y_i \geq 0, \forall i \in [m] \tag{D.19}$$

$$\mu - \nu_i = f_i(x), \forall i \in [m], \tag{D.20}$$

$$\nu_i \geq 0, \nu_i y_i = 0, \forall i \in [m], \tag{D.21}$$

**Lemma D.11** *If the strict complementarity assumption holds for* (1.2) *, there exists $\delta > 0$, such that if*

$$\|z\| \leq R(x^0, y^0, z^0),$$

*and*

$$\max\{\|x - x_+(y,z)\|, \|y - y_+(z)\|, \|x_+(y,z) - z\|\} < \delta$$

*we have*

$$\mathrm{dist}(y_+(z), y(z)) < \lambda \|x^*(z) - x(y_+(z), z)\|$$

*for some constant $\lambda > 0$.*

**Proof** By Lemma D.10 , if the strict complementarity assumption holds for (1.2) , there exists $\delta > 0$, such that if

$$\|z\| \leq R(x^0, y^0, z^0),$$

and

$$\max\{\|x - x_+(y,z)\|, \|y - y_+(z)\|, \|x_+(y,z) - z\|\} < \delta,$$

we have

$$\mathcal{A}[y_+(z)] = \mathcal{A}[y(z)],$$

for some $y(z) \in Y(z)$. Hence, we have

$$\mathcal{T}(x^*(z)) = \mathcal{T}(x(y_+(z), z)).$$

Let $\mathcal{T} = \mathcal{T}(x^*(z))$. Then for $i \notin \mathcal{T}$, $y_i(z) = (y_+(z))_i = 0$ and $\|y(z) - y_+(z)\| = \|(y(z))_{\mathcal{T}} - (y_+(z))_{\mathcal{T}}\|$. Using the optimality conditions for $x(y_+(z), z)$ (D.12) and $x^*(z)$ (D.17), we have

$$M^T(x(y_+(z), z))(y_+(z))_{\mathcal{T}} + \begin{Bmatrix} p(x(y_+(z), z) - z) \\ 0 \end{Bmatrix} = (0, 0, \cdots, 0, 1), \tag{D.22}$$

and

$$M^T(x^*(z))(y(z))_{\mathcal{T}} + \begin{Bmatrix} p(x^*(z) - z) \\ 0 \end{Bmatrix} = (0, 0, \cdots, 0, 1). \tag{D.23}$$

Note that (D.22) can be written as

$$M^T(x^*(z))(y_+(z))_{\mathcal{T}} = M^T(x^*(z))(y_+(z))_{\mathcal{T}} - M^T(x(y_+(z), z))(y_+(z))_{\mathcal{T}} - \begin{Bmatrix} p(x(y_+(z), z) - z) \\ 0 \end{Bmatrix}. \tag{D.24}$$

By (D.23) and (D.24), we have

$$M^T(x^*(z))((y(z))_{\mathcal{T}} - (y_+(z))_{\mathcal{T}}) = (M^T(x(y_+(z), z)) - M^T(x^*(z)))(y_+(z))_{\mathcal{T}} - \begin{Bmatrix} p(x(y_+(z), z) - x^*(z)) \\ 0 \end{Bmatrix}.$$

Therefore, taking norms to the above and the Lemma D.9, we have

$$\begin{aligned} \gamma\|(y_+(z))_{\mathcal{T}} - (y(z))_{\mathcal{T}}\| &\leq \sqrt{m}L\|x(y_+(z), z) - x^*(z)\|\|(y_+(z))_{\mathcal{T}}\| + p\|x(y_+(z), z) - x^*(z)\| \\ &\leq (\sqrt{m}L + p)\|x^*(z) - x(y_+(z), z)\|, \end{aligned}$$

where the first inequality uses the Lipschitz-continuity of $\nabla_x f_i$ and the second is because $\|y_+(z)\| \leq 1$. Hence, we finish the proof with $\lambda = (p + \sqrt{m}L)/\gamma$. ∎

**Proof** [Proof of Lemma B.13] By Lemma B.10 and Lemma D.11, we have

$$\|x(y_+(z), z) - x^*(z)\| \leq \frac{1 + \alpha L}{\lambda(p - L)}\|y - y_+(z)\|,$$

which finishes the proof with $\sigma_5 = \frac{1+\alpha L}{\lambda(p-L)}$. ∎

## E    Discussion of the strict complementarity condition

In this section, we discuss some issues about the strict complementarity assumption. First, notice that the min-max problem (1.1) and (1.2) are both variational inequalities. As mentioned in the main text of the paper, the strict complementarity assumption is common in the field of variation inequality [36, 37]. While this assumption is popular, it is still interesting to weaken the assumption. Inspired by Lemma D.7, we prove Theorem 3.6 and Lemma 4.2 using a weaker regularity assumption rather than the strict complementarity assumption:

**Assumption E.1**  *For any* $(x^*, y^*) \in W^*$, *the matrix* $M(x^*)$ *is of full column rank.*

Here recall that
$$M(x^*) = \left\{ J_{\mathcal{T}(x^*)} \quad \mathbf{1} \right\}.$$

We say that Assumption E.1 is weaker since the strict complementarity assumption (Assumption D.4) can imply Assumption E.1 according to Lemma D.7. For this assumption, we have the following two claims:

1. If we replace Assumption D.4 by Assumption E.1 in Theorem 3.6, we can attain a same result;

2. In a robust regression problem (will define in E.2), if the data is joint from a continuous distribution, this regularity assumption holds with probability 1.

### E.1    Replacing Assumption D.4 by Assumption E.1 in Theorem 3.6

In this section, we will see that we can prove the dual error bound (Lemma 4.2) using Assumption E.1 instead of Assumption D.4.

**Lemma E.2**  *Let*
$$x_+(y, z) = P_X(x - \nabla_x K(x, z; y)).$$

*If Assumption E.1 and the bounded level set assumption hold for* (1.2), *there exists* $\delta > 0$, *such that if*
$$\|z\| \le R(x^0, y^0, z^0),$$
*and*
$$\max\{\|x - x_+(y, z)\|, \|y - y_+(z)\|, \|x_+(y, z) - z\|\} < \delta$$
*we have*
$$\|x(y_+(z), z) - x^*(z)\| < \sigma_5 \|y - y_+(z)\|$$
*for some constant* $\sigma_5 > 0$.

Using this Lemma, we can prove Theorem 3.6 using Assumption E.1:

**Theorem E.3**  *Consider solving Problem 1.2 by Algorithm 2 or Algorithm 3. Suppose that Assumption E.1 holds and either Assumption 3.5 holds or assume* $\{z^t\}$ *is bounded. Then there exist constants*[1] $\beta'$ *and* $\beta''$ *depending on the problem such that the following holds.*

1. *(One-block case) If we choose the parameters in Algorithm 2 as in* (C.2) *and further let* $\beta < \beta'$, *then we have:*

    (a) *Every limit point of* $(x^t, y^t)$ *is a solution of* (1.2).
    (b) *The iteration complexity of Algorithm 2 to attain an* $\epsilon$-*solution is* $\mathcal{O}(1/\epsilon^2)$.

2. *(Multi-block case) Consider using Algorithm 3 to solve Problem 1.2. If we replace the condition for* $\alpha$ *in* (C.2) *by* $\alpha < \min\{\frac{1}{11L}, \frac{c^2(p-L)^2}{4L(1+c(p+L)N^{3/2}+c^2(p-L)^2)}\}$ *and further require* $\beta < 1/\sqrt{T}, \beta < 1/\sqrt{T}$ *and* $\beta < \beta''$, *then we have the same results as in the one-block case.*

## E.2 The rationality of Assumption E.1

Intuitively, the assumption E.1 holds for "generic problem". We rigorously justify this intuition for a simple problem. More specifically, we prove that this regularity assumption is generic for a robust regression problem using square loss, i.e., the regularity condition holds with probability 1 if the outputs of the data points are joint from some continuous distribution. Consider the following problem:

$$\min_{x \in \mathbb{R}^n} \max_{y \in Y} \frac{1}{2} y_i (\ell_i - \Psi(x, \xi_i))^2, \tag{E.1}$$

where $Y$ is the probability simplex , $\Psi(\cdot)$ is a smooth function used to fit the data (for example the neural network) and $\xi_i, \ell_i$ are the input and the output of the $i$-th data point. We define $\Psi_i(x) = \Psi(x, \xi_i)$ for convenience. We further make the following mild assumptions:

**Assumption E.4** $\ell_i$ *is joint independently from a continuous distribution over a positive measure set $\mathcal{L}_i \subseteq \mathbb{R}$.*

Here a continuous distribution over $\mathcal{L}_i$ means that for any zero measure set $\mathcal{S} \subseteq \mathcal{R}$, $\Pr(x \in \mathcal{S} \cap \mathcal{L}_i) = 0$. With assumption, for any zero measure set $\mathcal{S} \subseteq \mathbb{R}^m$, $\Pr((\ell_1, \cdots, \ell_m)^T \in \mathcal{S} \cap \prod_{i=1}^m \mathcal{L}_i) = 0$.

**Assumption E.5** *Let $\Psi(x) = (\Psi_1(x), \cdots, \Psi_m(x))^T$. Then $\Psi(\mathbb{R}^n) \cap \prod_{i=1}^m \mathcal{L}_i = \Omega$, where $\Omega$ is a zero measure set in $\prod_{i=1}^m \mathcal{L}_i$.*

This assumption means that $\min_x \max_{y \in Y} f_i(x) > 0$ with probability 1. This assumption is reasonable. If there exists an $x^*$ such that $\max_i f_i(x^*) = 0$, then becaus $f_i(x) \geq 0$, we have $f_i(x^*) = 0$ for all $i$. In this case, we do not need the min-max fomulation! We just need to solve the finite sum problem $\min_x \sum_{i=1}^m f_i(x)$. However, in many cases, the uncertainty is large, we do need the robust optimization formulation. So in these cases, Assumption E.5 is reasonable.

Moreover, we have the following lemma:

**Lemma E.6** *Suppose that Assumption E.4 holds. If $m > n$, Assumption E.5 holds with probability 1.*

**Proof** It is direct from the claim that a smooth map $\Psi$ maps a zero measure set into a zero measure set. Specializing to this lemma, the map $\Psi$ maps $\mathbb{R}^n$ into $\mathbb{R}^m$, hence the image $\Psi(\mathbb{R}^n)$ is of zero measure since $\mathbb{R}^n$ is a zero measure set of $\mathbb{R}^m$. Therefore, $\Psi(\mathbb{R}^n) \cap \prod_{i=1}^m \mathcal{L}_i$ is zero measure in $\mathbb{R}^m$. ∎

Then we have the following result:

**Proposition E.7** *Suppose that Assumption E.4 and Assumption E.5 hold. Then with probability 1, every solution of* (E.1) *satisfies Assumption E.1.*

## E.3 Proof of Lemma E.2 and Theorem E.3

For a set $\mathcal{S} \subseteq [m]$, we define

$$M_{\mathcal{S}}(x) = \left\{ J_{\mathcal{S}F(x; \ell_{\mathcal{S}})} \quad \mathbf{1} \right\},$$

where $J_{\mathcal{S}} F(x; \ell_{\mathcal{S}}) = ((\Psi_i(x) - \ell_i) \nabla_x \Psi_i(x) \mid i \in \mathcal{S})$.

Similar to the proof of Theorem 3.6, to prove Theorem E.3, we only to prove Lemma E.2. Hence, in this section, we only prove Lemma E.2. The proof is similar to the proof of Lemma 4.2. Hence we only give the main steps. First, similar to Lemma D.9, we have the following:

**Lemma E.8** *If Assumption E.1 holds for Problem* (E.1) *, there exists $\delta > 0, \gamma > 0$, such that if*

$$\|z\| \leq R(x^0, y^0, z^0),$$

*and*

$$\max\{\|x - x_+(y, z)\|, \|y - y_+(z)\|, \|x_+(y, z) - z\|\} < \delta,$$

*then $\gamma(M_{\mathcal{T}(y,z)}(x^*(z))) \geq \gamma$ and $\gamma(M_{\mathcal{T}(y,z)}(x(y_+(z), z))) \geq \gamma$, where*

$$\mathcal{T}(y, z) = \mathcal{T}(x^*(z)) \cup \mathcal{T}(x(y_+(z), z)).$$

**Proof** We prove it by contradiction. Suppose it is not true, there exist $\{x^k\}$, $\{y^k\} \subseteq Y$ and $\{z^k\} \subseteq \mathcal{B}(R(x^0, y^0, z^0))$ such that $\gamma(M_{\mathcal{T}^k}(x^*(z^k)))$, $\gamma(M_{\mathcal{T}^k}(x(y_+^k(z^k), z^k))) \to 0$ and

$$\max\{\|x^k - x_+^k(y^k, z^k)\|, \|y^k - y_+^k(z^k)\|, \|x_+^k(y^k, z^k) - z^k\|\} \to 0,$$

where $\mathcal{T}^k = \mathcal{T}(x^*(z^k)) \cup \mathcal{T}(x(y_+^k(z^k), z^k))$. Since $\mathcal{T}^k$ has only finite choice, without loss of generality, we assume that $\mathcal{T}^k = \mathcal{T}$ for any $k$(passing to a sub-sequence if necessary). By Lemma D.6, there exists a $\bar{z} \in X^*$ such that $z^k \to \bar{z}$. Hence, by Lemma B.2 and Lemma B.8, we have

$$x^*(z^k) \to x^*(\bar{z}) = \bar{z}.$$

Therefore by the definition of $\mathcal{T}(x)$, when $k$ is sufficiently large, $\mathcal{T}(x^*(z^k)) \subseteq \mathcal{T}(x^*(\bar{z})) = \mathcal{T}(\bar{z}))$. Moreover, since $\|y^k - y_+^k(z^k)\| \to 0$, by Lemma B.10, we have

$$\|x(\bar{y}_+^k(z^k), z^k) - x^*(z^k)\| \to 0.$$

and hence $\mathcal{T}(x(y_+^k(z^k), z^k)) \subseteq \mathcal{T}(\bar{z})$. Then $\mathcal{T}^k \subseteq \mathcal{T}(\bar{z})$ and $\gamma(M)_{\mathcal{T}^k} = 0$, which contradicts Assumption E.1. ∎

We then can attain a result similar to Lemma D.11.

**Lemma E.9** *If Assumption E.1 holds for* (1.2) *, there exists $\delta > 0$, such that if*
$$\|z\| \leq R(x^0, y^0, z^0),$$
*and*
$$\max\{\|x - x_+(y, z)\|, \|y - y_+(z)\|, \|x_+(y, z) - z\|\} < \delta$$
*we have*
$$\operatorname{dist}(y_+(z), y(z)) < \lambda\|x^*(z) - x(y_+(z), z)\|$$
*for some constant $\lambda > 0$.*

**Proof** By Lemma E.8, we can find a $\delta > 0$ and a $\gamma > 0$, such that if
$$\|z\| \leq R(x^0, y^0, z^0),$$
and
$$\max\{\|x - x_+(y, z)\|, \|y - y_+(z)\|, \|x_+(y, z) - z\|\} < \delta,$$
then $\gamma(M_{\mathcal{T}(y,z)}(x^*(z))) \geq \gamma$ and $\gamma(M_{\mathcal{T}(y,z)}(x(y_+(z), z))) \geq \gamma$, where
$$\mathcal{T}(y, z) = \mathcal{T}(x^*(z)) \cup \mathcal{T}(x(y_+(z), z)).$$
Let $\mathcal{T} = \mathcal{T}(y, z)$. Then for $i \notin \mathcal{T}$, $y_i(z) = (y_+(z))_i = 0$ and $\|y(z) - y_+(z)\| = \|(y(z))_{\mathcal{T}} - (y_+(z))_{\mathcal{T}}\|$. Using the optimality conditions for $x(y_+(z), z)$ (D.12) and $x^*(z)$ (D.17), we have

$$M_{\mathcal{T}}^T(x(y_+(z), z))(y_+(z))_{\mathcal{T}} + \begin{Bmatrix} p(x(y_+(z), z) - z) \\ 0 \end{Bmatrix} = (0, 0, \cdots, 0, 1), \qquad (\text{E.2})$$

and

$$M_{\mathcal{T}}^T(x^*(z))(y(z))_{\mathcal{T}} + \begin{Bmatrix} p(x^*(z) - z) \\ 0 \end{Bmatrix} = (0, 0, \cdots, 0, 1). \qquad (\text{E.3})$$

Note that (E.2) can be written as

$$M_{\mathcal{T}}^T(x^*(z))(y_+(z))_{\mathcal{T}} = M_{\mathcal{T}}^T(x^*(z))(y_+(z))_{\mathcal{T}} - M_{\mathcal{T}}^T(x(y_+(z), z))(y_+(z))_{\mathcal{T}} - \begin{Bmatrix} p(x(y_+(z), z) - z) \\ 0 \end{Bmatrix}.$$
$$(\text{E.4})$$

By (E.3) and (E.4), we have

$$M_{\mathcal{T}}^T(x^*(z))(y(z) - y_+(z)) = (M_{\mathcal{T}}^T(x(y_+(z), z)) - M_{\mathcal{T}}^T(x^*(z)))(y_+(z))_{\mathcal{T}} - p(x(y_+(z), z) - x^*(z)).$$

Therefore, taking norms to the above and the Lemma D.9, we have

$$\begin{aligned} \gamma\|(y_+(z))_{\mathcal{T}} - (y(z))_{\mathcal{T}}\| &\leq \sqrt{m}L\|x(y_+(z), z) - x^*(z)\|\|(y_+(z))_{\mathcal{T}}\| + p\|x(y_+(z), z) - x^*(z)\| \\ &\leq (\sqrt{m}L + p)\|x^*(z) - x(y_+(z), z)\|, \end{aligned}$$

where the first inequality uses the Lipschitz-continuity of $\nabla_x f_i$ and the second is because $\|y_+(z)\| \leq 1$. Hence, we finish the proof with $\lambda = (p + \sqrt{m}L)/\gamma$. ∎

Then Lemma E.9 and Lemma 4.3 yield Theorem E.3.

 **E.4 Proof of Proposition E.7**

For a set $\mathcal{S} \subseteq [m]$, we define

$$M_{\mathcal{S}}(x; \ell_{\mathcal{S}}) = \left\{ J_{\mathcal{S}F(x;\ell_{\mathcal{S}})} \quad \mathbf{1} \right\},$$

where $J_{\mathcal{S}}F(x; \ell_{\mathcal{S}}) = ((\Psi_i(x) - \ell_i)\nabla_x \Psi_i(x) \mid i \in \mathcal{S})$.

**Proof** Define the event $\mathcal{E}_{\mathcal{T},\mathcal{P}}$ to be: there exists a solution $(x^*, y^*) \in W^*$, such that $M(x^*)$ is not of full row rank, $\mathcal{T}(x^*) = \mathcal{T}$ and $\Psi_i(x^*) - \ell_i \geq 0$ for $i \in \mathcal{P}$ and $\Psi_i(x^*) - \ell_i$ for $i \notin \mathcal{P}$. Then Proposition E.7 is equivalent to the claim:

$$\Pr(\cup_{\mathcal{T} \subseteq [m], \mathcal{P} \subseteq \mathcal{T}} \mathcal{E}_{\mathcal{T},\mathcal{P}}) = 0.$$

Since there are only finite choice of the sets $\mathcal{T}$ and $\mathcal{P}$, we only need to prove that for any $\mathcal{T} \subseteq [m]$ and $\mathcal{P} \subseteq \mathcal{T}$, $\mathcal{E}_{\mathcal{T},\mathcal{P}}$ holds with probability 0, Without loss of generality, we let $\mathcal{T} = \{1, 2, \cdots, k\}$ and $\mathcal{P} = \{1, 2, \cdots, p\}$ with $p \leq k$. We define $\delta_i$ for $i \in [k]$ as $\delta_i = 1$ for $i \in \mathcal{P}$ and $\delta_i = -1$ otherwise. Then if $\mathcal{E}_{\mathcal{T},\mathcal{P}}$ holds, there exists an $x^* = (x_1^*, \cdots, x_n^*)^T \in X^*$ and $x_{n+1} \in \mathbb{R}$, such that

1. $(x_1^*, \cdots, x_n^*)^T \in X^*$;
2. $x_{n+1}^* \geq 0$;
3. $\mathcal{T}(x^*) = \mathcal{T}$;
4. $\Psi_i(x^*) - \ell_i = x_{n+1}^* \geq 0$ for $i \in \mathcal{P}$ and $\Psi_i(x^*) - \ell_i = -x_{n+1}^* \leq 0$ for $i \notin \mathcal{P}$.
5. $M_{\mathcal{T}}(x_1^*, \cdots, x_n^*; \ell_1, \cdots, \ell_k)$ is row rank deficient.

Define $\bar{X}_{\mathcal{T},\mathcal{P}}^*(\ell_1, \cdots, \ell_k)$ to be the set of all $x^* \in X^*$ satisfying the above conditions. Consider the map $G : \mathbb{R}^{n+1} \to \mathbb{R}^k$ defined as

$$G(x_1, \cdots, x_{n+1}) = (\Psi_1(x_1, \cdots, x_n) - \delta_1 x_{n+1}, \cdots, \Psi_k(x_1, \cdots, x_n) - \delta_k x_{n+1})^T.$$

Then $G(x_1^*, \cdots, x_{n+1}^*) = (\ell_1, \cdots, \ell_k)^T$ for any $(x_1^*, \cdots, x_{n+1}^*)^T \in \bar{X}_{\mathcal{T},\mathcal{P}}^*(\ell_1, \cdots, \ell_k)$. Define the set $\bar{X}_{\mathcal{T},\mathcal{P}} \subseteq \mathbb{R}^{n+1}$ be the collection of all $(x_1, \cdots, x_{n+1})$ satisfying:

1. $x_{n+1} > 0$.
2. there exist $\bar{\ell}_1, \cdots, \bar{\ell}_k$ with $\Psi_i(x_1, \cdots, x_n) - \bar{\ell}_i = x_{n+1}$ for $i \in \mathcal{P}$ and $\Psi_i(x_1, \cdots, x_n) - \ell_i = -x_{n+1}$ for $i \notin \mathcal{P}$.
3. $M_{\mathcal{T}}(x_1, \cdots, x_n; \bar{\ell}_1, \cdots, \bar{\ell}_k)$ is rank deficient.
4. $(\bar{\ell}_1, \cdots, \bar{\ell}_k)^T \in \prod_{i=1}^k \mathcal{L}_i$.

Therefore, if $\mathcal{E}_{\mathcal{T},\mathcal{P}}$ holds, we have

$$(\ell_1, \cdots, \ell_m)^T \in (G(\bar{X}_{\mathcal{T},\mathcal{P}}) \cap \prod_{i=1}^k \mathcal{L}_i) \times \prod_{i=k+1}^m \mathcal{L}_i \cup \Omega.$$

For $(x_1, \cdots, x_{n+1})^T \in \bar{X}_{\mathcal{T},\mathcal{P}}$, notice that $JG(x_1, \cdots, x_{n+1})$ is attained by doing elementary matrix transformation to the matrix $M_{\mathcal{T}}(x_1, \cdots, x_n; \bar{\ell}_1, \cdots, \bar{\ell}_k)$, i.e., multiplying the first $k$ columns of $M_{\mathcal{T}}(x_1, \cdots, x_n; \bar{\ell}_1, \cdots, \bar{\ell}_k)$ by $1/x_{n+1}$ and multiplying the $k + 1$-th column of $M_{\mathcal{T}}(x_1, \cdots, x_n; \bar{\ell}_1, \cdots, \bar{\ell}_k)$ by $-1$ and then multiplying the $i$-th row by $\delta_i$ for $i \in [n]$. Therefore, $M_{\mathcal{T}}(x_1, \cdots, x_n; \bar{\ell}_1, \cdots, \bar{\ell}_k)$ is also rank deficient.

Consequently, $G(x_1, \cdots, x_{n+1})$ with $(x_1, \cdots, x_{n+1})^T \in \bar{X}_{\mathcal{T},\mathcal{P}}$ is a critic value of $G$ (see [40]). Then by Sard's Theorem [40], $G(\bar{X}_{\mathcal{T},\mathcal{P}})$ is a zero measure set in $\mathbb{R}^k$. Hence, $G(\bar{X}_{\mathcal{T},\mathcal{P}}) \cap \prod_{i=1}^k \mathcal{L}_i$ is a zero measure set in $\prod_{i=1}^k \mathcal{L}_i$. Recall that if $\mathcal{E}_{\mathcal{T},\mathcal{P}}$ holds, we have

$$(\ell_1, \cdots, \ell_m)^T \in \mathcal{Z} = (G(\bar{X}_{\mathcal{T},\mathcal{P}}) \cap \prod_{i=1}^k \mathcal{L}_i) \times \prod_{i=k+1}^m \mathcal{L}_i \cup \Omega.$$

By the above analysis, $G(\bar{X}_{\mathcal{T},\mathcal{P}}) \cap \prod_{i=1}^k \mathcal{L}_i$ is a zero measure set in $\prod_{i=1}^k \mathcal{L}_i$. Hence, $(G(\bar{X}_{\mathcal{T},\mathcal{P}}) \cap \prod_{i=1}^k \mathcal{L}_i) \times \prod_{i=k+1}^m \mathcal{L}_i$ is a zero measure set in $\prod_{i=1}^m \mathcal{L}_i$. Also by Assumption E.5, $\Omega$ is a zero measure set in $\prod_{i=1}^m \mathcal{L}_i$. Consequently, $\mathcal{Z}$ is a zero measure set in $\prod_{i=1}^m \mathcal{L}_i$. Then by the continuity of the distribution of $\ell$, we finish the proof. ∎

 # F   Details in Experiments

Recall the procedure of training a robust neural network against adversarial attacks can be formulated as a min-max problem:

$$\min_{\mathbf{w}} \sum_{i=1}^{N} \max_{\delta_i, \text{ s.t. } |\delta_i|_\infty \leq \varepsilon} \ell(f(x_i + \delta_i; \mathbf{w}), y_i), \tag{F.1}$$

where $\mathbf{w}$ is the parameter of the neural network, the pair $(x_i, y_i)$ denotes the $i$-th data point, and $\delta_i$ is the perturbation added to data point $i$.

As (F.1) is nonconvex-nonconcave and thus difficult to solve directly, researchers introduce an approximation of (F.1) [20] where the approximated problem has a concave inner problem. The approximation is first replacing the inner maximization problem in F.1 with a finite max problem:

$$\min_{\mathbf{w}} \sum_{i=1}^{N} \max\left\{\ell(f(\hat{x}_{i0}(\mathbf{w}); \mathbf{w}), y_i), \ldots, \ell(f(\hat{x}_{i9}(\mathbf{w}); \mathbf{w}), y_i)\right\}, \tag{F.2}$$

where each $\hat{x}_{ij}(\mathbf{w})$ is the result of a targeted attack on sample $x_i$ by changing the output of the network to label $j$.

To obtain the targeted attack $\hat{x}_{ij}(\mathbf{w})$, we need to introduce an additional procedure. Recall the images in MNIST have 10 classifications, thus the last layer of the neural network architecture for learning classification have 10 different neurons. To obain any targeted attack $\hat{x}_{ij}(\mathbf{w})$, we perform gradient ascent for $K$ times:

$$x_{ij}^{k+1} = \text{Proj}_{B(x,\varepsilon)}\left[x_{ij}^k + \alpha \nabla_x (Z_j(x_{ij}^k, \mathbf{w}) - Z_{y_i}(x_{ij}^k, \mathbf{w}))\right], \ k = 0, \cdots, K-1,$$

and let $\hat{x}_{ij}(\mathbf{w}) = x_{ij}^K$. Here, $Z_j$ is the network logit before softmax corresponding to label $j$; $\alpha > 0$ is the step-size; and $\text{Proj}_{B(x,\varepsilon)}[\cdot]$ is the projection to the infinity ball with radius $\varepsilon$ centered at $x$. Using the same setting in [20], we set the iteration number as $K = 40$, the stepsize as $\alpha = 0.01$, and the perturbation level $\epsilon$ chosen from $\{0.0, 0.1, 0.2, 0.3, 0.4\}$.

Now we can replace the finite max problem (F.2) with a concave problem over a probabilistic simplex, where the entire problem is non-convex in $w$, but concave in $\mathbf{t}$:

$$\min_{\mathbf{w}} \sum_{i=1}^{N} \max_{\mathbf{t} \in \mathcal{T}} \sum_{j=0}^{9} t_j \ell\left(f\left(x_{ij}^K; \mathbf{w}\right), y_i\right), \ \mathcal{T} = \{(t_1, \cdots, t_m) \mid \sum_{i=1}^{m} t_i = 1, t_i \geq 0\}. \tag{F.3}$$

We use Convolutional Neural Network(CNN) with the architecture detailed in Table 3 in the experiments. This setting is the same as in [20].

| Layer Type | Shape |
|---|---|
| Convolution + ReLU | $5 \times 5 \times 20$ |
| Max Pooling | $2 \times 2$ |
| Convolution + ReLU | $5 \times 5 \times 50$ |
| Max Pooling | $2 \times 2$ |
| Fully Connected + ReLU | 800 |
| Fully Connected + ReLU | 500 |
| Softmax | 10 |

**Table 3:** Model Architecture for the MNIST dataset.

The results are listed in Table 2. The first three lines are the results obtained from [20] and the fourth line is obtained by using the code provided in [20] to train their algorithm. As for comparison, we run our algorithm 2 for the same number of iterations (100 iterations) with parameter $p = 0.2, \beta = 0.8$ and $\alpha = 0.5$. In the experiment, to compute the projection of a vector of dimension $d$ over the probability simplex, we use the algorithm from [41] which has a complexity $\mathcal{O}(d \log d)$.

## Footnotes

[1]These two constants, $\beta'$ and $\beta''$, are independent of $\epsilon$ and $T$ and will be discussed in the appendix.

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

## A   Notations

We first list some notations which will be used in the appendix.

1. $[m] = \{1, 2, \cdots, m\}$.
2. $W^*$ is the solution set of (1.1) or (1.2). $X^*$ is the set of all solutions $x^*$, i.e., $x^* \in X^*$ if there exists a $y^*$ such that $(x^*, y^*) \in W^*$.
3. $\mathcal{B}(r)$ is a Euclidian ball of radius $r$ for proper dimension.
4. $\mathrm{dist}(v, S)$ means the Euclidian distance from a point $v$ to a set $S$.
5. For a vector $v$, $v_i$ means the $i$-th component of $v$. For a set $\mathcal{S}$, $v_{\mathcal{S}} \in \mathbb{R}^{|\mathcal{S}|}$ is the vector containing all components $v_i$'s with $i \in \mathcal{S}$.
6. Let $\mathbf{A} \in \mathbb{R}^{n \times m}$ be a matrix and $\mathcal{S} \subseteq [m]$ be an index set. Then $\mathbf{A}_{\mathcal{S}}$ represents the row sub-matrix of $A$ corresponding to the rows with index in $\mathcal{S}$.
7. For a matrix $\mathbf{M}$, $\gamma(\mathbf{M})$ is the smallest singular value of $\mathbf{M}$.
8. The projection of a point $y$, onto a set $X$ is defined as $P_X(y) = \mathrm{argmin}_{x \in X} \frac{1}{2}\|x - y\|^2$.

## B   Proof of the two main theorems: one-block case

In this section, we prove the two main theorems in one-block case. The proof of the multi-block case is similar and will be given in the next section.

**Proof Sketch.**

- In Step 1, we will introduce the potential function $\phi^t$ which is shown to be bounded below. To obtain the convergence rate of the algorithms, we want to prove the potential function can make sufficient decrease at every iterate $t$, i.e., we want to show $\phi^t - \phi^{t+1} > 0$.
- In Step 2, we will study this difference $\phi^t - \phi^{t+1}$ and provide a lower bound of it in Proposition 4.2. Notice that a negative term (4.3) will show up in the lower bound, and we have to carefully analyze the magnitude of this term to obtain $\phi^t - \phi^{t+1} > 0$.
- Analyzing the negative term will be the main difficulty of the proof. In Step 3, we will discuss how to deal with this difficulty for solving Problem 1.1 and Problem 1.2 separately.
- Finally, we will show the potential function makes a sufficient decrease at every iterate as stated in (4.4), and will conclude our proof by computing the number of iterations to achieve an $\epsilon$-solution in Lemma B.12.

### B.1   The potential function and basic estimate

Recall that the potential function is:

$$\phi^t = \Phi(x^t, z^t; y^t) = K(x^t, z^t; y^t) - 2d(y^t, z^t) + 2P(z^t),$$

where

$$K(x, z; y) = f(x, y) + \frac{p}{2}\|x - z\|^2,$$
$$d(y, z) = \min_{x \in X} K(x, z; y),$$
$$P(z) = \min_{x \in X} \max_{y \in Y} K(x, z; y).$$

Also note that if $p > L$, $K(x, z; y)$ is strongly convex of $x$ with modular $p - L$ and $\nabla_x K(x, z; y)$ is Lipschitz-continuous of $x$ with a constant $L + p$. We also use the following notations:

1. $h(x, z) = \max_{y \in Y} K(x, z; y)$.
2. $x(y, z) = \arg\min_{x \in X} K(x, z; y)$, $x^*(z) = \arg\min_{x \in X} h(x, z)$.
3. The set $Y(z) = \arg\max_{y \in Y} d(y, z)$.
4. $y_+(z) = P_Y(y + \alpha \nabla_y K(x(y, z), z; y))$.
5. $x_+(y, z) = P_X(x - c \nabla_x K(x, z; y))$.

First of all, we can prove that $\phi^t$ is bounded from below:

**Lemma B.1** *We have*

$$\phi(x, y, z) \geq \underline{f}.$$

**Proof** By the definition of $d(\cdot)$ and $P(\cdot)$, we have

$$K(x, z; y) \geq d(y, z), \quad P(z) \geq d(y, z), \quad , P(z) \geq \underline{f}. \tag{B.1}$$

Hence, we have

$$
\begin{aligned}
\phi(x, y, z) &= P(z) + (K(x, z; y) - d(y, z)) + (P(z) - d(y, z)) \\
&\geq P(z) \\
&\geq \underline{f}.
\end{aligned}
$$

$\blacksquare$

Next, we state some "error bounds".

**Lemma B.2** *There exist constants $\sigma_1, \sigma_2, \sigma_3$ independent of $y$ such that*

$$\|x(y, z) - x(y, z')\| \leq \sigma_1 \|z - z'\|, \tag{B.2}$$
$$\|x^*(z) - x^*(z')\| \leq \sigma_1 \|z - z'\|, \tag{B.3}$$
$$\|x(y, z) - x(y', z)\| \leq \sigma_2 \|y - y'\|, \tag{B.4}$$
$$\|x^{t+1} - x(y^t, z^t)\| \leq \sigma_3 \|x^t - x^{t+1}\|, \tag{B.5}$$

*for any $y, y' \in Y$ and $z, z' \in X$, where $\sigma_1 = \frac{p}{-L+p}$, $\sigma_2 = \frac{2(p+L)}{p-L}$, $\sigma_3 = \frac{1+(c(-L+p))}{c(-L+p)}$..*

**Proof** The proofs of (B.2), (B.3) and (B.5) are the same as those in Lemma 3.6 in [30] and hence omitted. We only need to prove (B.4). Using the strong convexity of $K(\cdot, z; y)$ of $x$, we have

$$K(x(y, z), z; y) - K(x(y', z), z; y) \leq -\frac{-L + p}{2}\|x(y, z) - x(y', z)\|^2, \tag{B.6}$$

$$K(x(y, z), z; y') - K(x(y', z), z; y') \geq \frac{-L + p}{2}\|x(y, z) - x(y', z)\|^2. \tag{B.7}$$

Moreover, using the concavity of $K(x, z; \cdot)$ of $y$, we have

$$
\begin{aligned}
&K(x(y, z), z; y') - K(x(y, z), z; y) \\
&\leq \langle \nabla_y K(x(y, z), z; y), y' - y \rangle. \tag{B.8}
\end{aligned}
$$

Using the Lipschitz-continuity of $\nabla_y K(x, z; \cdot)$, we have

$$
\begin{aligned}
&K(x(y', z), z; y) - K(x(y', z), z; y') \\
&\leq \langle \nabla_y K(x(y', z), z; y), y' - y \rangle + \frac{L}{2}\|y - y'\|^2. \tag{B.9}
\end{aligned}
$$

Combining (B.6) and (B.9), we have

$$(-L+p)\|x(y,z)-x(y',z)\|^2 \tag{B.10}$$

$$\leq \langle \nabla_y K(x(y,z),z;y) - \nabla_y K(x(y',z),z;y), y'-y \rangle + \frac{L}{2}\|y-y'\|^2 \tag{B.11}$$

$$\leq (p+L)\|x(y',z)-x(y,z)\|\|y-y'\| + \frac{L}{2}\|y-y'\|^2, \tag{B.12}$$

where the last inequality uses the Cauchy-schwarz inequality and the Lipschitz-continuity of $\nabla_x K(\cdot,z;y)$ of $x$.

Let $\zeta = \|x(y,z)-x(y',z)\|/\|y-y'\|$. Then (B.10) becomes

$$\zeta^2 \leq \frac{p+L}{p-L}\zeta + \frac{L}{2(p-L)}.$$

Hence, we only need to solve the above quadratic inequality. We have

$$\begin{aligned}
\zeta^2 &\leq \frac{1}{2}\zeta^2 + \frac{1}{2}\left(\frac{p+L}{p-L}\right)^2 + \frac{L}{2(p-L)} \\
&\leq \frac{1}{2}\zeta^2 + \frac{1}{2}\left(\frac{p+L}{p-L}\right)^2 + \frac{p+L}{2(p-L)} \\
&\leq \frac{1}{2}\zeta^2 + \frac{1}{2}\left(\frac{p+L}{p-L}\right)^2 + \frac{1}{2}\left(\frac{p+L}{p-L}\right)^2 \\
&= \frac{1}{2}\zeta^2 + \left(\frac{p+L}{p-L}\right)^2,
\end{aligned}$$

where the first inequality is due to the AM-GM inequality and the third inequality is because $(p+L)/(p-L) > 1$. Therefore

$$\zeta \leq \sqrt{2}\frac{p+L}{p-L} < 2\frac{p+L}{p-L}.$$

Hence, we can take $\sigma_2 = 2(p+L)/(p-L)$ and finish the proof. ∎

The following lemma is a direct corollary of the above lemma:

**Lemma B.3** *The dual function* $d(\cdot,z)$ *is a differentiable function of* $y$ *with Lipschitz continuous gradient*

$$\nabla_y d(y,z) = \nabla_y K(x(y,z),z;y) = \nabla_y f(x(y,z),y)$$

*and*

$$\|\nabla_y d(y',z) - \nabla_y d(y'',z)\| \leq L_d\|y'-y''\|, \quad \forall\, y',y'' \in Y.$$

*with* $L_d = L + L\sigma_2$.

**Remark.** Note that if $p \geq 3L$, then we have $\sigma_2 = 2(p+L)/(p-L) \geq 4L$ and hence

$$L_d \geq 5L. \tag{B.13}$$

**Proof** Using Danskin's theorem in convex analysis [52], we know that $d$ is a differentiable function with

$$\nabla_y d(y,z) = \nabla_y K(x(y,z),z;y) = \nabla_y f(x(y,z),y).$$

To prove the Lipschitz-continuity, we have

$$\begin{aligned}
\|\nabla_y d(y',z) - \nabla_y d(y'',z)\| &= \|\nabla_y K(x(y',z),z;y') - \nabla_y K(x(y'',z),z;y'')\| \\
&\leq \|\nabla_y K(x(y',z),z;y') - \nabla_y K(x(y',z),z;y'')\| \\
&\quad + \|\nabla_y K(x(y',z),z;y'') - \nabla_y K(x(y'',z),z;y'')\| \\
&\leq L\|y'-y''\| + L\|x(y',z)-x(y'',z)\| \\
&\leq L\|y'-y''\| + L\sigma_2\|y'-y''\| = L_d\|y'-y''\|,
\end{aligned}$$

where the last inequality is due to Lemma B.2. ∎

We then prove the following basic estimate.

**Proposition B.4** *We let*

$$p > 3L, c < \frac{1}{p+L}, \alpha < \min\{\frac{1}{11L}, \frac{1}{4L\sigma_3^2}\} = \min\{\frac{1}{11L}, \frac{c^2(p-L)^2}{4L(1+c(p-L))^2}\}, \beta < \min\{\frac{1}{36}, \frac{(p-L)^2}{384p(p+L)^2}\}.$$

$$\tag{B.14}$$

*Then we have*

$$\phi^t - \phi^{t+1} \tag{B.15}$$

$$\geq \quad \frac{1}{8c}\|x^t - x^{t+1}\|^2$$

$$+\frac{1}{8\alpha}\|y^t - y_+^t(z^t)\|^2 + \frac{p}{8\beta}\|z^t - z^{t+1}\|^2 \tag{B.16}$$

$$-24p\beta\|x^*(z^t) - x(y_+^t(z^t), z^t)\|^2 \tag{B.17}$$

To prove this basic estimate, we need a series of lemmas.

**Lemma B.5 (Primal Descent)** *For any t, we have*

$$K(x^t, z^t; y^t) - K(x^{t+1}, z^{t+1}; y^{t+1}) \quad \geq \quad \frac{1}{2c}\|x^t - x^{t+1}\|^2 + \langle \nabla_y K(x^{t+1}, z^t; y^t), y^t - y^{t+1}\rangle$$

$$-\frac{L}{2}\|y^t - y^{t+1}\|^2 + \frac{p}{2\beta}\|z^t - z^{t+1}\|^2. \tag{B.18}$$

**Proof** Notice that the step of updating $x$ is a standard gradient projection, hence we have

$$K(x^t, z^t; y^t) - K(x^{t+1}, z^t; y^t) \geq \frac{1}{2c}\|x^t - x^{t+1}\|^2. \tag{B.19}$$

Next, because $\nabla_y K(x, z; y)$ is $L$-Lipschitz-continuous of $y$, we have

$$K(x^{t+1}, z^t; y^t) - K(x^{t+1}, z^t; y^{t+1}) \geq \langle \nabla_y K(x^{t+1}, z^t; y^t), y^t - y^{t+1}\rangle - \frac{L}{2}\|y^t - y^{t+1}\|^2. \tag{B.20}$$

Based on the update of variable $z^{t+1}$, i.e. $z^{t+1} = z^t + \beta(x^{t+1} - z^t)$, it is easy to show that

$$K(x^{t+1}, z^t; y^{t+1}) - K(x^{t+1}, z^{t+1}; y^{t+1}) \geq \frac{p}{2\beta}\|z^t - z^{t+1}\|^2.$$

Combining (B.19)-(B.21), we finish the proof. ∎

**Lemma B.6 (Dual Ascent)** *For any t, we have*

$$d(y^{t+1}, z^{t+1}) - d(y^t, z^t) \quad \geq \quad \langle \nabla_y d(y^t, z^t), y^{t+1} - y^t\rangle - \frac{L_d}{2}\|y^t - y^{t+1}\|^2$$

$$+\frac{p}{2}(z^{t+1} - z^t)^T(z^{t+1} + z^t - 2x(y^{t+1}, z^{t+1})) \tag{B.21}$$

$$= \quad \langle \nabla_y K(x(y^t, z^t), z^t; y^t), y^{t+1} - y^t\rangle - \frac{L_d}{2}\|y^t - y^{t+1}\|^2 \tag{B.22}$$

$$+\frac{p}{2}(z^{t+1} - z^t)^T(z^{t+1} + z^t - 2x(y^{t+1}, z^{t+1})) \tag{B.23}$$

**Proof** Using Lemma B.3, we have

$$-d(y^{t+1}, z^t) - (-d(y^t, z^t)) \quad \leq \quad -\langle \nabla_y d(y^t, z^t), y^{t+1} - y^t\rangle + \frac{L_d}{2}\|y^t - y^{t+1}\|^2$$

$$= \quad -\langle \nabla_y K(x^{t+1}, z^t; y^t), y^{t+1} - y^t\rangle + \frac{L_d}{2}\|y^t - y^{t+1}\|^2 \tag{B.24}$$

Next,

$$d(y^{t+1}, z^{t+1}) - d(y^{t+1}, z^t)$$

$$= \quad K(x(y^{t+1}, z^{t+1}), z^{t+1}; y^{t+1}) - K(x(y^{t+1}, z^t), z^t; y^{t+1})$$

$$\geq \quad K(x(y^{t+1}, z^{t+1}), z^{t+1}; y^{t+1}) - K(x(y^{t+1}, z^{t+1}), z^t; y^{t+1})$$

$$= \quad \frac{p}{2}\|x(y^{t+1}, z^{t+1}) - z^{t+1}\|^2 - \frac{p}{2}\|x(y^{t+1}, z^{t+1}) - z^t\|^2$$

$$= \quad \frac{p}{2}(z^{t+1} - z^t)^T(z^{t+1} + z^t - 2x(y^{t+1}, z^{t+1})). \tag{B.25}$$

Finally, using (B.24)-(B.25) we finish the proof. ∎

Recall that
$$Y(z) = \{y \in Y \mid \arg\max_{y \in Y} d(y, z)\}$$

Note that
$$P(z) = d(y(z), z),$$

for any $y(z) \in Y(z)$.

**Lemma B.7 (Proximal Descent)** *For any $t \geq 0$, there holds*
$$P(z^{t+1}) - P(z^t) \leq \frac{p}{2}(z^{t+1} - z^t)^T (z^t + z^{t+1} - 2x(y(z^{t+1}, z^t)), \tag{B.26}$$

*where $y(z^{t+1})$ is arbitrary $y$ belongs to the set $Y(z^{t+1})$.*

**Proof** In the rest of the Appendix, $y(z^{t+1})$ denote any $y$ belongs to the set $Y(z^{t+1})$. Using Kakutoni's Theorem, we have
$$\min_{x \in X} \max_{y \in Y} K(x, z; y) = \max_{y \in Y} \min_{x \in X} K(x, z; y),$$

which implies
$$\max_{y \in Y} d(y, z) = \min_{x \in X} h(x, z) = P(z).$$

Hence we have
$$P(z^{t+1}) - P(z^t)$$
$$\overset{(i)}{\leq} P(z^{t+1}) - d(y(z^{t+1}), z^t)$$
$$\overset{(ii)}{=} d(y(z^{t+1}), z^{t+1}) - d(y(z^{t+1}), z^t)$$
$$\overset{(iii)}{=} K(x(y(z^{t+1}), z^t), z^{t+1}; y(z^{t+1})) - K(x(y(z^{t+1}), z^t), z^t; y(z^{t+1}))$$
$$\overset{(iv)}{=} \frac{p}{2}(z^{t+1} - z^t)^T (z^{t+1} + z^t - 2x(y(z^{t+1}), z^t)),$$

where (i) and (iii) are because of (B.1), (ii) is because of the definition of $y(z^{t+1})$ and (iv) is from direct calculation. ∎

**Lemma B.8** $x^*(z) = x(y, z)$ *for any $y \in Y(z)$ and $x^*(z)$ is continuous in $z$. Moreover, we have*
$$z = x^*(z)$$

*if and only if $z \in X^*$.*

We define
$$y_+(z) = P_Y(y + \alpha \nabla_y K(x(y, z), z; y)).$$

Then we have

**Lemma B.9** *We have*
$$\|y^{t+1} - y_+^t(z^t)\| \leq \kappa \|x^t - x^{t+1}\|,$$

*where $\kappa = \alpha L \sigma_3$.*

**Proof** By the nonexpansiveness of the projection operator, we have
$$
\begin{aligned}
\|y^{t+1} - y_+^t(z^t)\| &= \|P_Y(y^t + \alpha \nabla_y K(x(y^t, z^t), z^t; y^t)) - P_Y(y^t + \alpha \nabla_y K(x^{t+1}, z^t; y^t))\| \\
&\leq \|(y^t + \alpha \nabla_y K(x(y^t, z^t), z^t; y^t)) - (y^t + \alpha \nabla_y K(x^{t+1}, z^t; y^t))\| \\
&\leq \alpha L \|x^{t+1} - x(y^t, z^t)\| \\
&\leq \alpha L \sigma_3 \|x^t - x^{t+1}\|,
\end{aligned}
$$

where the first inequality is due to the nonexpansiveness of the projection operator, the second is because of the Lipschitz-continuity of $\nabla_y K$ and the last inequality is because of (B.5). ∎

Now we can prove Proposition B.4.

**Proof** Using the three descent lemma, we have

$$
\Phi^t - \Phi^{t+1}
$$

$$
\geq \quad \frac{1}{2c}\|x^t - x^{t+1}\|^2 + \langle \nabla_y K(x^{t+1}, z^t; y^t), y^t - y^{t+1}\rangle - \frac{L}{2}\|y^t - y^{t+1}\|^2 + \frac{p}{2\beta}\|z^t - z^{t+1}\|^2
$$

$$
+ 2\langle \nabla_y K(x(y^t, z^t), z^t; y^t), y^{t+1} - y^t\rangle - \frac{2L_d}{2}\|y^t - y^{t+1}\|^2 + p(z^{t+1} - z^t)^T(z^{t+1} + z^t - 2x(y^{t+1}, z^{t+1}))
$$

$$
- p(z^{t+1} - z^t)^T(z^t + z^{t+1} - 2x(y(z^{t+1}), z^t))
$$

$$
= \quad \frac{1}{2c}\|x^t - x^{t+1}\|^2 - \frac{L + 2L_d}{2}\|y^t - y^{t+1}\|^2 + \frac{p}{2\beta}\|z^t - z^{t+1}\|^2 + \langle \nabla_y K(x^{t+1}, z^t; y^t), y^{t+1} - y^t\rangle
$$

$$
+ 2\langle \nabla_y K(x(y^t, z^t), z^t; y^t) - \nabla_y K(x^{t+1}, z^t; y^t), y^{t+1} - y^t\rangle \tag{B.27}
$$

$$
+ 2p(z^{t+1} - z^t)^T(x(y(z^{t+1}), z^t) - x(y^{t+1}, z^{t+1})). \tag{B.28}
$$

Using the property of the projection operator and the update of the dual variable $y^t$, we have

$$
\langle \nabla_y K(x^{t+1}, z^t; y^t), y^{t+1} - y^t\rangle \geq \frac{1}{\alpha}\|y^t - y^{t+1}\|^2.
$$

Substituting the above inequality into (B.27), we get

$$
\Phi^t - \Phi^{t+1}
$$

$$
\geq \quad \frac{1}{2c}\|x^t - x^{t+1}\|^2 + (\frac{1}{\alpha} - \frac{L + 2L_d}{2})\|y^t - y^{t+1}\|^2 + \frac{p}{2\beta}\|z^t - z^{t+1}\|^2
$$

$$
+ 2\langle \nabla_y K(x(y^t, z^t), z^t; y^t) - \nabla_y K(x^{t+1}, z^t; y^t), y^{t+1} - y^t\rangle
$$

$$
+ 2p(z^{t+1} - z^t)^T(x(y(z^{t+1}), z^t) - x(y^{t+1}, z^{t+1}))
$$

$$
\geq \quad \frac{1}{2c}\|x^t - x^{t+1}\|^2 + (\frac{1}{\alpha} - \frac{L + 10L}{2})\|y^t - y^{t+1}\|^2 + \frac{p}{2\beta}\|z^t - z^{t+1}\|^2
$$

$$
+ 2\langle \nabla_y K(x(y^t, z^t), z^t; y^t) - \nabla_y K(x^{t+1}, z^t; y^t), y^{t+1} - y^t\rangle
$$

$$
+ 2p(z^{t+1} - z^t)^T(x(y(z^{t+1}), z^t) - x(y^{t+1}, z^{t+1}))
$$

$$
\geq \quad \frac{1}{2c}\|x^t - x^{t+1}\|^2 + \frac{1}{2\alpha}\|y^t - y^{t+1}\|^2 + \frac{p}{2\beta}\|z^t - z^{t+1}\|^2
$$

$$
+ 2\langle \nabla_y K(x(y^t, z^t), z^t; y^t) - \nabla_y K(x^{t+1}, z^t; y^t), y^{t+1} - y^t\rangle
$$

$$
+ 2p(z^{t+1} - z^t)^T(x(y(z^{t+1}), z^t) - x(y^{t+1}, z^{t+1}))
$$

where the second inequality is because of (B.13) and the last inequality is because $\alpha \leq \frac{1}{11L}$.

Notice that

$$
2p(z^{t+1} - z^t)^T(x(y(z^{t+1}), z^t) - x(y^{t+1}, z^{t+1}))
$$

$$
= \quad 2p(z^{t+1} - z^t)^T((x(y(z^{t+1}), z^t) - x(y(z^{t+1}), z^{t+1})) + (x(y(z^{t+1})), z^{t+1}) - x(y^{t+1}, z^{t+1})))
$$

$$
= \quad 2p(z^{t+1} - z^t)^T(x(y(z^{t+1}), z^t) - x(y(z^{t+1}), z^{t+1}))
$$

$$
+ 2p(z^{t+1} - z^t)^T(x(y(z^{t+1})), z^{t+1}) - x(y^{t+1}, z^{t+1}))
$$

$$
\overset{(i)}{\geq} \quad -2p\sigma_1\|z^{t+1} - z^t\|^2 + 2p(z^{t+1} - z^t)^T(x(y(z^{t+1})), z^{t+1}) - x(y^{t+1}, z^{t+1}))
$$

$$
\overset{(ii)}{\geq} \quad -2p\sigma_1\|z^{t+1} - z^t\|^2 - \frac{p}{6\beta}\|z^{t+1} - z^t\|^2 - 6p\beta\|x(y(z^{t+1}), z^{t+1}) - x(y^{t+1}, z^{t+1})\|^2,
$$

where (i) is because of the Cauchy-Schwarz inequality and Lemma B.2 and (ii) is due to the AM-GM inequality. Also we have

$$
2\langle \nabla_y K(x(y^t, z^t), z^t; y^t) - \nabla_y K(x^{t+1}, z^t; y^t), y^{t+1} - y^t\rangle
$$

$$
\geq \quad -2\|\nabla_y K(x(y^t, z^t), z^t; y^t) - \nabla_y K(x^{t+1}, z^t; y^t)\| \cdot \|y^{t+1} - y^t\|
$$

$$
\geq \quad -2L\|x^{t+1} - x(y^t, z^t)\| \cdot \|y^t - y^{t+1}\|
$$

$$
\geq \quad -L\sigma_3^2\|y^t - y^{t+1}\|^2 - L\sigma_3^{-2}\|x^{t+1} - x(y^t, z^t)\|^2
$$

$$
\geq \quad -L\sigma_3^2\|y^t - y^{t+1}\|^2 - L\|x^{t+1} - x^t\|^2,
$$

where the first inequality is because of the Cauchy-Schwarz in equality, the second inequality is because $\nabla_y K = \nabla_y f$ is $L$-Lipschitz-continuous, the third inequality is due to the AM-GM inequality and the last is because of (B.5).

Hence we have

$$\Phi^t - \Phi^{t+1}$$
$$\geq \quad (\frac{1}{2c} - L)\|x^t - x^{t+1}\|^2 + (\frac{1}{2\alpha} - L\sigma_3^2)\|y^t - y^{t+1}\|^2 + (\frac{p}{2\beta} - 2p\sigma_1 - \frac{p}{6\beta})\|z^t - z^{t+1}\|^2$$
$$-6p\beta\|x(y(z^{t+1}), z^{t+1}) - x(y^{t+1}, z^{t+1})\|^2$$

By the conditions of $p, c$, we have

$$1/2c - L \geq 1/4c.$$

By the condition for $\alpha$, we have

$$\alpha < 1/(4L\sigma_3^2),$$

which yields

$$1/(2\alpha) - L\sigma_3^2 \geq 1/(4\alpha)$$

And by the conditions that $\beta < \frac{1}{36}$ and $p \geq 3L$ together with the definition of $\sigma_1$,

$$\frac{p}{2\beta} - 2p\sigma_1 - \frac{p}{6\beta} \geq \frac{p}{4\beta}.$$

Then we have

$$\Phi^t - \Phi^{t+1}$$
$$\geq \quad \frac{1}{4c}\|x^t - x^{t+1}\|^2 + \frac{1}{4\alpha}\|y^t - y^{t+1}\|^2 + \frac{p}{4\beta}\|z^t - z^{t+1}\|^2$$
$$-6p\beta\|x(y(z^{t+1}), z^{t+1}) - x(y^{t+1}, z^{t+1})\|^2 \tag{B.29}$$
$$= \quad \frac{1}{4c}\|x^t - x^{t+1}\|^2 + \frac{1}{4\alpha}\|y^t - y^{t+1}\|^2 + \frac{p}{4\beta}\|z^t - z^{t+1}\|^2$$
$$-6p\beta\|x^*(z^{t+1}) - x(y^{t+1}, z^{t+1})\|^2, \tag{B.30}$$

where the last equality is because of Lemma B.8. By Lemma B.9 and the convexity of the norm square function, we have

$$\|y^{t+1} - y^t\|^2 \quad = \quad \|(y^{t+1} - y_+^t(z^t)) + (y_+^t(z^t) - y^t)\|^2 \tag{B.31}$$
$$\geq \quad \|y^t - y_+^t(z^t)\|^2/2 - \|y^{t+1} - y_+^t(z^t)\|^2 \tag{B.32}$$
$$\geq \quad \|y^t - y_+^t(z^t)\|^2/2 - \kappa^2\|x^t - x^{t+1}\|^2. \tag{B.33}$$

Similarly, by Lemma B.9, (B.4) and the convexity of norm square function, we have

$$\|x^*(z^{t+1}) - x(y^{t+1}, z^{t+1})\|^2 \tag{B.34}$$
$$= \quad \|(x^*(z^{t+1}) - x^*(z^t)) + (x^*(z^t) - x(y_+^t(z^t), z^t)) \tag{B.35}$$
$$+(x(y_+^t(z^t), z^t) - x(y^{t+1}, z^t)) + (x(y^{t+1}, z^t) - x(y^{t+1}, z^{t+1}))\|^2 \tag{B.36}$$
$$\leq \quad 4\|x^*(z^{t+1}) - x^*(z^t)\|^2 + 4\|x^*(z^t) - x(y_+^t(z^t), z^t)\|^2 \tag{B.37}$$
$$+4\|x(y_+^t(z^t), z^t) - x(y^{t+1}, z^t)\|^2 + 4\|x(y^{t+1}, z^t) - x(y^{t+1}, z^{t+1})\|^2 \tag{B.38}$$
$$\leq \quad 4\sigma_1^2\|z^t - z^{t+1}\|^2 + 4\|x^*(z^t) - x(y_+^t(z^t), z^t)\|^2 \tag{B.39}$$
$$+4\sigma_2^2\kappa^2\|x^t - x^{t+1}\|^2 + 4\sigma_1^2\|z^t - z^{t+1}\|^2 \tag{B.40}$$
$$= \quad 8\sigma_1^2\|z^t - z^{t+1}\|^2 + 4\|x^*(z^t) - x(y_+^t(z^t), z^t)\|^2 \tag{B.41}$$
$$+4\sigma_2^2\kappa^2\|x^t - x^{t+1}\|^2. \tag{B.42}$$

Substituting (B.33) and (B.42) to (B.29) yields

$$\phi^t - \phi^{t+1}$$
$$\geq \quad (\frac{1}{4c} - 24p\beta\sigma_2^2\kappa^2 - \kappa^2/(4\alpha))\|x^t - x^{t+1}\|^2$$
$$+\frac{1}{8\alpha}\|y^t - y_+^t(z^t)\|^2 + (\frac{p}{4\beta} - 48p\beta\sigma_1^2)\|z^t - z^{t+1}\|^2$$
$$-24p\beta\|x^*(z^t) - x(y_+^t(z^t), z^t)\|^2.$$

Notice that
$$\alpha < 1/(4L\sigma_3^2) < 1/(4cL^2\sigma_3^2),$$
and hence $\kappa^2/(4\alpha) = \alpha^2 L^2 \sigma_3^2/(4\alpha) < 1/(16c)$. Also we have
$$\beta < 1/(96p\alpha\sigma_2^2),$$
thus
$$24p\beta\sigma_2^2\kappa^2 < \kappa^2/(4\alpha) \le 1/(16c).$$
Consequently, we have
$$(\frac{1}{4c} - 24p\beta\sigma_2^2\kappa^2 - \kappa^2/(4\alpha)) > 1/(8c).$$
By the definition of $\sigma_1$ and the conditions $p \ge 3L, \beta < \frac{1}{36}$, we have
$$(\frac{p}{4\beta} - 48p\beta\sigma_1^2) \ge \frac{p}{8\beta}.$$
Combining the above, we have
$$\begin{aligned}
&\phi^t - \phi^{t+1} \\
&\ge \quad \frac{1}{8c}\|x^t - x^{t+1}\|^2 \\
&\quad + \frac{1}{8\alpha}\|y^t - y_+^t(z^t)\|^2 + \frac{p}{8\beta}\|z^t - z^{t+1}\|^2 \\
&\quad - 24p\beta\|x^*(z^t) - x(y_+^t(z^t), z^t)\|^2,
\end{aligned}$$
which finishes the proof. ∎

## B.2 General nonconvex-concave case

We have the following error bound:

**Lemma B.10** *We have*
$$\begin{aligned}
(p - L)\|x^*(z^t) - x(y_+^t(z^t), z^t)\|^2 \quad &< \quad (1 + \alpha L)\|y^t - y_+^t(z^t)\| \cdot \mathrm{dist}(y_+^t(z^t), Y(z^t)). \\
&\le \quad (1 + \alpha L)\|y^t - y_+^t(z^t)\| \cdot D(Y),
\end{aligned}$$
*where $D(Y)$ is the diameter of $Y$.*

The proof will be given in the next section.

**Lemma B.11** *If*
$$\max\{\|x^t - x^{t+1}\|, \|y^t - y_+^t(z^t)\|, \|z^t - x^{t+1}\|\} \le \bar{\lambda}\epsilon, \tag{B.43}$$
*then $(x^{t+1}, y^{t+1})$ is a $\bar{\lambda}\epsilon-$ solution for some $\bar{\lambda} > 0$.*

**Proof**

By the update of $x^{t+1}$, we have
$$x^{t+1} = \arg\min_{x \in X}\{\langle\nabla_x f(x^t, y^t) + p(x^t - z^t), x - x^t\rangle + \frac{1}{2c}\|x - x^t\|^2 + \iota(x)\}.$$

Therefore, we have
$$0 \in \nabla_x f(x^{t+1}, y^t) + p(x^{t+1} - z^t) + \frac{1}{2c}(x^{t+1} - x^t) + \iota(x^{t+1}). \tag{B.44}$$

Similarly, we have
$$0 \in \arg\min_{y \in Y}\{-\nabla_y f(x^{t+1}, y^t) + \frac{1}{\alpha}(y^{t+1} - y^t) + \iota(y^{t+1})\}. \tag{B.45}$$

We let

$$u = (\nabla_x f(x^{t+1}, y^t) - \nabla_x f(x^t, y^t)) + (\nabla_x f(x^{t+1}, y^{t+1}) - \nabla_x f(x^{t+1}, y^t)) - p(x^{t+1} - z^t) - \frac{1}{2c}(x^{t+1} - x^t)$$

and

$$v = \nabla_y f(x^{t+1}, y^t) - \nabla_y f(x^{t+1}, y^{t+1}) - \frac{1}{\alpha}(y^{t+1} - y^t).$$

By the Lipschitz-continuity of $\nabla_x f(x, y)$, Lemma B.9 and (B.43), we have

$$
\begin{aligned}
\|u\| &\leq L\|x^t - x^{t+1}\| + L\|y^t - y^{t+1}\| + p\epsilon + \frac{1}{2c}\epsilon \\
&\leq (1 + p + 1/2c)\epsilon + L\|y^t - y_+^t(z^t)\| + \|y_+^t(z^t) - y^{t+1}\| \\
&\leq (1 + p + 1/2c)\epsilon + \epsilon + \kappa\epsilon,
\end{aligned}
$$

where the first and the second inequalities are both due to (B.43), the triangular inequality and the Lipschitz-continuity of $\nabla_x f(\cdot)$ and the last inequality is because of Lemma B.9. Similarly, we can prove that

$$\|v\| \leq (L + 1 + \kappa + \frac{1}{\alpha})\epsilon.$$

Hence, we finish the proof with $\eta = 2 + L + p + \kappa + \max\{1/2c, 1/\alpha\}$.

∎

We say that $\phi^t$ decreases sufficiently if

$$\phi^t - \phi^{t+1} \geq \frac{1}{16c}\|x^t - x^{t+1}\|^2 + \frac{1}{16\alpha}\|y^t - y_+^t(z^t)\|^2 + \frac{p\beta}{16}\|z^t - x^{t+1}\|^2. \tag{B.46}$$

**Lemma B.12** *Let $T > 0$. Then if for any $t \in \{0, 1, \cdots, T-1\}$, (B.46) holds, there must exist a $t \in \{1, 2, \cdots, T\}$ such that $(x^t, y^t)$ is an $C/\sqrt{T\beta}$-solution. Moreover, if for any $t \geq 0$, (B.46) holds, Then any limit point of $(x^t, y^t)$ is a solution of (1.2), and the iteration complexity of attaining an $\epsilon$−solution is $\mathcal{O}(1/\epsilon^2)$.*

**Proof**

We have

$$\phi^0 - \underline{f} \geq \sum_{t=0}^{T-1}(\phi^t - \phi^{t+1}) \tag{B.47}$$

$$\geq \min\{1/(16c), 1/(16\alpha), p/16\}\sum_{t=0}^{T-1}\max\{\|x^t - x^{t+1}\|^2, \tag{B.48}$$

$$\|y^t - y_+^t(z^t)\|^2, \beta\|x^{t+1} - z^t\|^2\}, \tag{B.49}$$

where the last inequality is due to (B.46). Therefore, there exists a $t \in \{0, 1, \cdots, T-1\}$ such that

$$\min\{1/(16c), 1/(16\alpha), p/16\}\max\{\|x^t - x^{t+1}\|^2, \|y^t - y_+^t(z^t)\|^2, \beta\|x^{t+1} - z^t\|^2\} \leq (\phi^0 - \underline{f})/T.$$

Since $\beta < 1$, we further get

$$\min\{1/(16c), 1/(16\alpha), p/16\}\max\{\|x^t - x^{t+1}\|^2, \|y^t - y_+^t(z^t)\|^2, \|x^{t+1} - z^t\|^2\} \leq (\phi^0 - \underline{f})/(T\beta).$$

Hence, by Lemma B.11, $(x^{t+1}, y^{t+1})$ is a $\sqrt{(\phi^0 - \underline{f})/((1/8c + 1/8\alpha + 16)T\beta)}$-solution. According to above analysis, If (B.46) holds for any $t$, we can attain an $\epsilon$-solution within

$$(\phi^0 - \underline{f})/(\beta \min\{1/(16c), 1/(16\alpha), p/16\}\epsilon^2)$$

iterations. Moreover, if (B.46) holds for any $t$, by (B.47), we have

$$\max\{\|x^t - x^{t+1}\|, \|y^t - y_+^t(z^t)\|, \|z^t - x^{t+1}\|\} \to 0. \tag{B.50}$$

Consequently, for any limit point $(\bar{x}, \bar{y})$ of $(x^t, y^t)$, there exists a $\bar{z}$ such that
$$\max\{\|\bar{x} - \bar{x}_+(\bar{y}, \bar{z})\|, \|\bar{y} - \bar{y}_+(\bar{z})\|, \|\bar{x}_+(\bar{y}, \bar{z}) - \bar{z}\|\} = 0,$$
which yields $(\bar{x}, \bar{y})$ is a stationary solution. Here
$$x_+(y, z) = P_X(x - \nabla_x K(x, z; y)).$$

$\blacksquare$

Now we are ready to prove Theorem 3.4.

**Proof** [Proof of Theorem 3.4] There are two cases (B.51) and (B.52) as discussed in the proof for the general nonconvex-concave problems in last subsection.

1. For some $t \in \{0, 1, \cdots, T-1\}$, we have
$$\frac{1}{2}\max\{\frac{1}{8c}\|x^t - x^{t+1}\|^2, \frac{1}{8\alpha}\|y^t - y^t_+(z^t)\|^2, \frac{p}{8\beta}\|z^t - z^{t+1}\|^2\} \le 24p\beta\|x^*(z^t) - x(y^t_+(z^t), z^t)\|^2.$$
(B.51)

2. For any $t \in \{0, 1, \cdots, T-1\}$,
$$\frac{1}{2}\max\{\frac{1}{8c}\|x^t - x^{t+1}\|^2, \frac{1}{8\alpha}\|y^t - y^t_+(z^t)\|^2, \frac{p}{8\beta}\|z^t - z^{t+1}\|^2\} \ge 24p\beta\|x^*(z^t) - x(y^t_+(z^t), z^t)\|^2$$
(B.52)

In the first case (B.51), we have
$$\begin{aligned}
\|y^t - y^t_+(z^t)\|^2 &\le 384p\beta\alpha\|x(y^t_+(z^t), z^t) - x^*(z^t)\|^2 \\
&\le 384p\beta\alpha\frac{(1+\alpha L)}{p-L}\|y^t - y^t_+(z^t)\|D(Y).
\end{aligned}$$

Hence, letting $\lambda_1 = 384p\alpha\frac{(1+\alpha L)}{p-L} \cdot D(Y)$, we have
$$\|y^t - y^t_+(z^t)\| \le \lambda_1 \beta.$$
(B.53)

Moreover,
$$\begin{aligned}
\|x^{t+1} - z^t\|^2 &= \|(z^{t+1} - z^t)/\beta\|^2 && \text{(B.54)} \\
&\overset{(i)}{\le} 384p\|x(y^t_+(z^t), z^t) - x^*(z^t)\|^2 && \text{(B.55)} \\
&\overset{(ii)}{\le} 384p\frac{1+\alpha L}{p-L}D(Y)\|y^t - y^t_+(z^t)\| && \text{(B.56)} \\
&\overset{(iii)}{\le} 384p\frac{1+\alpha L}{p-L}D(Y)\lambda_1\beta, && \text{(B.57)}
\end{aligned}$$

where Inequality (i) is due to Inequality (B.51) and (ii) is because of Lemma B.10 and (iii) is due to (B.53). We also have
$$\begin{aligned}
\|x^t - x^{t+1}\|^2 &\overset{(i)}{\le} 384cp\beta\|x^*(z^t) - x(y^t_+(z^t), z^t)\|^2 && \text{(B.58)} \\
&\overset{(ii)}{\le} 384pc\beta\frac{1+\alpha L}{p-L}D(Y)\|y^t - y^t_+(z^t)\| && \text{(B.59)} \\
&\overset{(iii)}{\le} 384pc\frac{1+\alpha L}{p-L}\lambda_1 D(Y)\beta^2, && \text{(B.60)}
\end{aligned}$$

where (i) is due to (B.51), (ii) is due to Lemma B.10 and (iii) is because of (B.53). Combining the above, in the first case, we have
$$\begin{aligned}
&\max\{\|x^t - x^{t+1}\|^2, \|y^t - y^t_+(z^t)\|^2, \|z^t - x^{t+1}\|^2\} && \text{(B.61)} \\
&\le \max\{\lambda_2\beta^2, \lambda_1^2\beta^2, \lambda_3\beta\}, && \text{(B.62)}
\end{aligned}$$

where $\lambda_2 = 384p\frac{1+\alpha L}{p-L}D(Y)\lambda_1$ and $\lambda_3 = 192pc\frac{1+\alpha L}{p-L}\lambda_1 D(Y)$ According to Lemma B.11, there exists a $\lambda > 0$ such that $(x^{t+1}, y^{t+1})$ is a $\lambda\max\{\beta, \sqrt{\beta}\}$-solution.

In the second case, we have

$$\phi^t - \phi^{t+1} \geq \frac{1}{16c}\|x^t - x^{t+1}\|^2 + \frac{1}{16\alpha}\|y^t - y_+^t(z^t)\|^2 + \frac{1}{16\beta}\|z^t - z^{t+1}\|^2$$

for any $t \in \{0, 1, \cdots, T-1\}$. By Lemma B.12, there exists a $t \in \{0, 1, \cdots, T-1\}$, such that $(x^{t+1}, y^{t+1})$ is a $\sqrt{(\phi^0 - \underline{f})/((1/8c + 1/8\alpha + 16)T\beta)}$-solution. Finally taking $\beta = 1/\sqrt{T}$ and combining the two cases with Lemma B.12 yield the desired results.

$\blacksquare$

## B.3 The max problem is over a discrete set

In this subsection, we prove Theorem 3.8. We will prove that under the strict complementarity assumption, the potential function $\phi^t$ decreases sufficiently after any iteration. Then by the following simple lemma, we can prove Theorem 3.8.

By the bounded level set assumption (Assumption 3.7) and the fact that $\psi(z) \leq P(z)$, for any $(x^0, y^0, z^0) \in \mathbb{R}^{n+m+n}$, there exists a constant $R(x^0, y^0, z^0) > 0$ such that

$$\{z \mid P(z) \leq \phi(x^0, y^0, z^0)\} \subseteq \mathcal{B}(R(x^0, y^0, z^0)).$$

Then we have the following "dual error bound". Note that this error bound is homogeneous compared to Lemma B.10.

**Lemma B.13** *Let*
$$x_+(y, z) = P_X(x - \nabla_x K(x, z; y)).$$
*If the strict complementarity assumption and the bounded level set assumption hold for* (1.2) *, there exists $\delta > 0$, such that if*
$$\|z\| \leq R(x^0, y^0, z^0),$$
*and*
$$\max\{\|x - x_+(y, z)\|, \|y - y_+(z)\|, \|x_+(y, z) - z\|\} < \delta$$
*we have*
$$\|x(y_+(z), z) - x^*(z)\| < \sigma_5\|y - y_+(z)\|$$
*for some constant $\sigma_5 > 0$.*

Equipped with the dual error bound, we can prove that the potential function decreases after any iteration in the following proposition:

**Proposition B.14** *Suppose the conditions in Theorem 3.8 holds, we have*

$$\phi^t - \phi^{t+1} \geq \frac{1}{16c}\|x^t - x^{t+1}\|^2 + \frac{1}{16\alpha}\|y^t - y_+^t(z^t)\|^2 + \frac{p}{16\beta}\|z^t - z^{t+1}\|^2. \tag{B.63}$$

**Proof** We set $\beta < \min\{\delta/\sqrt{\lambda_2}, \delta/\lambda_1, \delta^2/\lambda_3, 1/(384p\alpha\sigma_5^2)\}$. First, we prove that

$$\phi^t - \phi^{t+1} \geq \frac{1}{16c}\|x^t - x^{t+1}\|^2 + \frac{1}{16\alpha}\|y^t - y_+^t(z^t)\|^2 + \frac{p}{16\beta}\|z^t - z^{t+1}\|^2. \tag{B.64}$$

and
$$\|z^t\| < R(x^0, y^0, z^0)$$
for any $t \geq 0$. We prove it by induction. We will prove that

1. If $\|z^t\| \leq R(x^0, y^0, z^0)$, then

   $$\phi^t - \phi^{t+1} \geq \frac{1}{16c}\|x^t - x^{t+1}\|^2 + \frac{1}{16\alpha}\|y^t - y_+^t(z^t)\|^2 + \frac{p}{16\beta}\|z^t - z^{t+1}\|^2. \tag{B.65}$$

2. If $\phi^{t+1} \leq \phi^t$, we have $\|z^{t+1}\| \leq R(x^0, y^0, z^0)$.

For $t = 0$, it is trivial that $\|z^t\| \leq R(x^0, y^0, z^0)$. For the first step, assume that we have $\|z^t\| \leq R(x^0, y^0, z^0)$. There are two cases:

1. For some $t$, we have

$$\frac{1}{2}\max\{\frac{1}{8c}\|x^t-x^{t+1}\|^2, \frac{1}{8\alpha}\|y-y_+^t(z^t)\|^2, \frac{p}{8\beta}\|z^t-z^{t+1}\|^2\} \le 24p\beta\|x^*(z^t)-x(y_+^t(z^t), z^t)\|^2.$$

(B.66)

2. For any $t$,

$$\frac{1}{2}\max\{\frac{1}{8c}\|x^t-x^{t+1}\|^2, \frac{1}{8\alpha}\|y-y_+^t(z^t)\|^2, \frac{p}{8\beta}\|z^t-z^{t+1}\|^2\} \ge 24p\beta\|x^*(z^t)-x(y_+^t(z^t), z^t)\|^2$$

(B.67)

For the first case, as in the last subsection, we have

$$\begin{aligned}
& \max\{\|x^t - x^{t+1}\|^2, \|y^t - y_+^t(z^t)\|^2, \|x^{t+1} - z^t\|^2\} \\
\le \quad & \max\{\lambda_2\beta^2, \lambda_1^2\beta^2, \lambda_3\beta\} \\
\le \quad & \delta^2.
\end{aligned}$$

Hence, we can make use of Lemma B.13. In fact, we have

$$\begin{aligned}
24p\beta\|x(y_+^t(z^t), z^t) - x^*(z^t)\|^2 & \le \quad 24p\beta\sigma_5^2\|y^t - y_+^t(z^t)\| \\
& \le \quad \frac{1}{16\alpha}\|y^t - y_+^t(z^t)\|^2,
\end{aligned}$$

which yields (B.46) together with (B.15). For the second case, (B.46) holds as in the last subsection. Hence, if $\|z^t\| \le R(x^0, y^0, z^0)$, we have (B.46). For the second step, if (B.46) holds for $0, 1, \cdots, (t-1)$, we have

$$\begin{aligned}
P(z^{t+1}) & \le \quad \phi^{t+1} \\
& \le \quad \phi^0.
\end{aligned}$$

Hence, $z^{t+1} \in \mathcal{B}(R(x^0, y^0, z^0))$. Combining these, for any $t \ge 0$, $\|z^t\| \le R(x^0, y^0, z^0)$ and (B.46) holds. Then the theorem comes from Lemma B.12. ∎

# C  The multi-block cases

The proofs for the multi-block case is similar to the one-block case. In this section, we briefly introduce the proof of them. Note that the only differences for proving the theorem s are Lemma B.5, (B.5) and Proposition 4.1. Instead, we have the following:

**Lemma C.1 (Primal Descent)** *For any t, we have*

$$\begin{aligned}
K(x^t, z^t; y^t) - K(x^{t+1}, z^{t+1}; y^{t+1}) & \ge \quad \frac{1}{2c}\|x^t - x^{t+1}\|^2 + \langle \nabla_y K(x^{t+1}, z^t; y^t), y^t - y^{t+1}\rangle \\
& \quad -\frac{L}{2}\|y^t - y^{t+1}\|^2 + \frac{p}{2\beta}\|z^t - z^{t+1}\|^2.
\end{aligned}$$ (C.1)

The proof of it is the same as Lemma 5.3 in [30]. The error bound (B.5) becomes:

**Lemma C.2** *We have*
$$\|x^{t+1} - x(y^t, z^t)\| \le \sigma_3'\|x^t - x^{t+1}\|,$$
*where $\sigma_3' = (c(p - L) + 1 + c(L + p)N^{3/2})/c(p - L)$.*

The proof of Lemma C.2 is similar to Lemma 5.2 in [30] hence omitted here. Because of the above two differences, we have a replacement of Proposition B.4:

**Proposition C.3** *We let*

$$p > 3L, c < \frac{1}{p + L}, \alpha < \min\{\frac{1}{11L}, \frac{c^2(p - L)^2}{4L(1 + c(p + L)N^{3/2} + c(p - L))^2}\}, \min\{\frac{1}{36}, \beta < \frac{(p - L)^2}{384p(p + L)^2}\}.$$

(C.2)

*Then we have*

$$\phi^t - \phi^{t+1} \tag{C.3}$$

$$\geq \quad \frac{1}{4c}\|x^t - x^{t+1}\|^2$$

$$+\frac{1}{4\alpha}\|y^t - y_+^t(z^t)\|^2 + \frac{p}{8\beta}\|z^t - z^{t+1}\|^2 \tag{C.4}$$

$$-24p\beta\|x^*(z^t) - x(y_+^t(z^t), z^t)\|^2 \tag{C.5}$$

The proof of Proposition C.3 is similar to Proposition B.4 hence omitted.

# D  Proof of the error bound lemmas

## D.1  Proof of lemma B.10

Let

$$x_+(y, z) = P_X(x - c\nabla_x K(x, z; y))$$

and

$$y_+(z) = P_Y(y + \alpha\nabla_y K(x(y, z), z; y)).$$

Then Lemma B.10 can be written as

**Lemma D.1** *We have*

$$(p - L)\|x^*(z) - x(y_+(z), z)\|^2 \quad < \quad (1 + \alpha L)\|y - y_+(z)\| \cdot \text{dist}(y_+(z), Y(z)).$$
$$\leq \quad (1 + \alpha L)\|y - y_+(z)\| \cdot D(Y),$$

*where $D(Y)$ is the diameter of $Y$.*

**Proof**  By the strong convexity of $K(\cdot, z; y)$, we have

$$K(x^*(z), z, ; y_+(z)) - K(x(y_+(z), z)z; y_+(z)) \quad \geq \quad \frac{p - L}{2}\|x(y_+(z), z) - x^*(z)\|^2 \tag{D.1}$$

$$K(x(y_+(z), z), z; y(z)) - K(x^*(z), z; y(z)) \quad \geq \quad \frac{p - L}{2}\|x(y_+(z), z) - x^*(z)\|^2, \tag{D.2}$$

where $y(z)$ is an arbitrary vector in $Y(z)$. Notice that $y_+(z)$ is the maximizer of the following problem:

$$\max_{\bar{y}\in Y}\{K(x(y_+(z), z), z; \bar{y}) - \delta^T(y, y_+(z); z)\bar{y}\},$$

where

$$\delta(y, y_+(z); z) = (y_+(z) + \alpha\nabla_{\bar{y}}K(x(y_+(z), z), z; y_+(z))) - (y + \alpha_{\bar{y}}K(x(y_+(z), z), z; y))$$

satisfies

$$\|\delta(y, y_+(z); z)\| < (1 + \alpha L)\|y - y_+(z)\|,$$

by the Lipschitz-continuity of $\nabla_y K = \nabla_y f$. Hence, we have

$$K(x(y_+(z), z), z; y(z)) - \delta^T(y, y_+(z); z)y(z)$$
$$\leq \quad K(x(y_+(z), z), z; y_+(z)) - \delta^T(y, y_+(z); z)y_+(z).$$

Then, we have the following estimates:

$$K(x(y_+(z), z), z; y(z)) - K(x(y_+(z), z), z; y_+(z)) \tag{D.3}$$

$$\leq \quad (y(z) - y_+(z))^T\delta(y, y_+(z); z) \tag{D.4}$$

$$\leq \quad \|y_+(z) - y(z)\| \cdot (1 + \alpha L)\|y - y_+(z)\|. \tag{D.5}$$

Also because $y(z)$ maximizes

$$\max_{\bar{y}\in Y} K(x^*(z), \bar{y}; z),$$

we have

$$K(x^*(z), z; y(z)) \geq K(x^*(z), z; y_+(z)). \tag{D.6}$$

Since $y(z)$ is an arbitrary vector in $Y(z)$, combining (D.1), (D.3), (D.6), we have

$$(p - L)\|x^*(z) - x(y_+(z), z)\|^2 < (1 + \alpha L)\|y - y_+(z)\| \cdot \text{dist}(y_+(z), Y(z)),$$

which is the desired result.  ∎

## D.2 Proof of Lemma B.13

For a pair of min-max solution of (1.2), the KKT conditions in the following hold:

$$J^T F(x^*)y = 0, \tag{D.7}$$

$$\sum_{i=1}^{m} y_i = 1, \tag{D.8}$$

$$y_i \geq 0, \forall i \in [m] \tag{D.9}$$

$$\mu - \nu_i = f_i(x), \forall i \in [m], \tag{D.10}$$

$$\nu_i \geq 0, \nu_i y_i = 0, \forall i \in [m], \tag{D.11}$$

where $\mu$ is the multiplier of the equality constraint $\sum_{i=1}^{m} y_i = 1$ and $\nu_i$ is the multiplier for the inequality constraint $y_i \geq 0$.

**Definition D.2** *For $y \in Y$, we define the active set*

$$\mathcal{A}[y] = \{i \in [m] \mid y_i = 0\}.$$

*We also define the inactive set of $y$ as follows:*

$$\mathcal{I}[y] = \{i \in [m] \mid y_i > 0\}.$$

**Definition D.3** *For an $x \in \mathbb{R}^n$, we define the top coordinate set $\mathcal{T}(x)$ as the collection of all indexes of the top coordinates of $F(x)$, i.e., $f_i(x) > f_j(x)$ if $i \in \mathcal{T}(x), j \notin \mathcal{T}(x)$ and $f_i(x) = f_j(x)$ if $i, j \in \mathcal{T}(x)$.*

According to the KKT conditions, it is easy to see that for $(x, y) \in W^*$,

$$\mathcal{I}[y] \subseteq \mathcal{T}(x).$$

Recall that we have the following strict complementarity condition:

**Assumption D.4** *For any $(x, y)$ satisfying* (D.7)*, we have*

$$\nu_i > 0, \forall i \in \mathcal{A}[y].$$

It is easy to see that if the strict complementarity assumption holds,

$$\mathcal{I}[y] = \mathcal{T}(x)$$

for $(x, y) \in W^*$. Then we can prove the following "dual error bound".

**Lemma D.5** *If the strict complementarity assumption holds for* (1.2) *, there exists $\delta > 0$, such that if*

$$\|z\| \leq R(x^0, y^0, z^0),$$

*and*

$$\max\{\|x - x_+(y, z)\|, \|y - y_+(z)\|, \|x_+(y, z) - z\|\} < \delta$$

*we have*

$$\|x(y_+(z), z) - x^*(z)\| < \sigma_5 \|y - y_+(z)\|$$

*for some constant $\sigma_5 > 0$.*

To prove this, we need the following lemmas. First, we prove that if the residuals go to zero, the iteration points converge to a solution.

**Lemma D.6** *If $\{z^k\}$ is a sequence with $\|z^k\| \leq R(x^0, y^0, z^0)$ and*

$$\max\{\|x^k - x_+^k(y^k, z^k)\|, \|y^k - y_+^k(z^k)\|, \|x_+^k(y^k, z^k) - z^k\|\} \to 0,$$

*there exists a sub-sequence of $\{z^k\}$ converging to some $\bar{z} \in X^*$.*

**Proof** It is just a direct corollary of Lemma B.11. ∎

**Lemma D.7** *Let*

$$M(x) = \left\{ J_{\mathcal{T}(x)} F(x) \quad \mathbf{1} \right\}.$$

*Then if $(x, y) \in W^*$, the matrix $M(x)$ is of full row rank.*

**Proof** We prove it by contradiction. If for some $(x^*, y^*, \mu^*, \nu^*)$ satisfying (D.7), $M(x^*)$ is not of full row rank. Without loss of generality, we assume that $\mathcal{T}(x^*) = \{1, 2, \cdots, |\mathcal{T}(x^*)|\}$ Then there exists a nonzero vector $v \in \mathbb{R}^{|\mathcal{T}(x^*)|}$ such that

$$M^T(x^*)v = 0.$$

Let $d = \min_{i \in \mathcal{I}[y^*]} \{y_i / |v_i|\}$. Then we define a vector $y' \in \mathbb{R}^m$ as:

$$y'_i = y^* - dv_i, \quad when \quad i \in \mathcal{I}[y^*];$$
$$y'_i = 0, \quad when \quad otherwise.$$

Notice that $y^*_i = 0$ for any $i \notin \mathcal{T}(x^*)$. Then $y'$ satisfies

$$J^T F(x^*) y' = 0,$$
$$\sum_{i=1}^m y'_i = 1,$$
$$y'_i \geq 0, i \in \mathcal{T}(x^*),$$
$$y'_i = 0, i \notin \mathcal{T}(x^*).$$

Therefore, $(x^*, y', \mu, \nu)$ still satisfies (D.7) . Moreover, let $i_0 \in \mathcal{I}[y^*]$ satisfying $d = y^*_{i_0} / v_{i_0}$. Then $y'_{i_0} = \nu_{i_0} = 0$. This is a contradiction to the strict complementarity assumption. ∎

We then have the following corollary from the above lemma and (D.7):

**Corollary D.8** *For any $x^* \in X^*$, there exists only one $y \in Y$ such that $(x^*, y) \in W^*$ and there exists only one $(\mu, \nu)$ such that $(x^*, y, \mu, \nu)$ satisfies (D.7).*

**Proof** First, this $(y, \mu, \nu)$ must exist due to the existence of a solution. Next, the solution $y$ must satisfy

$$M^T(x^*)y = (0, 0, \cdots, 0, 1)^T.$$

By Lemma D.7, $M^T(x^*)$ is of full column rank hence the solution of $y$ is unique. Furthermore, since $\sum_{i=1}^m y_i = 1$, there is at least one $i$ such that $y_i > 0, \nu_i = 0$. Without loss of generality, we assume that $y_1 > 0, \nu_1 = 0$. Then $\mu = f_1(x^*)$ by (3.5). Further by (3.5), $\nu_i = f_i(x^*) - \mu, i = 2, 3, \cdots, m$. Hence, $\mu, \nu_i$ are uniquely defined. ∎

**Lemma D.9** *If the strict complementarity assumption holds for (1.2) , there exists $\delta > 0, \gamma > 0$, such that if*

$$\|z\| \leq R(x^0, y^0, z^0),$$

*and*

$$\max\{\|x - x_+(y, z)\|, \|y - y_+(z)\|, \|x_+(y, z) - z\|\} < \delta$$

$\gamma(M(x^*(z))) \geq \gamma$ *and* $\gamma(M(x(y, z))) \geq \gamma$.

**Proof** We prove it by contradiction. Suppose it is not true, there exists $\{z^k\} \subseteq \mathcal{B}(R(x^0, y^0, z^0))$ such that $\gamma(M(x^*(z^k))) \to 0$ and

$$\max\{\|x^k - x^k_+(y^k, z^k)\|, \|y^k - y^k_+(z^k)\|, \|x^k_+(y^k, z^k) - z^k\|\} \to 0.$$

Since $\mathcal{T}(x)$ has only finite choice, without loss of generality, we assume that $\mathcal{T}(x^*(z^k)) = \mathcal{T}$ for any $k$(passing to a sub-sequence if necessary). By Lemma D.6, there exists a $\bar{z} \in X^*$ such that $z^k \to \bar{z}$. We let

$$\tilde{M}(\bar{z}) = \lim_{k \to \infty} M(x^*(z^k)) = \left\{ J_{\mathcal{T}}(x^*(\bar{z})) \quad \mathbf{1} \right\}.$$

By the continuity of $x^*(\cdot)$((B.3) of Lemma B.2) and the continuity of the function of taking the least singular value, we know that

$$\gamma(\tilde{M}(\bar{z})) = 0,$$

where we also use the fact that $x^*(\bar{z}) = \bar{z}$ by Lemma B.8. Moreover, according to the definition of $\mathcal{T}[\bar{z}]$, we have $f_i(x^*(z^k)) > f_j(x^*(z^k))$ for any $k$ with $i \in \mathcal{T}, j \notin \mathcal{T}$. Therefore, we have $f_i(\bar{z}) \geq f_j(\bar{z})$ for $i \in \mathcal{T}, j \notin \mathcal{T}$. Consequently, we have

$$\mathcal{T} \subseteq \mathcal{T}[\bar{z}].$$

Therefore $\tilde{M}(x^*(\bar{z}))$ is a row sub-matrix of $M(\bar{z})$. Consequently, $M(\bar{z})$ is not of full row rank. This is a contradiction! For $x(y, z)$, it is similar to prove the desired result. Hence the details are omitted. ∎

The following lemma shows that if the residuals are small, the active set of $y_+(z)$ and $y(z) \in Y(z)$ are the same.

**Lemma D.10** *If the strict complementarity assumption holds for* (1.2) *, there exists $\delta > 0$, such that if*

$$\|z\| \leq R(x^0, y^0, z^0),$$

*and*

$$\max\{\|x - x_+(y, z)\|, \|y - y_+(z)\|, \|x_+(y, z) - z\|\} < \delta,$$

*we have*

$$\mathcal{A}[y_+(z)] = \mathcal{A}[y(z)], \text{ for some } y(z) \in Y(z).$$

**Proof** We prove it by contradiction. Suppose that there exists a sequence $\{(x^k, y^k, z^k)\}$ such that

$$\max\{\|x^k - x_+^k(y^k, z^k)\|, \|y^k - y_+^k(z^k)\|, \|x_+^k(y^k, z^k) - z^k\|\} \to 0$$

and

$$\mathcal{A}[y_+^k(z^k)] \neq \mathcal{A}[y(z^k)].$$

Since $\{y_+^k(z^k)\}, \{z^k\}$ are bounded, we assume that $y_+^k(z^k) \to \bar{y}, z^k \to \bar{z}$. We write down the KKT condition for $(x(y_+^k(z^k), z^k), y_+^k(z^k))$ as follows:

$$J^T F(x(y_+^k(z^k), z^k))y_+^k(z^k) + p(x(y_+^k(z^k), z^k) - z^k) = 0, \tag{D.12}$$

$$\sum_{i=1}^m (y_i^k)_+(z^k) = 1, \tag{D.13}$$

$$(y_i^k)_+(z^k) \geq 0, \forall i \in [m] \tag{D.14}$$

$$\frac{1}{\alpha}(y_i^k)_+(z^k) - \frac{1}{\alpha}y_i^k + f_i(x(y^k, z^k)) + \mu^k - \nu_i^k = f_i(x(y_+^k(z^k), z^k)), \forall i \in [m], \tag{D.15}$$

$$\nu_i^k \geq 0, \nu_i^k(y_i^k)_+(z^k) = 0, \forall i \in [m], \tag{D.16}$$

It is not hard to check that $\mu, \nu$ are bounded. Hence, we assume that $\mu^k \to \bar{\mu}$ and $\nu^k \to \bar{\nu}$. We take limit to (D.12) and make use of the fact that

$$\|y^k - y_+^k(z^k)\| \to 0$$

together with Lemma B.2. We then attain that $(x(\bar{y}, \bar{z}), \bar{y})$ is a min-max solution of (1.2), i.e., $(x(\bar{y}, barz), \bar{y}, \bar{\mu}, \bar{\nu})$ satisfies (D.7). By the strict complementarity assumption, $\bar{\nu}_i > 0$ for $i \in \mathcal{A}[\bar{y}]$ and $\bar{y}_i > 0$ for $i \notin \mathcal{A}[\bar{y}]$. Hence, for $k$ sufficiently large, we have $\mathcal{A}[y_+^k(z^k)] = \mathcal{A}[\bar{y}]$. Similarly, when $k$ is sufficiently large, we have

$$\mathcal{A}[y(z^k)] = \mathcal{A}[\bar{y}].$$

∎

We also write down the KKT conditions for $x^*(z)$ for some $z$.

$$J^T F(x^*(z))y + p(x^*(z) - z) = 0, \qquad (\text{D.17})$$

$$\sum_{i=1}^{m} y_i = 1, \qquad (\text{D.18})$$

$$y_i \geq 0, \forall i \in [m] \qquad (\text{D.19})$$

$$\mu - \nu_i = f_i(x), \forall i \in [m], \qquad (\text{D.20})$$

$$\nu_i \geq 0, \nu_i y_i = 0, \forall i \in [m], \qquad (\text{D.21})$$

**Lemma D.11** *If the strict complementarity assumption holds for* (1.2)*, there exists $\delta > 0$, such that if*

$$\|z\| \leq R(x^0, y^0, z^0),$$

*and*

$$\max\{\|x - x_+(y,z)\|, \|y - y_+(z)\|, \|x_+(y,z) - z\|\} < \delta$$

*we have*

$$\text{dist}(y_+(z), y(z)) < \lambda \|x^*(z) - x(y_+(z), z)\|$$

*for some constant $\lambda > 0$.*

**Proof** By Lemma D.10 , if the strict complementarity assumption holds for (1.2) , there exists $\delta > 0$, such that if

$$\|z\| \leq R(x^0, y^0, z^0),$$

and

$$\max\{\|x - x_+(y,z)\|, \|y - y_+(z)\|, \|x_+(y,z) - z\|\} < \delta,$$

we have

$$\mathcal{A}[y_+(z)] = \mathcal{A}[y(z)],$$

for some $y(z) \in Y(z)$. Hence, we have

$$\mathcal{T}(x^*(z)) = \mathcal{T}(x(y_+(z), z)).$$

Let $\mathcal{T} = \mathcal{T}(x^*(z))$. Then for $i \notin \mathcal{T}$, $y_i(z) = (y_+(z))_i = 0$ and $\|y(z) - y_+(z)\| = \|(y(z))_{\mathcal{T}} - (y_+(z))_{\mathcal{T}}\|$. Using the optimality conditions for $x(y_+(z), z)$ (D.12) and $x^*(z)$ (D.17), we have

$$M^T(x(y_+(z), z))(y_+(z))_{\mathcal{T}} + \left\{\begin{matrix} p(x(y_+(z), z) - z) \\ 0 \end{matrix}\right\} = (0, 0, \cdots, 0, 1), \qquad (\text{D.22})$$

and

$$M^T(x^*(z))(y(z))_{\mathcal{T}} + \left\{\begin{matrix} p(x^*(z) - z) \\ 0 \end{matrix}\right\} = (0, 0, \cdots, 0, 1). \qquad (\text{D.23})$$

Note that (D.22) can be written as

$$M^T(x^*(z))(y_+(z))_{\mathcal{T}} = M^T(x^*(z))(y_+(z))_{\mathcal{T}} - M^T(x(y_+(z), z))(y_+(z))_{\mathcal{T}} - \left\{\begin{matrix} p(x(y_+(z), z) - z) \\ 0 \end{matrix}\right\}. \qquad (\text{D.24})$$

By (D.23) and (D.24), we have

$$M^T(x^*(z))((y(z))_{\mathcal{T}} - (y_+(z))_{\mathcal{T}}) = (M^T(x(y_+(z), z)) - M^T(x^*(z)))(y_+(z))_{\mathcal{T}} - \left\{\begin{matrix} p(x(y_+(z), z) - x^*(z)) \\ 0 \end{matrix}\right\}.$$

Therefore, taking norms to the above and the Lemma D.9, we have

$$\begin{aligned} \gamma\|(y_+(z))_{\mathcal{T}} - (y(z))_{\mathcal{T}}\| &\leq \sqrt{m}L\|x(y_+(z), z) - x^*(z)\|\|(y_+(z))_{\mathcal{T}}\| + p\|x(y_+(z), z) - x^*(z)\| \\ &\leq (\sqrt{m}L + p)\|x^*(z) - x(y_+(z), z)\|, \end{aligned}$$

where the first inequality uses the Lipschitz-continuity of $\nabla_x f_i$ and the second is because $\|y_+(z)\| \leq 1$. Hence, we finish the proof with $\lambda = (p + \sqrt{m}L)/\gamma$. ∎

**Proof** [Proof of Lemma B.13] By Lemma B.10 and Lemma D.11, we have

$$\|x(y_+(z), z) - x^*(z)\| \leq \frac{1 + \alpha L}{\lambda(p - L)}\|y - y_+(z)\|,$$

which finishes the proof with $\sigma_5 = \frac{1+\alpha L}{\lambda(p-L)}$. ∎

# E   Discussion of the strict complementarity condition

In this section, we discuss some issues about the strict complementarity assumption. First, notice that the min-max problem (1.1) and (1.2) are both variational inequalities. As mentioned in the main text of the paper, the strict complementarity assumption is common in the field of variation inequality [39, 40]. While this assumption is popular, it is still interesting to weaken the assumption. Inspired by Lemma D.7, we prove Theorem 3.8 and Lemma 4.2 using a weaker regularity assumption rather than the strict complementarity assumption:

**Assumption E.1** *For any $(x^*, y^*) \in W^*$, the matrix $M(x^*)$ is of full column rank.*

Here recall that

$$M(x^*) = \left\{ J_{\mathcal{T}(x^*)} \quad \mathbf{1} \right\}.$$

We say that Assumption E.1 is weaker since the strict complementarity assumption (Assumption D.4) can imply Assumption E.1 according to Lemma D.7. For this assumption, we have the following two claims:

1. If we replace Assumption D.4 by Assumption E.1 in Theorem 3.8, we can attain a same result;

2. In a robust regression problem (will define in E.2), if the data is joint from a continuous distribution, this regularity assumption holds with probability 1.

## E.1   Replacing Assumption D.4 by Assumption E.1 in Theorem 3.8

In this section, we will see that we can prove the dual error bound (Lemma 4.2) using Assumption E.1 instead of Assumption D.4.

**Lemma E.2** *Let*

$$x_+(y, z) = P_X(x - \nabla_x K(x, z; y)).$$

*If Assumption E.1 and the bounded level set assumption hold for* (1.2) *, there exists $\delta > 0$, such that if*

$$\|z\| \le R(x^0, y^0, z^0),$$

*and*

$$\max\{\|x - x_+(y, z)\|, \|y - y_+(z)\|, \|x_+(y, z) - z\|\} < \delta$$

*we have*

$$\|x(y_+(z), z) - x^*(z)\| < \sigma_5 \|y - y_+(z)\|$$

*for some constant $\sigma_5 > 0$.*

Using this Lemma, we can prove Theorem 3.8 using Assumption E.1:

**Theorem E.3** *Consider solving problem 1.2 by Algorithm 2 or Algorithm 3. Suppose that Assumption E.1 holds and either Assumption 3.7 holds or assume $\{z^t\}$ is bounded. Then there exist constants $\beta'$ and $\beta''$ (independent of $\epsilon$ and $T$) such that the following holds:*

1. *(One-block case) If we choose the parameters in Algorithm 2 as in* (3.3) *and further let $\beta < \beta'$ , then*

    (a) *Every limit point of $(x^t, y^t)$ is a solution of* (1.2).
    (b) *The iteration complexity of Algorithm 2 to obtain an $\epsilon$-stationary solution is $\mathcal{O}(1/\epsilon^2)$.*

2. *(Multi-block case) Consider using Algorithm 3 to solve Problem 1.2. If we replace the condition for $\alpha$ in* (3.3) *by* (3.4) *and require $\beta$ satisfying $\beta < \epsilon^2$ and $\beta < \beta''$, then we have the same results as in the one-block case.*

## E.2 The rationality of Assumption E.1

Intuitively, the assumption E.1 holds for "generic problem". We rigorously justify this intuition for a simple problem. More specifically, we prove that this regularity assumption is generic for a robust regression problem using square loss, i.e., the regularity condition holds with probability 1 if the outputs of the data points are joint from some continuous distribution. Consider the following problem:

$$\min_{x\in\mathbb{R}^n}\max_{y\in Y}\frac{1}{2}y_i(\ell_i-\Psi(x,\xi_i))^2, \tag{E.1}$$

where $Y$ is the probability simplex , $\Psi(\cdot)$ is a smooth function used to fit the data (for example the neural network) and $\xi_i, \ell_i$ are the input and the output of the $i$-th data point. We define $\Psi_i(x) = \Psi(x,\xi_i)$ for convenience. We further make the following mild assumptions:

**Assumption E.4** $\ell_i$ *is joint independently from a continuous distribution over a positive measure set* $\mathcal{L}_i \subseteq \mathbb{R}$.

Here a continuous distribution over $\mathcal{L}_i$ means that for any zero measure set $\mathcal{S} \subseteq \mathcal{R}$, $\Pr(x \in \mathcal{S} \cap \mathcal{L}_i) = 0$. With assumption, for any zero measure set $\mathcal{S} \subseteq \mathbb{R}^m$, $\Pr((\ell_1, \cdots, \ell_m)^T \in \mathcal{S}\cap\prod_{i=1}^{m}\mathcal{L}_i) = 0$.

**Assumption E.5** *Let* $\Psi(x) = (\Psi_1(x), \cdots, \Psi_m(x))^T$. *Then* $\Psi(\mathbb{R}^n) \cap \prod_{i=1}^{m}\mathcal{L}_i = \Omega$, *where* $\Omega$ *is a zero measure set in* $\prod_{i=1}^{m}\mathcal{L}_i$.

This assumption means that $\min_x \max_{y\in Y} f_i(x) > 0$ with probability 1. This assumption is reasonable. If there exists an $x^*$ such that $\max_i f_i(x^*) = 0$, then becaus $f_i(x) \geq 0$, we have $f_i(x^*) = 0$ for all $i$. In this case, we do not need the min-max fomulation! We just need to solve the finite sum problem $\min_x \sum_{i=1}^{m} f_i(x)$. However, in many cases, the uncertainty is large, we do need the robust optimization formulation. So in these cases, Assumption E.5 is reasonable.

Moreover, we have the following lemma:

**Lemma E.6** *Suppose that Assumption E.4 holds. If* $m > n$, *Assumption E.5 holds with probability* 1.

**Proof** It is direct from the claim that a smooth map $\Psi$ maps a zero measure set into a zero measure set. Specializing to this lemma, the map $\Psi$ maps $\mathbb{R}^n$ into $\mathbb{R}^m$, hence the image $\Psi(\mathbb{R}^n)$ is of zero measure since $\mathbb{R}^n$ is a zero measure set of $\mathbb{R}^m$. Therefore, $\Psi(\mathbb{R}^n)\cap\prod_{i=1}^{m}\mathcal{L}_i$ is zero measure in $\mathbb{R}^m$. ∎

Then we have the following result:

**Proposition E.7** *Suppose that Assumption E.4 and Assumption E.5 hold. Then with probability* 1, *every solution of* (E.1) *satisfies Assumption E.1.*

## E.3 Proof of Lemma E.2 and Theorem E.3

For a set $\mathcal{S} \subseteq [m]$, we define

$$M_{\mathcal{S}}(x) = \left\{ J_{\mathcal{S}F(x;\ell_{\mathcal{S}})} \quad \mathbf{1} \right\},$$

where $J_{\mathcal{S}}F(x;\ell_{\mathcal{S}}) = ((\Psi_i(x) - \ell_i)\nabla_x\Psi_i(x) \mid i \in \mathcal{S})$.

Similar to the proof of Theorem 3.8, to prove Theorem E.3, we only to prove Lemma E.2. Hence, in this section, we only prove Lemma E.2. The proof is similar to the proof of Lemma 4.2. Hence we only give the main steps. First, similar to Lemma D.9, we have the following:

**Lemma E.8** *If Assumption E.1 holds for Problem* (E.1) *, there exists* $\delta > 0, \gamma > 0$, *such that if*

$$\|z\| \leq R(x^0, y^0, z^0),$$

*and*

$$\max\{\|x - x_+(y,z)\|, \|y - y_+(z)\|, \|x_+(y,z) - z\|\} < \delta,$$

*then* $\gamma(M_{\mathcal{T}(y,z)}(x^*(z))) \geq \gamma$ *and* $\gamma(M_{\mathcal{T}(y,z)}(x(y_+(z),z))) \geq \gamma$, *where*

$$\mathcal{T}(y,z) = \mathcal{T}(x^*(z)) \cup \mathcal{T}(x(y_+(z),z)).$$

**Proof** We prove it by contradiction. Suppose it is not true, there exist $\{x^k\}$, $\{y^k\} \subseteq Y$ and $\{z^k\} \subseteq \mathcal{B}(R(x^0, y^0, z^0))$ such that $\gamma(M_{\mathcal{T}^k}(x^*(z^k)))$, $\gamma(M_{\mathcal{T}^k}(x(y^k_+(z^k), z^k))) \to 0$ and

$$\max\{\|x^k - x^k_+(y^k, z^k)\|, \|y^k - y^k_+(z^k)\|, \|x^k_+(y^k, z^k) - z^k\|\} \to 0,$$

where $\mathcal{T}^k = \mathcal{T}(x^*(z^k)) \cup \mathcal{T}(x(y^k_+(z^k), z^k))$. Since $\mathcal{T}^k$ has only finite choice, without loss of generality, we assume that $\mathcal{T}^k = \mathcal{T}$ for any $k$(passing to a sub-sequence if necessary). By Lemma D.6, there exists a $\bar{z} \in X^*$ such that $z^k \to \bar{z}$. Hence, by Lemma B.2 and Lemma B.8, we have

$$x^*(z^k) \to x^*(\bar{z}) = \bar{z}.$$

Therefore by the definition of $\mathcal{T}(x)$, when $k$ is sufficiently large, $\mathcal{T}(x^*(z^k)) \subseteq \mathcal{T}(x^*(\bar{z})) = \mathcal{T}(\bar{z})$. Moreover, since $\|y^k - y^k_+(z^k)\| \to 0$, by Lemma B.10, we have

$$\|x(\bar{y}^k_+(z^k), z^k) - x^*(z^k)\| \to 0.$$

and hence $\mathcal{T}(x(y^k_+(z^k), z^k)) \subseteq \mathcal{T}(\bar{z})$. Then $\mathcal{T}^k \subseteq \mathcal{T}(\bar{z})$ and $\gamma(M)_{\mathcal{T}^k} = 0$, which contradicts Assumption E.1. ∎

We then can attain a result similar to Lemma D.11.

**Lemma E.9** *If Assumption E.1 holds for* (1.2) *, there exists $\delta > 0$, such that if*
$$\|z\| \leq R(x^0, y^0, z^0),$$

*and*
$$\max\{\|x - x_+(y, z)\|, \|y - y_+(z)\|, \|x_+(y, z) - z\|\} < \delta$$
*we have*
$$\text{dist}(y_+(z), y(z)) < \lambda \|x^*(z) - x(y_+(z), z)\|$$
*for some constant $\lambda > 0$.*

**Proof** By Lemma E.8, we can find a $\delta > 0$ and a $\gamma > 0$, such that if
$$\|z\| \leq R(x^0, y^0, z^0),$$

and
$$\max\{\|x - x_+(y, z)\|, \|y - y_+(z)\|, \|x_+(y, z) - z\|\} < \delta,$$
then $\gamma(M_{\mathcal{T}(y,z)}(x^*(z))) \geq \gamma$ and $\gamma(M_{\mathcal{T}(y,z)}(x(y_+(z), z))) \geq \gamma$, where
$$\mathcal{T}(y, z) = \mathcal{T}(x^*(z)) \cup \mathcal{T}(x(y_+(z), z)).$$
Let $\mathcal{T} = \mathcal{T}(y, z)$. Then for $i \notin \mathcal{T}$, $y_i(z) = (y_+(z))_i = 0$ and $\|y(z) - y_+(z)\| = \|(y(z))_\mathcal{T} - (y_+(z))_\mathcal{T}\|$. Using the optimality conditions for $x(y_+(z), z)$ (D.12) and $x^*(z)$ (D.17), we have

$$M^T_\mathcal{T}(x(y_+(z), z))(y_+(z))_\mathcal{T} + \left\{ \begin{matrix} p(x(y_+(z), z) - z) \\ 0 \end{matrix} \right\} = (0, 0, \cdots, 0, 1), \qquad (E.2)$$

and

$$M^T_\mathcal{T}(x^*(z))(y(z))_\mathcal{T} + \left\{ \begin{matrix} p(x^*(z) - z) \\ 0 \end{matrix} \right\} = (0, 0, \cdots, 0, 1). \qquad (E.3)$$

Note that (E.2) can be written as

$$M^T_\mathcal{T}(x^*(z))(y_+(z))_\mathcal{T} = M^T_\mathcal{T}(x^*(z))(y_+(z))_\mathcal{T} - M^T_\mathcal{T}(x(y_+(z), z))(y_+(z))_\mathcal{T} - \left\{ \begin{matrix} p(x(y_+(z), z) - z) \\ 0 \end{matrix} \right\}. \qquad (E.4)$$

By (E.3) and (E.4), we have

$$M^T_\mathcal{T}(x^*(z))(y(z) - y_+(z)) = (M^T_\mathcal{T}(x(y_+(z), z)) - M^T_\mathcal{T}(x^*(z)))(y_+(z))_\mathcal{T} - p(x(y_+(z), z) - x^*(z)).$$

Therefore, taking norms to the above and the Lemma D.9, we have

$$\gamma\|(y_+(z))_\mathcal{T} - (y(z))_\mathcal{T}\| \leq \sqrt{m}L\|x(y_+(z), z) - x^*(z)\|\|(y_+(z))_\mathcal{T}\| + p\|x(y_+(z), z) - x^*(z)\|$$
$$\leq (\sqrt{m}L + p)\|x^*(z) - x(y_+(z), z)\|,$$

where the first inequality uses the Lipschitz-continuity of $\nabla_x f_i$ and the second is because $\|y_+(z)\| \leq 1$. Hence, we finish the proof with $\lambda = (p + \sqrt{m}L)/\gamma$. ∎

Then Lemma E.9 and Lemma 4.3 yield Theorem E.3.

### E.4 Proof of Proposition E.7

For a set $\mathcal{S} \subseteq [m]$, we define

$$M_{\mathcal{S}}(x; \ell_{\mathcal{S}}) = \left\{ J_{\mathcal{S}F(x;\ell_{\mathcal{S}})} \quad \mathbf{1} \right\},$$

where $J_{\mathcal{S}}F(x; \ell_{\mathcal{S}}) = ((\Psi_i(x) - \ell_i)\nabla_x\Psi_i(x) \mid i \in \mathcal{S})$.

**Proof** Define the event $\mathcal{E}_{\mathcal{T},\mathcal{P}}$ to be: there exists a solution $(x^*, y^*) \in W^*$, such that $M(x^*)$ is not of full row rank, $\mathcal{T}(x^*) = \mathcal{T}$ and $\Psi_i(x^*) - \ell_i \geq 0$ for $i \in \mathcal{P}$ and $\Psi_i(x^*) - \ell_i$ for $i \notin \mathcal{P}$. Then Proposition E.7 is equivalent to the claim:

$$\Pr(\cup_{\mathcal{T}\subseteq[m],\mathcal{P}\subseteq\mathcal{T}}\mathcal{E}_{\mathcal{T},\mathcal{P}}) = 0.$$

Since there are only finite choice of the sets $\mathcal{T}$ and $\mathcal{P}$, we only need to prove that for any $\mathcal{T} \subseteq [m]$ and $\mathcal{P} \subseteq \mathcal{T}$, $\mathcal{E}_{\mathcal{T},\mathcal{P}}$ holds with probability 0, Without loss of generality, we let $\mathcal{T} = \{1, 2, \cdots, k\}$ and $\mathcal{P} = \{1, 2, \cdots, p\}$ with $p \leq k$. We define $\delta_i$ for $i \in [k]$ as $\delta_i = 1$ for $i \in \mathcal{P}$ and $\delta_i = -1$ otherwise. Then if $\mathcal{E}_{\mathcal{T},\mathcal{P}}$ holds, there exists an $x^* = (x_1^*, \cdots, x_n^*)^T \in X^*$ and $x_{n+1} \in \mathbb{R}$, such that

1. $(x_1^*, \cdots, x_n^*)^T \in X^*$;
2. $x_{n+1}^* \geq 0$;
3. $\mathcal{T}(x^*) = \mathcal{T}$;
4. $\Psi_i(x^*) - \ell_i = x_{n+1}^* \geq 0$ for $i \in \mathcal{P}$ and $\Psi_i(x^*) - \ell_i = -x_{n+1}^* \leq 0$ for $i \notin \mathcal{P}$.
5. $M_{\mathcal{T}}(x_1^*, \cdots, x_n^*; \ell_1, \cdots, \ell_k)$ is row rank deficient.

Define $\bar{X}_{\mathcal{T},\mathcal{P}}^*(\ell_1, \cdots, \ell_k)$ to be the set of all $x^* \in X^*$ satisfying the above conditions. Consider the map $G: \mathbb{R}^{n+1} \to \mathbb{R}^k$ defined as

$$G(x_1, \cdots, x_{n+1}) = (\Psi_1(x_1, \cdots, x_n) - \delta_1 x_{n+1}, \cdots, \Psi_k(x_1, \cdots, x_n) - \delta_k x_{n+1})^T.$$

Then $G(x_1^*, \cdots, x_{n+1}^*) = (\ell_1, \cdots, \ell_k)$ for any $(x_1^*, \cdots, x_{n+1}^*)^T \in \bar{X}_{\mathcal{T},\mathcal{P}}^*(\ell_1, \cdots, \ell_k)$. Define the set $\bar{X}_{\mathcal{T},\mathcal{P}} \subseteq \mathbb{R}^{n+1}$ be the collection of all $(x_1, \cdots, x_{n+1})$ satisfying:

1. $x_{n+1} > 0$.
2. there exist $\bar{\ell}_1, \cdots, \bar{\ell}_k$ with $\Psi_i(x_1, \cdots, x_n) - \bar{\ell}_i = x_{n+1}$ for $i \in \mathcal{P}$ and $\Psi_i(x_1, \cdots, x_n) - \ell_i = -x_{n+1}$ for $i \notin \mathcal{P}$.
3. $M_{\mathcal{T}}(x_1, \cdots, x_n; \bar{\ell}_1, \cdots, \bar{\ell}_k)$ is rank deficient.
4. $(\bar{\ell}_1, \cdots, \bar{\ell}_k)^T \in \prod_{i=1}^k \mathcal{L}_i$.

Therefore, if $\mathcal{E}_{\mathcal{T},\mathcal{P}}$ holds, we have

$$(\ell_1, \cdots, \ell_m)^T \in (G(\bar{X}_{\mathcal{T},\mathcal{P}}) \cap \prod_{i=1}^k \mathcal{L}_i) \times \prod_{i=k+1}^m \mathcal{L}_i \cup \Omega.$$

For $(x_1, \cdots, x_{n+1})^T \in \bar{X}_{\mathcal{T},\mathcal{P}}$, notice that $JG(x_1, \cdots, x_{n+1})$ is attained by doing elementary matrix transformation to the matrix $M_{\mathcal{T}}(x_1, \cdots, x_n; \bar{\ell}_1, \cdots, \bar{\ell}_k)$, i.e., multiplying the first $k$ columns of $M_{\mathcal{T}}(x_1, \cdots, x_n; \bar{\ell}_1, \cdots, \bar{\ell}_k)$ by $1/x_{n+1}$ and multiplying the $k+1$-th column of $M_{\mathcal{T}}(x_1, \cdots, x_n; \bar{\ell}_1, \cdots, \bar{\ell}_k)$ by $-1$ and then multiplying the $i$-th row by $\delta_i$ for $i \in [n]$. Therefore, $M_{\mathcal{T}}(x_1, \cdots, x_n; \bar{\ell}_1, \cdots, \bar{\ell}_k)$ is also rank deficient.

Consequently, $G(x_1, \cdots, x_{n+1})$ with $(x_1, \cdots, x_{n+1})^T \in \bar{X}_{\mathcal{T},\mathcal{P}}$ is a critic value of $G$ (see [53]). Then by Sard's Theorem [53], $G(\bar{X}_{\mathcal{T},\mathcal{P}})$ is a zero measure set in $\mathbb{R}^k$. Hence, $G(\bar{X}_{\mathcal{T},\mathcal{P}}) \cap \prod_{i=1}^k \mathcal{L}_i$ is a zero measure set in $\prod_{i=1}^k \mathcal{L}_i$. Recall that if $\mathcal{E}_{\mathcal{T},\mathcal{P}}$ holds, we have

$$(\ell_1, \cdots, \ell_m)^T \in \mathcal{Z} = (G(\bar{X}_{\mathcal{T},\mathcal{P}}) \cap \prod_{i=1}^k \mathcal{L}_i) \times \prod_{i=k+1}^m \mathcal{L}_i \cup \Omega.$$

By the above analysis, $G(\bar{X}_{\mathcal{T},\mathcal{P}}) \cap \prod_{i=1}^k \mathcal{L}_i$ is a zero measure set in $\prod_{i=1}^k \mathcal{L}_i$. Hence, $(G(\bar{X}_{\mathcal{T},\mathcal{P}}) \cap \prod_{i=1}^k \mathcal{L}_i) \times \prod_{i=k+1}^m \mathcal{L}_i$ is a zero measure set in $\prod_{i=1}^m \mathcal{L}_i$. Also by Assumption E.5, $\Omega$ is a zero measure set in $\prod_{i=1}^m \mathcal{L}_i$. Consequently, $\mathcal{Z}$ is a zero measure set in $\prod_{i=1}^m \mathcal{L}_i$. Then by the continuity of the distribution of $\ell$, we finish the proof. ∎

# F Details in Experiments

Recall the procedure of training a robust neural network against adversarial attacks can be formulated as a min-max problem:

$$\min_{\mathbf{w}} \sum_{i=1}^{N} \max_{\delta_i, \text{ s.t. } |\delta_i|_\infty \leq \varepsilon} \ell(f(x_i + \delta_i; \mathbf{w}), y_i), \tag{F.1}$$

where $\mathbf{w}$ is the parameter of the neural network, the pair $(x_i, y_i)$ denotes the $i$-th data point, and $\delta_i$ is the perturbation added to data point $i$.

As (F.1) is nonconvex-nonconcave and thus difficult to solve directly, researchers introduce an approximation of (F.1) [20] where the approximated problem has a concave inner problem. The approximation is first replacing the inner maximization problem in F.1 with a finite max problem:

$$\min_{\mathbf{w}} \sum_{i=1}^{N} \max \{\ell(f(\hat{x}_{i0}(\mathbf{w}); \mathbf{w}), y_i), \ldots, \ell(f(\hat{x}_{i9}(\mathbf{w}); \mathbf{w}), y_i)\}, \tag{F.2}$$

where each $\hat{x}_{ij}(\mathbf{w})$ is the result of a targeted attack on sample $x_i$ by changing the output of the network to label $j$.

To obtain the targeted attack $\hat{x}_{ij}(\mathbf{w})$, we need to introduce an additional procedure. Recall the images in MNIST have 10 classifications, thus the last layer of the neural network architecture for learning classification have 10 different neurons. To obain any targeted attack $\hat{x}_{ij}(\mathbf{w})$, we perform gradient ascent for $K$ times:

$$x_{ij}^{k+1} = \text{Proj}_{B(x,\varepsilon)} \left[ x_{ij}^k + \alpha \nabla_x (Z_j(x_{ij}^k, \mathbf{w}) - Z_{y_i}(x_{ij}^k, \mathbf{w})) \right], \ k = 0, \cdots, K-1,$$

and let $\hat{x}_{ij}(\mathbf{w}) = x_{ij}^K$. Here, $Z_j$ is the network logit before softmax corresponding to label $j$; $\alpha > 0$ is the step-size; and $\text{Proj}_{B(x,\varepsilon)}[\cdot]$ is the projection to the infinity ball with radius $\varepsilon$ centered at $x$. Using the same setting in [20], we set the iteration number as $K = 40$, the stepsize as $\alpha = 0.01$, and the perturbation level $\epsilon$ chosen from $\{0.0, 0.1, 0.2, 0.3, 0.4\}$.

Now we can replace the finite max problem (F.2) with a concave problem over a probabilistic simplex, where the entire problem is non-convex in $w$, but concave in $\mathbf{t}$:

$$\min_{\mathbf{w}} \sum_{i=1}^{N} \max_{\mathbf{t} \in \mathcal{T}} \sum_{j=0}^{9} t_j \ell\left(f\left(x_{ij}^K; \mathbf{w}\right), y_i\right), \ \mathcal{T} = \{(t_1, \cdots, t_m) \mid \sum_{i=1}^{m} t_i = 1, t_i \geq 0\}. \tag{F.3}$$

We use Convolutional Neural Network(CNN) with the architecture detailed in Table 3 in the experiments. This setting is the same as in [20].

| Layer Type | Shape |
|---|---|
| Convolution + ReLU | $5 \times 5 \times 20$ |
| Max Pooling | $2 \times 2$ |
| Convolution + ReLU | $5 \times 5 \times 50$ |
| Max Pooling | $2 \times 2$ |
| Fully Connected + ReLU | 800 |
| Fully Connected + ReLU | 500 |
| Softmax | 10 |

**Table 3:** Model Architecture for the MNIST dataset.

The results are listed in Table 2. The first three lines are the results obtained from [20] and the fourth line is obtained by using the code provided in [20] to train their algorithm. As for comparison, we run our algorithm 2 for the same number of iterations (100 iterations) with parameter $p = 0.2, \beta = 0.8$ and $\alpha = 0.5$. In the experiment, to compute the projection of a vector of dimension $d$ over the probability simplex, we use the algorithm from [54] which has a complexity $\mathcal{O}(d \log d)$.

**Figure 2:** Convergence speed of Smoothed-GDA and the algorithm in [20] on CI-FAR10.

We also perform robust training on CIFAR10 [55] and comparing with the algorithm in [20] after 30 epochs. As shown in Figure 2, our algorithm still outperforms [20] in convergence speed. To obtain the targeted attack at each epoch, we set the iteration number $K$ as 10, the stepsize as $0.007$, and the perturbation level as $0.031$ which are the same settings appear in [51]. We achieve robust accuracy $38.5\%$ and testing accuracy $82.6\%$ which are comparable to the results from the literature [56] in robust training.