[Reviews · NeurIPS 2020]

Review 1

Summary and Contributions: This paper considers min max optimization problems with a particular focus on problems where the maximum is over a finite set of nonconvex functions. They show that adding a quadratic term to the objective dragging the minimization variables towards a weighted average of their past values gives a converge. ================ EDIT: I stand by my assessment that this paper should be an accept. I feel the authors' rebuttal and addresses all of my complaints and trust a small revision could handle them.

Strengths: This paper proposes a smoothed objective similar in theme to Moreau envelops that occur in typical minimization problems by adding a quadratic term to the minimization. Running GDA with this simple modification to the objective gives an algorithm which converges at the typical rate of O(\epsilon^-4) that one expects for GDA with some form of averaging. The core strength of this paper is in showing that a strict complementarity condition on the dual problem improves this rate to O(\epsilon^-2). This is a substantial improvement. The area of minimax optimization is certainly of great modern importance and even the restricted finite maximum form considered here has many practical applications.

Weaknesses: The arguments given against multi-loop methods seem quite weak. The paper says (without proof) that any multiloop method must have at least O(1/\eps^2) outer iterations. I see no reason why this must be the case. Its true that the multiloop methods referenced have this many outer iterations required, but nothing seems fundamental about that limit. A more clever choice of subproblem could resolve that issue. As written the algorithm requires knowledge of the optimization problems smoothness constant L to select the quadratic added and stepsizes used. An adaptive scheme for removing this dependence would improve the applicability and likely the performance of this method. The core result of the paper relies on strict complementarity holding and the primal iterates staying bounded. Neither of these assumptions are given much motivation in the main text. The key ideas of the argument in favor of complementarity should be included from Appendix E. Some motivation for the boundedness assumption is much needed. In particular, the claim in the contribution section that the paper avoids needing to assume the X domain is compact is undermined by subsequently assuming the primal iterates stay bounded. Once this has been assumed, one could intersect X with a large enough ball to make it compact without changing the iterates, so compactness is effectively assumed. This claim should be removed or put in full context of what will be later assumed.

Correctness: The proofs all appear to be correct and the experiments a reasonable justification of the proposed method's effectiveness.

Clarity: The paper is well-written, but the appendix is difficult to read (there is minimal prose describing what is being done in the proofs and connecting the ideas together).

Relation to Prior Work: The method gives adequate discussion of the recent works in nonconvex concave minimax optimization. Potentially more references could be given to the traditional smoothing literature in optimization (Moreau smoothing and Nesterov smoothing both being related to the technique used here).

Reproducibility: Yes

Additional Feedback:


Review 2

Summary and Contributions: The authors propose a simple single loop scheme for solving non-convex/concave min-max problems which achieves a O(1/epsilon^4) convergence rate for attaining an epsilon-stationary point. Subsequently, this paper focuses on a characteristic class of non-convex/concave (or more precisely non-convex/linear) problems with many important applications. In the latter class an O(1/epsilon^2) rate is achieved, which to the best of my knowledge is the best rate to date for this class of problems. Further, the utility of the algorithm is supported by experiments on a robust neural network training problem on MNIST.

Strengths: Among the elements that add value to this work is the fact that the proposed algorithm involves a single loop. This is important since a large number of known min-max algorithms entail solving sub-problems in each iteration, which leads to complicated double and triple loop schemes. Also, I believe that the focus (and rate improvement) on problem class (1.2) is worthwhile since it captures a wide range of important applications (e.g robust optimization over multiple domains, problems in fair machine learning). Finally, the relaxation of the compactness assumption for the constraint set X is appreciated since that would extend the applicability of the results to unconstrained problems. In conclusion, this work develops a simple single loop algorithm for non-convex/concave problems which improves the state of the art convergence rate for an important subclass of problems. However, the practicality of the assumptions (i.e., if they hold in a number of useful machine learning problems under reasonable scenarios) underlying the latter result is not completely clear. Therefore, I would suggest the authors to revise their work and resubmit it.

Weaknesses: 1. There is a recent work in single loop algorithms for non-convex/concave problems [A]. The latter work appeared on Arxiv on 06/03 and thus it is understandable that the authors were not aware of it at the time of submission. However, I think that it is important to see now how the proposed algorithm and the results of this paper compare to the ones given in [A]. Does this work offer any advantages compared to [A]? 2. One of my main concerns is about the strict complementarity assumption in the results of problem (1.2). The O(1/epsilon^2) result for the problem class (1.2) is the central one and so it is important to establish that this assumption is reasonable and holds in practice in a number of machine learning problems. The effort made on the supplementary material to show that a weaker version of strict complementarity (which still ensures the same convergence results) holds for a typical machine learning problem (robust regression) is appreciated. However, I am still not completely persuaded about the practicality of this assumption (either the strict complementarity or its weaker version). I believe that a wider range of examples (e.g machine learning problems) are required in order to illustrate the rationality of this assumption in practice. 3. The experiments are performed over the MNIST dataset. In my opinion, this does not offer sufficiently strong indications about the superiority of the proposed algorithm. Additional experiments potentially over more complex datasets are needed in order to support the utility and the advantages of the new algorithm.

Correctness: yes

Clarity: yes

Relation to Prior Work: Please see my comments above.

Reproducibility: Yes

Additional Feedback:


Review 3

Summary and Contributions: In this paper, the authors propose a single loop algorithm, they denote as Smoothed-GDA, for solving non-convex concave min-max optimization problems. The proposed method alternatively performs gradient descent and gradient ascent steps on the objective function with an added quadratic proximal term. The authors show that the algorithm finds an epsilon-stationary solution in at least O(epsilon^{-4}). Once applied to minimizing the pointwise maximum of a finite collection of non-convex functions, the algorithm achieves an O(epsilon^{-2}) iteration complexity. The authors further extend the Smoothed-GDA to the multi-block setting while achieving the same convergence rates.

Strengths: The authors used a smoothing technique to propose a variant of gradient descent-ascent (GDA) algorithm for solving non-convex concave min-max optimization problems. Due to its useful applications, solving the former class of problems in the non-convex setting has recently gained significant attention. Hence, the topic under study belongs to a vibrant field of research. In contrast to the (non)-convex strongly-concave case, GDA exhibits an oscillating behavior when the inner maximization problem is concave. To stabilize this oscillating behavior, the authors introduced a smoothing technique by adding a quadratic proximal term to the objective that uses an auxiliary sequence that is updated at every iteration. The authors established convergence of the algorithm and computed the iteration complexity in non-convex concave settings. To show convergence, the paper proposed a novel potential function and showed sufficient decrease at every iteration. When applied to the problem of minimizing the pointwise maximum of a finite collection of non-convex functions, the algorithm converges with optimal rate. The claims and theoretical results seem to be correct. Moreover, the proof technique used to establish the theoretical results are involved. The authors managed to present the ideas in a concise and organized manner.

Weaknesses: 1. The title used for the paper is misleading since the rate O(epsilon^{-2}) can only be achieved for a special class of non-convex concave min-max problems. More specifically, this rate is attained when using the algorithm to minimize the pointwise maximum of finite collection of non-convex functions. For general non-convex concave the rate is O(epsilon^{-4}). This is clarified in the body of the paper, but the title is clearly misleading. 2. To prove the result for the pointwise maximum case, the authors require strict complementarity assumption (Assumption 3.4). It is not clear in the paper when this assumption (or even its relaxed replacement in the appendix) holds in practice. I believe the authors should discuss methods to check whether this assumption holds for a given objective function. 3. Also looking at assumption 3.5, does it imply that the domain X is bounded? It is stated in the contributions that the compact of the domain X is relaxed. However, it is not clear whether assumption 3.5 affects that claim. 4. The authors mentioned that relaxing the compactness of the domain X significantly extends the applicability of the algorithm. I believe this statement should be clarified through practical examples. 5. The proof of lemma B.8 is missing. Below are some minor comments: 1. The authors state that multi-block algorithms cannot be easily adapted to solve problems with the multi-block structure due to the acceleration steps. The acceleration steps in such algorithms are mainly used to improve the iteration complexity and hence can be relaxed. If relaxed, existing algorithms can be easily adapted to solve problems with multi-block structure. 2. It is not clear how B.10 is attained. 3. Line 21 (Typo): convex in x and concave in y. 4. Line 107 (Typo): category instead of categories. 5. Algorithms 1, 2 and 3: the projection operator is not defined in the main body. 6. The proximal function can be confused with the projection operator. 7. Line 378: missing bracket ')'. 8. Lemma B.8: continuous 'in' instead of 'of. 9. Lemma B.10: missing bracket '('. 10. Lemma B.11: lambda should be replaced with \bar{lambda}. 11. Proof of Lemma B.11: in the first expression we should have 2c instead of c. Also the expression under line 417 has a typo. 12. Expression B.48: the label covers part of the inequality. 13. New line not needed on line 445. 14. Line 458: additional 'the'. 15. Line 512 (typo): M(x) instead of M(x)(x). 16. Line 513: should we have I{y^star} instead of I. 17. (D.24): Should we have an added vector [0, ..., 0, 1]. 18. Line 542: Should lambda in sigma_5 be in the numerator?

Correctness: The claims and theoretical results seem to be sound and correct. The empirical efficiency was illustrated by applying the algorithm to a robust neural network training application. In comparing the convergence speed of the proposed algorithm compared to [20], the authors did not mention whether the same step sizes were used in both methods which raises a concern on the viability of the comparison. To illustrate the efficiency of the algorithm, the authors applied the

Clarity: The main body of the paper is concise and clearly written. However, the proof in the supplementary material has many typos and one proof is missing.

Relation to Prior Work: The related material are referenced and well-discussed in the paper. The authors clearly positioned their work in the related field and discussed their contributions in comparison to other similar works.

Reproducibility: Yes

Additional Feedback: Post Rebuttal: Thank your for the author's response. After reading the rebuttal, my main concerns on the title name and compactness of the domain were alleviated.

[Author Response · NeurIPS 2020]

1 We thank the reviewers for their valuable feedback. We first address a general comment.

2 **1. Strict complementarity assumption:** The strict complementarity assumption is used in many papers [Forsgren et al., SIAM Rev.], [Carbonetto et al.,NIPS'08], [Liang et al.,NIPS'14], [Namkoong et al., NIPS'16], [Lu et al, arxiv:1907.04450]. It is generically true (i.e. holds with probability 1) if there is a linear term in the objective function and the data is from a continuous distribution (similar to [Lu et al., arxiv:1907.04450], [29]). The weaker assumption mentioned in the appendix generically holds for robust regression problems with square loss.

7 **Response to reviewer 1:**

8 **1.** *Arguments of multi-loop methods seem quite weak?* Indeed the statement for multi-loop methods is not a rigorous claim. It is an observation that the existing algorithms all require $O(\epsilon^{-2})$ outer iterations. We will replace this statement by "the existing works on multi-loop algorithms require $\mathcal{O}(\epsilon^{-2})$ outer iterations".

11 **2.** *Adaptive version of this algorithm is more applicable in practice.* Thank you for your suggestion. We agree and will address this issue in the future work.

13 **3.** *a) Move claims of complementarity to main text. b) Motivation for the boundedness assumption is much needed.* We will move the claims as you suggest. As mentioned in Line 186, the bounded-level-set assumption is to ensure the iterates stay bounded. The bounded level set assumption, a.k.a. coerciveness assumption, is widely used in many papers [Cannelli1 et al., arXiv:160704818], [Hong et al., ICML'17], [Cannelli1 et al., arXiv:160704818]. Bounded-iterates-assumption itself is common in optimization too (see [Xu et al., arxiv:1408.2597], [Défossez et al., arxiv:2003.02395], and [Carbonetto et al., NIPS'08] ). In practice, people usually add a regularizer to the objective function to make the level set and the iterates bounded (see, e.g., [Liang et al., arxiv:1912.13472] for a neural network example). We will include these motivations.

21 **4.** *Appendix is hard to read.* We included a proof sketch in Appendix B; will move to main text and add more intuition.

22 **5.** *More reference on traditional smoothing literature.* We will add references on Moreau smoothing (e.g. Section 3.1 in the book on Proximal Algorithms by Parikh and Boyd]) and Nesterov smoothing [Nesterov, Math. Program., 2005].

24 **6.** *claim of contribution on removing compact X should be removed or put in full context of what will be later assumed.* Thanks for the comment. We will add a sentence in the introduction near the claim: note that we do not need any boundness assumption for general non-convex-concave problem, but we still have an assumption of bounded iterates (which is weaker than compact $X$) for the point-maximum problems.

28 **Response to reviewer 2:**

29 **1.** *Comparison to recent work [A].* Thank you for your comment. We were not aware of it. [A] proposes a single-loop algorithm for min-max problems by performing GDA to a regularized version of the original min-max problem. Using primal-dual analysis, they also prove that their algorithm attains an $\epsilon$-solution with $\mathcal{O}(\epsilon^{-4})$ iterations for nonconvex-concave problems as in our first result. Their paper differs from ours in two aspects: i) The algorithms are different: we add a proximal term centered at the auxiliary variable $z$, while [A] adds a regularization term $\alpha_t(\|x\|^2 - \|y\|^2)$ that is diminishing ii) They do not prove $\mathcal{O}(\epsilon^{-2})$ complexity for the pointwise maximum case. We will add a discussion.

35 **2.** *Practicality of strict complementarity assumptions.* Please refer to the discussion of the assumptions at the beginning of this response. Also, we will further study whether we can remove or relax these assumptions in the future.

37 **3.** *Additional experiments?* We have run more experiments on CIFAR10. Our algorithm achieve similar or better accuracy than [20] on the robust training tasks under different perturbation levels. As for training speed, we takes only 9 epochs to reach loss value 0.01, while the other algorithm in [20] takes at least 30 epochs to reach the same value.

40 **Response to reviewer 3:**

41 **1.** *Title is misleading.* Thank you for your comment. We will change the title to "A single-loop smoothed gradient descent-ascent algorithm for nonconvex-concave min-max problems". Further comment is welcome.

43 **2.** *Strict complementarity assumption holds in practice? How to check?* Please refer to the discussion of the assumptions at the beginning of this response and Appendix E. It is a common assumption and can not be directly checked.

45 **3-(a)** *How Assumption 3.5 impacts the claim of contribution of removing compact X?* Assumption 3.5 is indepenent of that claim. The claim "current algorithm does not require compact $X$" is for "general nonconvex-concave problems", as stated in the beginning of that paragraph; but Assumption 3.5 is for pointwise maximum problem class. We will clarify this; see Response 6 to Reviewer 1.

49 **3-(b)** *Does assumption 3.5 imply compact X?* There are examples for which X is unbounded while the level set is bounded. For example, consider $\min_x \max_{i\in[m]} f_i(x)$ where $f_i(x) = \|A_i x - b_i\|^2 + \sum_{j=1}^n x_j^2/(1 + x_j^2)$ with $[A_1, A_2, ..., A_m]$ full column rank. Here the domain $X = \mathbb{R}^n$ is not compact but $\max_i f_i(x)$ has bounded level sets.

52 **4.** *Does relaxing the compactness of X extend the applicability of the algorithm?* i) There exist applications; for the example in 3-(b): our result still applies, but [24] does not directly apply. ii) Anyhow, we change it to "However, the current algorithm does not require the compactness of the domain $X$" since we did not intend to list many specific applications for this point.

56 **5.** *Proof of Lemma B.8 is missing.* Thanks for pointing it out. We will add it in the revised version.

57 **6.** *Typos.* We really appreciate your careful reading and pointing out these typos. We will address them in revised version.

[Meta-Review · NeurIPS 2020]

Originally, the paper got three positive scores: 7,7,6, all with very high confidences. Basically, there was no critical issues raised by the reviewers, except suggesting the enhancement on experiments and clarifying some points. During discussion, all the reviewers agreed that the paper should be accepted and Reviewer #3 raised his/her score to 7. So all the reviewers reached a consensus. Thus the AC decided to accept the paper. However, the AC also have the following comments after a quick reading. Hope the authors could take into account when revising the paper. 1. The main concern is the convergence results (Theorems 3.3 and 3.6) in this paper. The proposed algorithms including Algorithms 2 and 3 are designed to solve the problem (3.2), which is different from Problem (1.1) or Problem (1.2). What’s the gap between them? In particular, the results in Theorems 3.3 and 3.6 hold due to the strong convexity. However, this important assumption is missing. 2. The complexity comparison in Table 1 is questionable. As stated in this paper, both the proposed algorithm and the algorithm in [24] have the same iteration complexity, O(1/\epsilon^4), for solving general nonconvex-concave problems. Therefore, the authors should report the results for a fair comparison. 3. For the same class of nonconvex-concave problems, both the proposed algorithm and the algorithm in [24] have an identical convergence rate. What’s the advantage of the proposed algorithm? 4. There is an important parameter p in the model (3.2). How to choose it for the theoretical results and experiments? 5. The authors introduce an auxiliary variable z and a momentum acceleration step for the update of z. It is not clear that what the contribution of the momentum acceleration is. 6. In Line 142, what’s \beta? And what’s the difference between \beta and \beta_t in Algorithms 2 and 3, as well as in Theorem 3.3 ad Theorem 3.6? 7. The experimental analysis is less convincing. The authors should give the experimental results of the algorithms in the related work [22, 23, 24], which are used to verify the convergence results and advantages of the proposed algorithm. 8. The authors should define some symbols, e.g., what’s \beta_t in Algorithms 2 and 3. 9. There are many key errors and typos. Line 22, “inof” should be “of”. Line 87, “Section 5” should be “Section 2”. Line 144, “see 3” should be “see Algorithm 3”.